# UNRAVELING MODEL-AGNOSTIC META-LEARNING VIA THE ADAPTATION LEARNING RATE

**Yingtian Zou, Fusheng Liu, Qianxiao Li**
National University of Singapore, Singapore
{yingtian, fusheng}@u.nus.edu, qianxiao@nus.edu.sg

## ABSTRACT

Model-Agnostic Meta-Learning (MAML) aims to find initial weights that allow fast adaptation to new tasks. The adaptation (inner loop) learning rate in MAML plays a central role in enabling such fast adaptation. However, how to choose this value in practice and how this choice affects the adaptation error remains less explored. In this paper, we study the effect of the adaptation learning rate in meta-learning with mixed linear regression. First, we present a principled way to estimate optimal adaptation learning rates that minimize the population risk of MAML. Second, we interpret the underlying dependence between the optimal adaptation learning rate and the input data. Finally, we prove that compared with empirical risk minimization (ERM), MAML produces an initialization with a smaller average distance to the task optima, consistent with previous practical findings. These results are corroborated with numerical experiments.

## 1 INTRODUCTION

*Meta-learning* or *learning to learn* provides a paradigm where a machine learning model aims to find a general solution that can be quickly adapted to new tasks. Due to its fast adaptability, meta-learning has been widely applied to challenging tasks such as few-shot learning (Vinyals et al., 2016; Snell et al., 2017; Rusu et al., 2018), continual learning (Finn et al., 2019; Javed & White, 2019), and neural architecture search (Zhang et al., 2019; Lian et al., 2019). One promising approach in meta-learning is Model-Agnostic Meta-Learning (MAML) (Finn et al., 2017), which consists of two loops of optimization. In the outer loop, MAML aims to learn a good *meta-initialization* that can be quickly adapted to new task in the inner loop with limited adaptation (parameter optimization) steps. The double loops optimization serve as "learning-to-adapt" process, thus enabling the trained model to adapt to new tasks faster than direct Empirical Risk Minimization (ERM) algorithms (Finn et al., 2017; Raghu et al., 2020). Recent works (Nichol et al., 2018; Fallah et al., 2020; Collins et al., 2020; Raghu et al., 2020) attribute the fast adaptability to the phenomenon that the learned meta-initialization lies in the vicinity of all task solutions. However, the theoretical justification of this empirical statement, and more generally how fast adaptability of MAML depends on the inner loop optimization remains unclear. As a key component of MAML, the adaptation (inner loop) learning rate (hereafter called $\alpha$) is shown empirically to plays a crucial role in determining the performance of the learned meta-initialization (Rajeswaran et al., 2019). In particular, the value of $\alpha$ bridges ERM and MAML, in the sense that the latter reduces to the former when $\alpha = 0$. However, from a theoretical viewpoint, the dependence of MAML performance on the choice of $\alpha$ remains unclear, and furthermore, there lacks a precise practical guideline on how to pick a near-optimal value.

In this paper, we address these issues by answering the following two questions: (1) *How to choose the optimal $\alpha$ that minimizes population risk of MAML?* (2)*What is the effect of $\alpha$ on fast adaptability of MAML?* To this end, we consider the mixed linear regression problem with random feature models. For the first question, we derive the optimal $\alpha$ which minimizes the population risk of MAML in the limit of an infinite number of tasks. This can then be used to estimate an effective $\alpha$ prior to training. Moreover, we analyze the underlying statistical dependency between the optimal $\alpha$ and the input data, e.g. relation to the moments of data distribution. This in turn allows the heuristic application of our results beyond linear models, and we demonstrate this with experiments. To answer the second question, we compare MAML with an ERM algorithm (without inner loop optimization) in order to reflect the effect of $\alpha$ in optimization. As stated in many works, like Nichol et al. (2018), that

meta-initialization learned by MAML in parameter space is close to all training tasks thus contributes to fast adaptability. We conduct an experiment and observe that MAML with a not too large $\alpha$ yields a shorter mean distance to task optima than ERM. To justify this empirical finding, we define a metric measuring the expected geometric distance between the learned meta-initialization and task optima. We prove that in our setting, the MAML solution indeed possess a smaller value of this metric compared with that of ERM for small $\alpha$, providing theoretical evidence for the observed phenomena. Our contributions can be summarized as follows:

- We provide a principled way to select the optimal adaptation learning rate $\alpha^*$ for MAML which minimizes population risk (Theorem 1 & Proposition 1). We also interpret the underlying statistical dependence of $\alpha^*$ to input data (Corollary 1) with two examples.

- We validate the observation that MAML learns a good meta-initialization in the vicinity of the task optima, which reveals the connection between the adaptation learning rate $\alpha$ and the fast adaptability in optimization. (Theorem 2)

- We also extend our result about the choice of $\alpha^*$ to more practical regime, including deep learning. All of our theoretical results are well corroborated with experimental results.

## 2 PROBLEM FORMULATION

We study the MAML algorithm under the mixed linear regression setting. Suppose we have a task $T$ that is sampled from the distribution $\mathcal{D}(T)$. Each task $T$ corresponds to a linear relationship

$$\boldsymbol{y}_T = \Phi(X_T)\boldsymbol{a}_T, X_T = \begin{pmatrix} - & \boldsymbol{x}_{T,1} & - \\ & \cdots & \\ - & \boldsymbol{x}_{T,K} & - \end{pmatrix}, X_T \in \mathbb{R}^{K \times d_x}, \Phi(X_T) \in \mathbb{R}^{K \times d}, \boldsymbol{a}_T \in \mathbb{R}^d.$$

(2.1)

where $X_T \in \mathbb{R}^{K \times d_x}$ is the input data of task $T$ which has $K$ vector samples $\{\boldsymbol{x}_{T,1}, ..., \boldsymbol{x}_{T,K}\}, \boldsymbol{x}_{T,j} \in \mathbb{R}^{d_x}$ i.i.d sampled from $\mathcal{D}(\boldsymbol{x})$ [1]. For each input data, we have a mapping $\phi : \mathbb{R}^{d_x} \to \mathbb{R}^d$ transform each point of $X_T$ from input data space $\mathbb{R}^{d_x}$ to a $d$-dimensional feature space $\mathbb{R}^d$ where we denote the transformation of all data in task $T$ by $\Phi(X_T) = [\phi(X_{T,1}), ..., \phi(X_{T,K})]^\top$ as the *feature* of that task. Then, we assume optimal solution $\boldsymbol{a}_T \in \mathbb{R}^d$ for task $T$ is i.i.d sampled from $\mathcal{D}(\boldsymbol{a})$. The corresponding label $\boldsymbol{y}_T \in \mathbb{R}^K$ can be obtained from (2.1).

Our target is to learn a model to minimize the risk of different tasks across $\mathcal{D}(T)$. Note that each task $T$ is determined by a feature-solution pair $(\Phi(X_T), \boldsymbol{a}_T)$. Therefore, we can formulate this multi-task problem with parameter space $\mathbb{R}^d$ and loss function $\ell$ as

$$\min_{\boldsymbol{w} \in \mathbb{R}^d} \mathbb{E}_{T \sim \mathcal{D}(T)} \left[ \ell\left(\boldsymbol{w}; T\right) \right] = \min_{\boldsymbol{w} \in \mathbb{R}^d} \mathbb{E}_{\boldsymbol{a} \sim \mathcal{D}(\boldsymbol{a})} \mathbb{E}_{X \sim \mathcal{D}(\boldsymbol{x})} \left[ \ell\left(\boldsymbol{w}; \Phi(X), \boldsymbol{a}\right) \right]$$

(2.2)

To solve this problem, ERM and MAML algorithms yield different iterations. Specifically, ERM uses all data from all tasks to directly minimize the square error loss $\ell$, such that population risk of ERM is

$$\mathcal{L}_r(\boldsymbol{w}, K) := \mathbb{E}_{\boldsymbol{a} \sim \mathcal{D}(\boldsymbol{a})} \mathbb{E}_{X \sim \mathcal{D}(\boldsymbol{x})} \frac{1}{K} \left\| \Phi(X)\boldsymbol{w} - \Phi(X)\boldsymbol{a} \right\|_2^2$$

(2.3)

As a counterpart, MAML first adapts with an adaptation learning rate $\alpha$ on each task using its training set – a subset of task data in the inner loop. Then, in the outer loop, MAML minimizes the evaluation loss for each adapted task-specific solution using a validation set. For simplicity, since data is i.i.d sampled from the same distribution, we first consider the setting where all data in each task is used as training set and validation set in our main results. We present later the extension of these results to the case with a different train-validation split. (Please refer to Appendix H.1)

Thus, the general population risk of one-step MAML is defined by

$$\mathcal{L}_m(\boldsymbol{w}, \alpha, K) := \mathbb{E}_{\boldsymbol{a} \sim \mathcal{D}(\boldsymbol{a})} \mathbb{E}_{X \sim \mathcal{D}(\boldsymbol{x})} \frac{1}{K} \left[ \ell\left( \underbrace{\boldsymbol{w} - \alpha \nabla_{\boldsymbol{w}} \ell\left(\boldsymbol{w}; \Phi(X), \boldsymbol{a}\right)}_{\text{Inner Loop}}; \Phi(X), \boldsymbol{a} \right) \right]$$

(2.4)

---

[1] For simplicity, we denote this sampling and stacking multiple examples to a matrix process as $X \sim \mathcal{D}(\boldsymbol{x})$

In practice, we use the empirical objective function as a surrogate objective function. We first sample $N$ tasks with task optima $\{\boldsymbol{a}_1, ..., \boldsymbol{a}_N\}$ from $\mathcal{D}(\boldsymbol{a})$ and then sample $K$ data for each task. Then, the empirical risk of MAML can be specified as $\hat{\mathcal{L}}_m$

$$\hat{\mathcal{L}}_m(\boldsymbol{w}, \alpha, N, K) := \frac{1}{NK} \sum_{i=1}^{N} \left\| \Phi(X_i) \boldsymbol{w}_i' - \Phi(X_i) \boldsymbol{a}_i \right\|_2^2 \tag{2.5}$$

where $\boldsymbol{w}_i' = \left[ \boldsymbol{w} - 2\alpha \Phi(X_i)^\top \left( \Phi(X_i) \boldsymbol{w} - \Phi(X_i) \boldsymbol{a}_i \right) / K \right]$ is adapted parameters of task $i$ after inner loop. Correspondingly, we apply ERM algorithm to the same problem by removing inner loop (setting $\alpha = 0$), thus the empirical risk of ERM is denoted as $\hat{\mathcal{L}}_r(\boldsymbol{w}, N, K)$. In addition, we follow the original MAML (Finn et al., 2017) to use the same $\alpha$ for training and testing.

**Notation** We denote an optimal adaptation learning rate as $\alpha^*$. Global minima of empirical risk of MAML and ERM (when they are unique) are denoted by $\boldsymbol{w}_m, \boldsymbol{w}_r$. We write $\{1, ..., N\}$ as $[N]$ and use $\| \cdot \|$ to denote the Euclidean norm. We use subscripts to index the matrices/vectors corresponding to task instances, and bracketed subscripts to index the entries of matrices. Other notations are summarized in Appendix Table 1.

**Assumption 1** (Normalization). *For simplicity, we consider a centered parameter space such that* $\mathbb{E}_{\boldsymbol{a} \sim \mathcal{D}(\boldsymbol{a})}[\boldsymbol{a}] = \boldsymbol{0}$ *and* $Var[\boldsymbol{a}] = \sigma_a^2$.

**Assumption 2** (Bounded features). *With probability 1, the covariance matrix of input features* $\Phi(X)^\top \Phi(X)$ *is positive definite and has uniformly bounded eigenvalues from above by* $\lambda_S > 0$ *and below by* $\lambda_I > 0$.

## 3 MAIN RESULTS

In this section, we analyze MAML through the adaptation learning rate $\alpha$. Our derived insights are summarized into three theoretical results: (1) The estimation of an optimal adaptation learning rate $\alpha^*$ which minimizes MAML population risk; (2) The statistical meaning of $\alpha^*$ in terms of the data distribution, and (3) The geometric interpretation of the effect of $\alpha$ on fast adaptability of MAML compared to ERM.

### 3.1 ON THE OPTIMAL ADAPTATION LEARNING RATE $\alpha^*$

We focus on the underparameterized case ($K \geq d$). Given the empirical objective functions $\hat{\mathcal{L}}_r, \hat{\mathcal{L}}_m$ defined in (2.5), we can derive the global minima by the first-order optimality condition. We obtain the global minimum of ERM $\boldsymbol{w}_r$ and minimum of MAML $\boldsymbol{w}_m$ in the following closed-forms,

$$\boldsymbol{w}_r = \boldsymbol{w}_r \left( \{\Phi(X_i), \boldsymbol{a}_i\}_{i \in [N]} \right) = \left( \sum_{i \in [N]} \Phi(X_i)^\top \Phi(X_i) \right)^{-1} \left( \sum_{j \in [N]} \Phi(X_j)^\top \Phi(X_j) \boldsymbol{a}_j \right)$$

$$\boldsymbol{w}_m(\alpha) = \boldsymbol{w}_m \left( \{C_i(\alpha), \boldsymbol{a}_i\}_{i \in [N]} \right) = \left( \sum_{i \in [N]} C_i(\alpha)^\top C_i(\alpha) \right)^{-1} \left( \sum_{j \in [N]} C_j(\alpha)^\top C_j(\alpha) \boldsymbol{a}_i \right) \tag{3.1}$$

where $C_i(\alpha) = \Phi(X_i) \left[ I - (2\alpha/K)\Phi(X_i)^\top \Phi(X_i) \right], C_i(\alpha) \in \mathbb{R}^{K \times d}$ can be viewed as the adapted feature of task $i$. Observe that $\boldsymbol{w}_m(\alpha)$ (and thus the MAML algorithm) depends on $\alpha$. If $\alpha = 0$, MAML reduces to ERM. For large $\alpha$, instabilities may occur, thus there may exist an optimum, $\alpha^*$ that minimizes the MAML population risk. The later intuition is worthwhile to be proved, from which we do not have a principled way to guide the choice of optimal hyperparameter $\alpha^*$ for MAML so far. To this end, we focus on the generalization error by taking the population risk on the global minimum of empirical risk. In particular, we consider the population risk of the MAML optimizer in the average sense, where the average population risk is

$$\bar{\mathcal{L}}_m(\alpha, N, K) = \mathbb{E}_{\boldsymbol{w}_m} \mathcal{L}_m(\boldsymbol{w}_m, \alpha, K) \tag{3.2}$$

whose minimizer we denote as $\alpha^*(N, K)$. In this way, we eliminate randomness of the global minimum $\boldsymbol{w}_m$ learned from sampled tasks. The following theorem gives a precise value of $\alpha^*(N, K)$ in the limit $N \to \infty$.

**Theorem 1.** *Under assumptions 1 & 2, we have as $N \to \infty$, $\alpha^*(N, K) \to \alpha^*_{lim}(K)$, where*

$$\alpha^*_{lim}(K) = \frac{K \operatorname{tr}[\mathbb{E}_X[(\Phi(X)^\top \Phi(X))^2]]}{2 \operatorname{tr}[\mathbb{E}_X[(\Phi(X)^\top \Phi(X))^3]]}, \tag{3.3}$$

$\Phi(X) \in \mathbb{R}^{K \times d}$, *K is the sample size per task and N is the number of tasks.*

The proof is found in Appendix B. In this theorem, we give the nearly optimum $\alpha^*_{lim}$ which is an alternative form for true optimal $\alpha$, namely $\alpha^*$, to minimize the MAML generalization error. As dictated in (3.3), the desired $\alpha^*$ is determined by the feature covariance matrix in expectation.

**Remark.** *The precise derivation of the case where N is finite is complicated, thus we derive the limiting case here as an estimator of true $\alpha^*$. Our estimation $\alpha^*_{lim}$ is the unique minimum. We will show later that this allows us to compute near optimal values efficiently in practice, each of which is close to the optimal $\alpha^*(N, K)$ in corresponding problem.*

**Remark.** *The estimator (3.3) can be generalized to different scenarios. For overparameterized models, we obtain a similar result for the minimum norm solution if the number of tasks N is limited ($NK \ll d$). Further, we show a computationally efficient estimator (H.15) in Appendix H.2. For deep learning, we can compute a range of effective $\alpha$ values based on $\alpha^*_{lim}$. We also give the numerical form when the training data is different from the test data in each task. These are presented in Appendix H.4 and H.1 respectively.*

In the above we considered the average population risk (3.2). This simplifies the calculations of finding the $\alpha^*$. Below, we justify this simplification by showing that in the limit of large number of tasks, the average population risk is a good estimate of the true population risk.

**Proposition 1** (Informal). *Assume $\boldsymbol{u} = C(\alpha)^\top C(\alpha)\boldsymbol{a}$ is sub-gaussian random variable with sub-gaussian norm $\|\boldsymbol{u}_{(i)}\|_{\Psi_2} \leq L$, assumption 1 & 2 hold, then with probability at least $1 - \delta$ that*

$$\left| \mathcal{L}_m(\boldsymbol{w}_m, \alpha, K) - \bar{\mathcal{L}}_m(\alpha, N, K) \right| \leq \frac{L^2}{K} \max \left\{ \sqrt{\frac{d\varepsilon(\alpha, K)}{N^2} \log \frac{2}{\delta}}, \frac{\varepsilon(\alpha, K)}{N} \log \frac{2}{\delta} \right\} \tag{3.4}$$

*where $\varepsilon(\alpha, K) = \mathcal{O}(1/(c_0 + \alpha)^2)$. Here $c_0 > 0$ is a constant and d is the feature size.*

The proof is found in Appendix C. Proposition 1 complements Theorem 1 by guaranteeing that the gap between the average population risk and population risk with same argument $\alpha$ will disappear along with $N$ goes to infinity. Large $\alpha$ makes the bound tighter while small $\alpha$ makes $\varepsilon(\alpha, K)$ converge to a positive constant; thus (3.4) provides a non-vacuous bound with regard to $\alpha$. Hence, it is justified to make an estimation of $\alpha^*$ using the average case. By Theorem 1 and Proposition 1, we give an explicit form to estimate $\alpha^*$ for MAML where this estimation $\alpha^*_{lim}$ is not too far from the true $\alpha^*$ of a specific case. Later experiments show our estimation $\alpha^*_{lim}$ is close to true $\alpha^*$ in both underparameterized and overparameterized models (see Section 5.1). This is meaningful for selecting an $\alpha^*$ minimizing MAML risk, instead of randomly choosing it. Previous work (Bernacchia, 2021) explores on this by giving a range of $\alpha^*$ may exist for the linear model. Instead, we show a fine result that we provide a certain value estimator of $\alpha^*$. (Details refer to Appendix H.5)

**Relation to data distribution.** After estimating the value of $\alpha^*$ through Theorem 1, we are now interested in the statistical interpretation of $\alpha^*$. In particular, we aim to summarize the dependence of an estimation of $a^*$ on the distribution of the inputs and tasks. This in turn allows us to devise strategies for choosing near optimal $\alpha$ for MAML beyond the simple settings considered here.

**Corollary 1.** *With a feature mapping $\phi : \mathbb{R}^{d_x} \to \mathbb{R}^d$ for each data $\boldsymbol{x} \in \mathbb{R}^{d_x}$, the $\alpha^*_{lim}$ in Theorem 1 will satisfy the following inequality*

$$\frac{1}{2d\sigma^2(\phi(\boldsymbol{x}_1), \ldots, \phi(\boldsymbol{x}_K))} \leq \alpha^*_{lim} \leq \frac{d}{2\sigma^2(\phi(\boldsymbol{x}_1), \ldots, \phi(\boldsymbol{x}_K))} \tag{3.5}$$

*where $\sigma^2(\phi(\boldsymbol{x}_1), \ldots, \phi(\boldsymbol{x}_K))$ is variance of the feature.*

See proof in Appendix D. According to Corollary 1, we can see that $\alpha^*_{lim}$ is bounded by the statistics of the input data. These bounds are governed by the standard derivation terms. More specifically, our estimator (3.3) holds an inverse relationship to higher order moment of data distribution while its

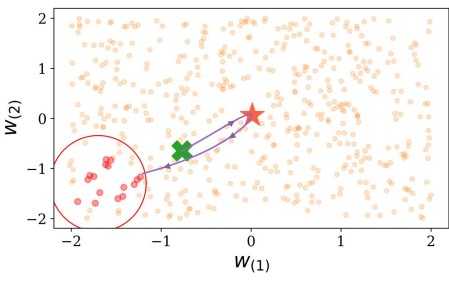
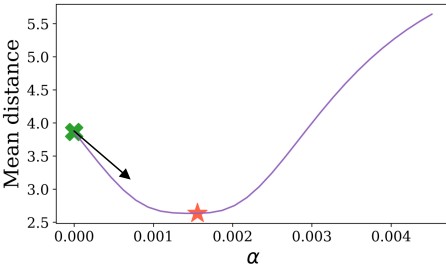

(a) Visualization of solutions and trajectory     (b) Mean solution distances

Figure 1: (a) Visualization of trajectory of MAML solution $\boldsymbol{w}_m(\alpha)$. Orange dots are task optima $\{\boldsymbol{a}_i\}_{[N]}$ of sampled tasks, where location of $\boldsymbol{a}_i$ is decided by its entries. Red dots highlighted in red circle are newly coming tasks. Green cross is $\boldsymbol{w}_r, (\alpha = 0)$ while the purple trajectory is generated as $\alpha$ increasing. Red star is $\boldsymbol{w}_m(\alpha^*_{lim})$. (b) Average euclidean distances of $\boldsymbol{w}_m(\alpha)$ and $\{\boldsymbol{a}_i\}_{[N]}$ display corresponding points in left figure. Black arrow is the tangent line. Best viewed in colors.

bounds (3.5) have an inverse relationship to data variance. As a consequence, the $\alpha^*_{lim}$ for different problems mainly depend on the standard derivations. For example, $\alpha^*_{lim}$ and thus $\alpha^*$ will shrink to zero as the variance of data increases and vice versa. In other words, small $\alpha$ is tailored to those tasks with large data variance when the model size is fixed. To illustrate the insight more clearly, we present two examples – regression with polynomial basis functions (Example 1) and the case where $\Phi(X)$ is a random matrix with a prescribed distribution (see Appendix E). In this following example, we narrow the range and get the exact relationship where the expression of $\alpha^*_{lim}$ rather than its bounds depend on data variance and model size $d$. In later experiments, we also validate this relationship on various models with different basis functions.

**Example 1** (Polynomial basis function). *Assume we have $K$ i.i.d samples $x_1, ..., x_K \sim \mathcal{N}(0, \sigma^2)$ for each task. Consider polynomial basis function $\phi : \mathbb{R} \to \mathbb{R}^d$, where $\phi(x) = (1, ..., x^{d-1})$. Then value of $\alpha^*_{lim}$ has an inverse relationship to $\sigma^2$ and dimension $d$ (Proof is in Appendix E).*

### 3.2 GEOMETRIC INTERPRETATION OF MAML ADAPTATION

In another direction, we aim to investigate geometric properties of the meta-initialization learned by MAML as $\alpha$ varies. In previous experimental investigations, it is suggested that MAML learns the near meta-initialization to all tasks Nichol et al. (2018) or trade-offs on easy and hard tasks Fallah et al. (2020). We can also observe the new phenomena in toy experiments. As shown in the Figure 1 (a), we sampled 500 tasks in $\mathbb{R}^2$ parameter space. Specifically, we i.i.d sample and stack data as $X_i \in \mathbb{R}^{K \times 2}, \sim \mathcal{D}(\boldsymbol{x})$ and task optima $\boldsymbol{a}_i \sim \mathcal{D}(\boldsymbol{a}), \boldsymbol{a}_i \in \mathbb{R}^2$ (scattered orange dots) for each task $i$. Green cross shows the location of MAML solution $\boldsymbol{w}_m(\alpha)$ with $\alpha = 0$, namely ERM solution $\boldsymbol{w}_r$. Since $\mathcal{D}(\boldsymbol{x}), \mathcal{D}(\boldsymbol{a})$ are some symmetric zero-mean distributions, the optimal solution is expected at the origin. When several new training tasks (with higher penalties) have been added as shown in the red circle area, then new $\boldsymbol{w}_r$ will be closer to new tasks. Along $\alpha$ increasing, $\boldsymbol{w}_m(\alpha)$ generates a trajectory shown as the purple curve. The dynamics of global minimum $\boldsymbol{w}_m(\alpha)$ will start from green cross and move away from the red circle until reach an optimum location of point (red star $\boldsymbol{w}_m(\alpha^*_{lim})$) which minimizes the total distances to red and orange dots (Other cases shown in Appendix H.4).

It indicates that the effect of $\alpha$ (inner loop optimization) is to help MAML minimize total distances to all training task optima. Unlike ERM learning a biased solution to dense tasks area, MAML converges to a distance-aware solution that tries to minimize the distances within one-step adaptation at stepsize $\alpha$. The $\alpha^*_{lim}$ is the optimum adaptation stepsize to learn the optimum location of point, or nearest point, to all tasks. Figure 1 (b) displays the mean distance for each point in purple trajectory to all tasks. As we can see, the distance decreases at beginning as $\alpha$ increases until reach the minimum.

To theoretically prove the insight in Figure 1, we characterize it by measuring the average post-adaptation distance between the meta-initialization (global minimum) learned by a specific algorithm and task optima in a task distribution.

**Definition 1** (Average Distance under Fast Adaptation). *Given task distribution $\mathcal{D}(T)$, meta-initialization $\boldsymbol{w}_{\mathcal{A}}^0$ learned by algorithm $\mathcal{A}$, optimum $\boldsymbol{a}_T$ of task $T$, the average distance under t-step ($t \geq 0$) fast adaptation is defined by*

$$\mathcal{F}_t(\boldsymbol{w}_{\mathcal{A}}^0) := \mathop{\mathbb{E}}_{T \sim \mathcal{D}(T)} \|\boldsymbol{w}_{\mathcal{A},T}^t - \boldsymbol{a}_T\|^2, \quad \boldsymbol{w}_{\mathcal{A},T}^t = \boldsymbol{w}_{\mathcal{A}}^0 - \eta \sum_{j=1}^t \nabla_{\boldsymbol{w}} \ell(\boldsymbol{w}_{\mathcal{A},T}^j, T) \qquad (3.6)$$

*where $\boldsymbol{w}_{\mathcal{A},T}^t$ is the adapted parameter of task $T$ with $t$ steps, $\eta$ is the step size, $\ell$ is the loss function.*

$\mathcal{F}_t$ evaluates the distance between adapted parameters and true task optimum for a given meta-initialization at any adaptation step $t$. If $t$ is small, $\mathcal{F}_t$ describes the fast adaptation error in solution distance of the meta-initialization learned by an algorithm. Hence, we can measure the fast adaptability of MAML with $\mathcal{F}_t(\boldsymbol{w}_m)$. Observe that for small $\alpha$, $\boldsymbol{w}_m$ can be linearized as

$$\boldsymbol{w}_m(\alpha) = \boldsymbol{w}_r + \alpha \nabla_\alpha \boldsymbol{w}_m(0) + \mathcal{O}(\alpha^2) \qquad (3.7)$$

In this regime, the effect of MAML is dictated by the $\alpha$ gradient $\nabla_\alpha \boldsymbol{w}_m(0)$, which can be visualized as the tangent of the purple curve at the green cross in Figure 1(b). By comparing $\mathcal{F}_t$ of the meta-initializations $\boldsymbol{w}_r, \boldsymbol{w}_m$ learned by ERM and MAML, we are able to find the connection between $\alpha$ and the fast adaptability in meta-learning, at least in the small alpha regime. For simplicity, we assume that the input data features are uncorrelated, thus the covariance matrix is diagonal.

**Theorem 2.** *Let $\boldsymbol{w}_m(\alpha), \boldsymbol{w}_r$ be the meta-initializations learned from $T_1, ..., T_N$ by MAML and ERM. With $\mathcal{F}_t(\cdot)$, under Assumption 1 & 2, for any $\alpha \in \left[ 0, \frac{-2\lambda_S^4 K + K\sqrt{4\lambda_S^8 + 1.5\widetilde{c}\lambda_I^4(4\lambda_S^6 - \lambda_I^6)/\lambda_S^3}}{\lambda_I^2(4\lambda_S^6 - \lambda_I^6)} \right]$ at number of step $t$, we have*

$$\mathbb{E}_{T_1,...,T_N \sim \mathcal{D}(T)} [\mathcal{F}_t(\boldsymbol{w}_r) - \mathcal{F}_t(\boldsymbol{w}_m(\alpha))] \geq \left(1 - \frac{2\eta}{K}\lambda_S\right)^{2t} \frac{4\alpha d^2 \widetilde{c}}{NK\lambda_S^3} \qquad (3.8)$$

*where $\eta$ is the step size in Definition 1, $\widetilde{c} > 0$ is a constant.*

See proof in Appendix G. This theorem prove our insight at small $\alpha$ that MAML has smaller average solution distance than ERM. As it illustrated in Theorem 2, at any step $t \geq 0$, $\mathcal{F}_t(\boldsymbol{w}_m(\alpha)) \leq \mathcal{F}_t(\boldsymbol{w}_r)$ holds if $\alpha$ is smaller than some constant. This means adapting to different tasks with MAML meta-initialization leads to shorter average solution distance than ERM's at any number of adaptation steps. But the gap will disappear along number of steps $t$ increasing to infinity, which is sensible. Note that even $t = 0$, this inequality still holds true. Therefore the meta-initialization of MAML $\boldsymbol{w}_m$ has shorter expected distance to new task than ERM $\boldsymbol{w}_r$ before adaptation. Theorem 2 has revealed the connection between $\alpha$ and fast adaptability. Even with small $\alpha$, MAML learns a more adaptive solution than ERM which is closer to the new tasks in expectation enabling quick approximation. It benefits from learning a closer meta-initialization for all tasks on average. Thus $\alpha$ plays a role in learning a distance-aware solution. This result is consistent with our observation in the Figure 1.

Compared to ERM algorithms, the fast adaptability of MAML stems from the learned meta-initialization determined by the adaptation learning rate $\alpha$. When facing a multi-task problem, traditional ERM algorithms bias its learned initialization to minimize the averaged risk. However, this strategy fails to take the further adaptation into account, and thus learns a solution far from unknown task optima. On the contrary, MAML learns a distant-aware meta-initialization and converges to the vicinity of all task optima with a limited adaptation budget (Nichol et al., 2018; Rajeswaran et al., 2019), or tends to favor "hard tasks" (Fallah et al., 2020; Collins et al., 2020). Hence, before adaptation, ERM may have lower population risk than MAML. However, after adaptation, the situation will reverse since MAML can adapt to most unknown task optima closer (see Figure 5(a)). This benefit is also illustrated by (Zhou et al., 2020) that the shorter solution distance leads to a better meta-initialization for fast adaptation. We note that "task hardness" may not always be easy to define, especially for non-linear cases (Collins et al., 2020). Here, we instead focus on directly analyzing the geometric distance (Theorem 2), which has substantiated the aforementioned findings in optimization behavior from different angles.

## 4   RELATED WORK

Meta learning learns a general solution based on previous experience which can be quickly adapted to unknown tasks (Finn et al., 2017; Li et al., 2017; Snell et al., 2017; Vinyals et al., 2016; Nichol

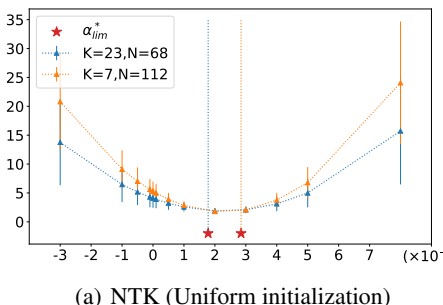 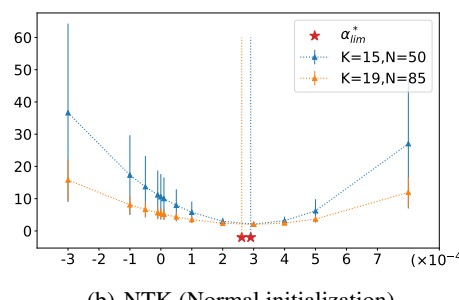

(a) NTK (Uniform initialization)          (b) NTK (Normal initialization)

Figure 2: Loss of overparameterized quadratic regression with regard to $\alpha$. Triangles in the dash-dot line is the mean loss across whole tasks. The error bar denotes $95\%$ confidence interval on different tasks. Red stars are estimations $\alpha^*_{lim}$.

et al., 2018; Grant et al., 2018; Harrison et al., 2018; Rusu et al., 2018; Rajeswaran et al., 2019; Finn & Levine, 2018; Rajeswaran et al., 2019; Finn et al., 2018; Yin et al., 2020). One promising approach to meta-learning is Model-Agnostic Meta-Learning (MAML) (Finn et al., 2017) which learns a meta-initialization such that the model can adapt to a new task via only few steps gradient descent. Understanding the fast adaptability of meta-learning algorithm, especially on MAML, is now an important question. As a variant of MAML, (Nichol et al., 2018) attribute fast adaptation to the shorter solution distance, and devises a first-order approximation algorithm based on this intuition. Other first-order methods (Denevi et al., 2019; Zhou et al., 2019) try to achieve adaptation by adding a regularized term to get a distance-aware solution. (Raghu et al., 2020) shows that even performing inner loop optimization on part of the parameters still leads to fast adaptation. A shared empirical finding of these results is that MAML produces initial weights that are closer to the population optimum of individual tasks on average, and it is argued that this partly contributes to its fast adaptibility. Here, we present a rigorous result that confirms the distance reduction property of MAML, at least in the considered setting, lending theoretical backing to these empirical observations.

On the theoretical front, analyses of meta-learning mainly focus on generalization error bounds and convergence rates (Amit & Meir, 2018; Denevi et al., 2019; Finn et al., 2019; Balcan et al., 2019; Khodak et al., 2019; Zhou et al., 2019; Fallah et al., 2020; Ji et al., 2020b; Zhou et al., 2020; Ji et al., 2020a). For example, Fallah et al. (2020) studies MAML by recasting it as SGD on a modified loss function and bound the convergence rate using the batch size and smoothness of the loss function. Ji et al. (2020b) extend this result to the multi-step version of MAML. Other works (Charles & Konečnỳ, 2020; Wang et al., 2021; Gao & Sener, 2020; Collins et al., 2020) investigate the MAML optimization landscape and the trade-off phenomena in terms of task difficulty: e.g. MAML tend to find meta-initializations that are closer to difficult tasks. However, the effect of inner loop learning rate $\alpha$ on the MAML dynamics and learned solution are not explored in these works.

Of particular relevance is the work of Bernacchia (2021), which derives, under an ideal setting of Gaussian inputs and regression coefficients, a range of $\alpha$ values that can help guide its choice. In this paper, we adopt a more general setting, where we do not assume specific input distributions. We derive a precise optimal value of $\alpha$ (instead of a range), which can be estimated from input data. Furthermore, we show using experiments that the optimal values may not be negative (c.f. Bernacchia (2021)) in the standard meta-learning setting, where the same $\alpha$ is used for training and testing.

## 5   EXPERIMENTS

### 5.1   ESTIMATION OF $\alpha^*$

We verify our theorem through Neural Tangent Kernel (NTK) (Jacot et al., 2018) and deep learning on the Omniglot dataset (Lake et al., 2011). In the former setting, we followed the problem setup in (Bernacchia, 2021) to perform quadratic regression. Different from their model size of 60, we used a two-layer Neural Tangent Kernel (NTK) (Jacot et al., 2018) with sufficiently wide hidden layers (size $10,000$). Then, we can estimate $\alpha^*$ by the neural tangent feature to obtain $\alpha^*_{est} = 1/(2NK\tilde{\sigma}^2)$

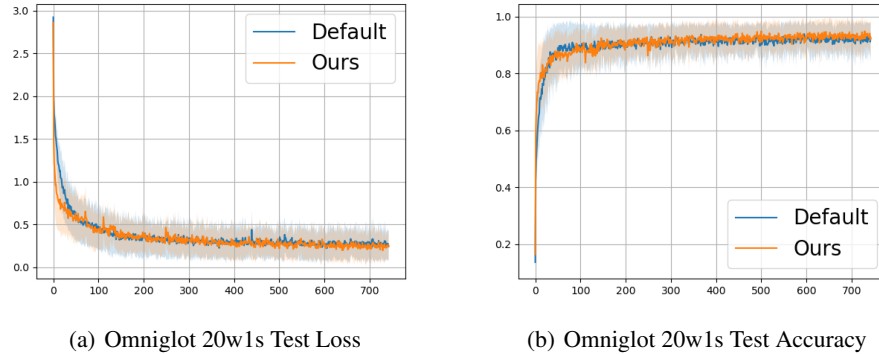

(a) Omniglot 20w1s Test Loss  (b) Omniglot 20w1s Test Accuracy

Figure 3: Test loss and accuracy on Omniglot 20-way 1-shot classification. The blue and orange line represent the test loss (left) and test accuracy (right) of original configuration in ANIL (Raghu et al., 2020) paper and our online estimation. The shadows are the standard deviation of multiple experiments with different seeds.

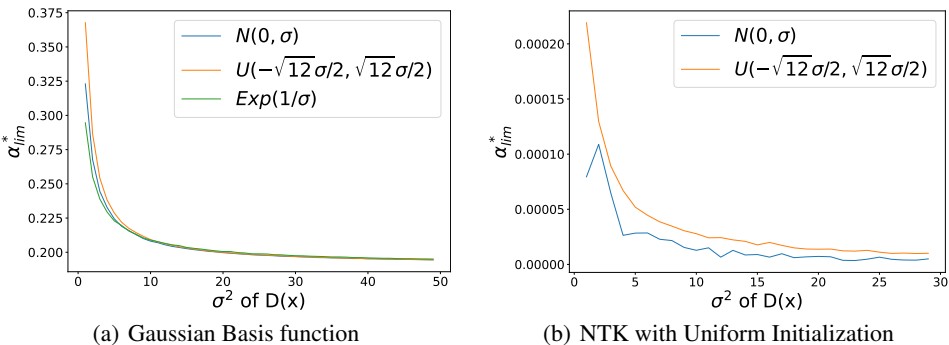

(a) Gaussian Basis function  (b) NTK with Uniform Initialization

Figure 4: Value of $\alpha^*_{lim}$ along the data variance, $\sigma^2$. Different curves are different data distributions. (a)The feature of Gaussian basis function. These curves can be perfectly fitted by an inverse proportional function. (b) The feature of uniformly initialized NTK model.

($\tilde{\sigma}$ is the variance of NTK feature, whose derivation is found in Appendix H.2). Shown as a vertical dotted line ending with the red star in Figure 2, we can see our estimation is nearly optimal. To reduce fortuity, we choose arbitrary values of $N, K$ to compute the estimation $\alpha^*_{lim}$. Furthermore, we also test our estimation on uniform initialization with other groups of hyper-parameters and obtained similar results. Then, for deep learning classification, we use online estimation to compute $\alpha^*$ for ANIL Raghu et al. (2020) on the Omniglot dataset Lake et al. (2011). To keep training stable, we normalize the features before the last layer and compute the corresponding $\alpha^*_{est}$. Then, we compare our estimation scheme with the default selection method where the model and training learning rates are the same. Test loss and accuracy are reported with mean and variance in Figure 3. Both training schemes achieve similar results after $4 \times 10^4$ iterations. We only plot the first $1.5 \times 10^4$ iters (20 iters per scale) to see the differences clearly. As shown, our estimation of $\alpha^*$ converges faster than that in the default configuration. Other experimental parameters and additional results, including non-central distributions and deep regression experiments, are found in Appendix H.4. Overall, these experiments suggest that our estimation derived in the idealized linear setting can guide practical hyper-parameter selection.

## 5.2 RELATION OF DATA VARIANCE AND OPTIMAL $\alpha$

In this section, we verified our theoretical results of $\alpha^*_{lim}$ and its relation to data variance. As drawn the Figure 4, value of $\alpha^*_{lim}$ and data variance have an inverse relationship. We first verified

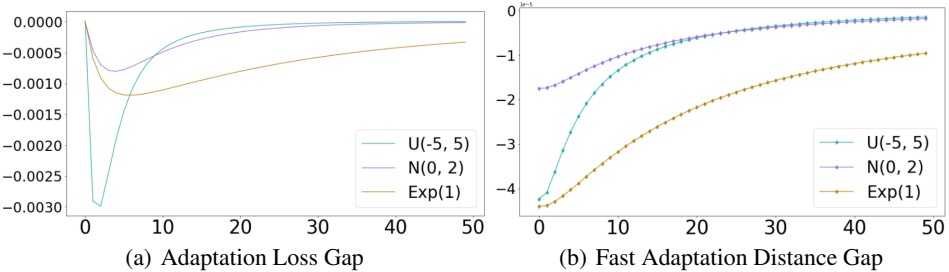

(a) Adaptation Loss Gap           (b) Fast Adaptation Distance Gap

Figure 5: With different data distributions, $x_i \sim \mathcal{U}(-5, 5), \mathcal{N}(0, 2), Exp(1)$ (curves in different colors) (a) the loss difference between MAML and ERM with $t$ steps adaptation on each task $\sum_i [\ell_t^i(\boldsymbol{w}_m) - \ell_t^i(\boldsymbol{w}_r)]$ ( $\ell_t^i(\boldsymbol{w}_m)$ is the $t$-step adaptation loss on task $i$ from the MAML learned initialization $\boldsymbol{w}_m$) and (b) Average solution distance gap of MAML and ERM after $t$-step adaptation, $\mathcal{F}_t(\boldsymbol{w}_m) - \mathcal{F}_t(\boldsymbol{w}_r)$.

this with a Gaussian basis function $\Phi(X)_{(ij)} = \exp(-\left(X_{(ij)} - \mu_j\right)^2 / 2\sigma_i^2)$. Then, we conducted experiments on three different data distributions: normal distribution $N(0, \sigma)$, uniform distribution $U(-\sqrt{12}\sigma/2, \sqrt{12}\sigma/2)$ and exponential distribution $Exp(1/\sigma)$. From in (a), we can see the smooth curves perfectly fitted with some inverse proportional function e.g. $y = 0.35/\sigma^2$. Next, we used NTK as the basis function to verify our result in overparameterized regime. We used two layers MLP with width$= 10, 240$ and uniform initialization to compute the neural tangent feature. As we can see from Figure 4(b), the diagram also shows the tendency that $\alpha_{lim}^*$ decreases as $\sigma$ increasing. As a consequence, variance, as a part of the statistical property of data, will influence $\alpha^*$.

## 5.3 FAST ADAPTATION

To understand the effect of $\alpha$ on $\mathcal{F}_t$, we set $\alpha = 10^{-4}$ to train MAML such that its global minimum $\boldsymbol{w}_m$ is inched from ERM $\boldsymbol{w}_r$. Then, we tracked their adaptation losses and adaptation errors with growing adaptation steps, shown in the Figure 5. Adaptation loss for task $i$ is defined by $\ell_t^i(\boldsymbol{w}) = \|\Phi(X_i)\text{Adapt}(\boldsymbol{w}, i, t, \eta) - \boldsymbol{y}_i\|^2$ where $\text{Adapt}(\boldsymbol{w}, i, t, \eta)$ is $t$-step adaptation parameter with learning rate $\eta = 1e - 5$. The adaptation loss difference between MAML and ERM is described as $\sum_{i=1}^{5000} \ell_t^i(\boldsymbol{w}_m) - \ell_t^i(\boldsymbol{w}_r)$. From Figure 5 (a) we can see, the loss of MAML is marginally higher than ERM before adapting. But the difference dramatically decreases to negative values, which illustrates that MAML has better performance than ERM with only few steps adaptation. Similar results appear on various data distributions: uniform distribution $\mathcal{U}(-5, 5)$, normal distribution $\mathcal{N}(0, 2)$ and exponential distribution $Exp(1)$. It makes sense because $\boldsymbol{w}_r, \boldsymbol{w}_m$ are the minimizers of non-adaptation loss and one-step adaptation loss, respectively. Then we plot the difference of adaptation errors in distance $\mathcal{F}_t(\boldsymbol{w}_m) - \mathcal{F}_t(\boldsymbol{w}_r)$ along adaptation step $t$. In Figure 5(b) we can see, $\mathcal{F}_t$ of MAML is always smaller than ERM's, including $t = 0$. Since $\mathcal{F}_t$ measures distances of adapted solution and task optimum solution, this result has substantiated our Theorem 2. Furthermore, it also demonstrate that the effect of $\alpha$, even it is small, is acting as the guide to find a distance-aware meta-initialization for target tasks which possesses faster adaptability compared to ERM.

## 6 CONCLUSION

In this paper, we investigated MAML through the lens of adaptation learning rate $\alpha$. We gave a principled way to estimate an optimal adaptation learning rate $\alpha^*$ minimizing MAML population risk. We also try to interpret the role of $\alpha$ statistically and geometrically. Further investigation has revealed the underlying data statistics that $\alpha^*$ depends on. This statistical dependency also motivates us to explore other effect of $\alpha$, such as the optimization behavior in a geometric context. By studying the role of $\alpha$ on optimization, we confirmed theoretically that MAML obtains solutions with shorter average distance to individual task optima than ERM - an empirical observation that was suggested to contributes to MAML's fast adaptability. We believe these results are instructive in contributing to the theoretical understanding of meta-learning and its algorithm design.

## 7 ACKNOWLEDGEMENT

This research/project is supported by the National Research Foundation, Singapore under its AI Singapore Programme (AISG Award No: AISG-GC-2019-001-2A). Any opinions, findings and conclusions or recommendations expressed in this material are those of the author(s) and do not reflect the views of National Research Foundation, Singapore. Q. Li is supported by the National Research Foundation, Singapore, under the NRF fellowship (NRF-NRFF13-2021-0005).

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

## A  DEFINITIONS, NOTATIONS AND LEMMAS

**Notation**   We denote an optimal adaptation learning rate as $\alpha^*$. Global minima of empirical risk of MAML and ERM (when they are unique) are denoted by $\boldsymbol{w}_m, \boldsymbol{w}_r$. We write $\{1, ..., N\}$ as $[N]$ and use $\|\cdot\|$ to denote the Euclidean norm. We use subscripts to index the matrices/vectors corresponding to task instances, and bracketed subscripts to index the entries of matrices. For function $f$ depends on $a, b, x$, we omit other variables by $f(..., x)$ when we discuss with $x$.

Table 1: High-frequency notation table.

| Symbols | Definition | Symbols | Definition |
|---|---|---|---|
| $\alpha$ | Adaptation learning rate | $K, k_1, k_2$ | All/train/val/ sample size per task |
| $\alpha^*$ | Optimal adaptation learning rate | $\mathcal{L}_m, \mathcal{L}_r$ | Population risk of MAML/ERM |
| $\alpha^*_{lim}, \alpha^*_{est}$ | Estimation (limit) of $\alpha^*$ | $\hat{\mathcal{L}}_m, \hat{\mathcal{L}}_r$ | Empirical risk of MAML/ERM |
| $\lambda_I, \lambda_S$ | Min/max of eigenvalues | $\bar{\mathcal{L}}_m$ | Average population risk |
| $\boldsymbol{a}_i$ | Task optimum of task $i$ | $N$ | Number of training tasks |
| $d$ | Feature dimension | $\boldsymbol{w}_m, \boldsymbol{w}_r$ | Global minimum of MAML/ERM |

**Definition 2** (Gamma Convergence). *Let $F_n : \mathcal{X} \to \mathbb{R}$ for each $n \in \mathbb{N}$. We say that $(F_n)_{n\in\mathbb{N}}$ $\Gamma$-converges to $F : \mathcal{X} \to \mathbb{R}$, and write $\Gamma - \lim_{n\to\infty} F_n = F$ or $F_n \xrightarrow{\Gamma} F$, if*

- *For every $x \in \mathcal{X}$ and every $(x_n)_{n\in\mathbb{N}}$ such that $x_n \to x$ in $\mathcal{X}$*

$$F(x) \leq \liminf_{n\to\infty} F_n(x_n)$$

- *for every $x \in \mathcal{X}$, there exists some $(x_n)_{n\in\mathbb{N}}$ such that $x_n \to x$ in $\mathcal{X}$ and*

$$F(x) \geq \limsup_{n\to\infty} F_n(x_n)$$

**Definition 3** (Lower Semicontinuous Envelope). *Given $F : \mathcal{X} \to \overline{\mathbb{R}}$, the lower semicontinuous envelope (or relaxation) of $F$ is the "greatest lsc function bounded above by $F$":*

$$F^{\mathrm{lsc}}(x) := \sup\{G(x) \mid G : \mathcal{X} \to \overline{\mathbb{R}} \text{ is lsc and } G \leq F \text{ on } \mathcal{X}\}$$
$$= \inf\left\{\liminf_{n\to\infty} F(x_n) \mid (x_n)_{n\in\mathbb{N}} \subseteq \mathcal{X} \text{ and } x_n \to x\right\}$$

**Lemma 1** (Remark 2.2, (Braides, 2006)). *If $F_n$ uniform converge to $F$, then $F_n \xrightarrow{\Gamma} F^{lsc}$ where $F^{lsc}$ is* Lower Semicontinuous Envelope *of F.*

**Lemma 2** ($\Gamma$-Convergence, (Braides, 2006)). *Let $X$ be a topological space. Let $\{F_n\}$ be a equi-coercive family of functions and let $F_n$ $\Gamma$-converges to $F$ in $X$, then*

- *$\lim_{n\to\infty} d_n = d$ where $d_n = \inf_{x\in X}$ and $d = \inf_{x\in X} F(x)$. That is, the minima converges $F_n(x)$*

- *The minimizers of $F_n$ converge to a minimizer of $F$.*

**Proposition 2.** *If both $A$ and $B$ are positive semidefinite, the inequality is true:*

$$\mathrm{tr}(AB) \leq \mathrm{tr}(A)\,\mathrm{tr}(B). \tag{A.1}$$

*and if $A$ is $n$-by-$n$ symmetric PSD, we have*

$$\mathrm{tr}(A^2) \geq \frac{\mathrm{tr}(A)^2}{n}. \tag{A.2}$$

*Proof.* Let $a = tr(A)$. For PSD matrix $A$, we have $A \preceq aI$. Then

$$tr(AB) = tr(B^{1/2}AB^{1/2}) \leq tr(B^{1/2}(aI)B^{1/2}) = tr(A) tr(B). \tag{A.3}$$

For second inequality, we can apply spectral decomposition on $A$ as $A = QDQ^{-1}$. So we have

$$tr(A) = tr(QDQ^{-1}) = tr(DQQ^{-1}) = tr(D) = \sum_i \lambda_i.$$

where $\{\lambda_i\}, i \in [1, n]$ is the eigenvalues of matrix $A$. Then by Cauchy-Schwarz inequality we can get

$$tr(A)^2 = \left( \sum_{i=1}^n \lambda_i \right)^2 \leq n \left( \sum_{i=1}^n \lambda_i^2 \right) = n \, tr(A^2)$$

$\square$

**Lemma 3** (Hanson-Wright inequality, (Rudelson & Vershynin, 2013))**.** *Let $X = (X_1, \ldots, X_n) \in \mathbb{R}^n$ be a random vector with independent components $X_i$ which satisfy $\mathbb{E}X_i = 0$ and $\|X_i\|_{\psi_2} \leq L$. Let $A$ be an $n \times n$ matrix. Then, for every $t \geq 0$,*

$$\mathbb{P}\left\{ \left| X^\top AX - \mathbb{E}X^\top AX \right| > t \right\} \leq 2 \exp\left[ -c \min\left( \frac{t^2}{L^4\|A\|_{\text{HS}}^2}, \frac{t}{L^2\|A\|} \right) \right]$$

*where $\|\xi\|_{\psi_2} = \sup_{p \geq 1} p^{-1/2} \left( \mathbb{E}|X|^p \right)^{1/p}$ is sub-gaussian norm, $\|A\| = \max_{x \neq 0} \|Ax\|_2/\|x\|_2$ is operator norm and $\|A\|_{HS} = (\sum_{i,j} |a_{i,j}|^2)^{1/2}$ is Hilbert-Schmidt (or Frobenius) norm.*

**Definition 4** (Fast Adaptation Error)**.** *Given the task distribution $\mathcal{D}(T)$, meta-initialization $\boldsymbol{w}^0$, the optimal solution of task $T$ is $\boldsymbol{w}_T^*$, then $t$-step fast adaptation error is defined by*

$$\mathcal{F}_t(\boldsymbol{w}^0, \mathcal{D}(T)) := \mathop{\mathbb{E}}_{T \sim \mathcal{D}(T)} \|\boldsymbol{w}_T^t - \boldsymbol{w}_T^*\|_2^2 \tag{A.4}$$

*where $\boldsymbol{w}_T^t = \boldsymbol{w}^0 - \eta \sum_j^t \nabla_{\boldsymbol{w}_i^j} \ell_i(\boldsymbol{w}_T^j)$ is the adapted parameter of task $T$ with $t$ steps.*

**Lemma 4** (Ruhe's trace inequality, (Marshall et al., 1979))**.** *If $A, B$ are $n \times n$ positive semidefinite Hermitian matrices with eigenvalues,*

$$a_1 \geq \cdots \geq a_n \geq 0, \quad b_1 \geq \cdots \geq b_n \geq 0$$

*respectively, then*

$$\sum_{i=1}^n a_i b_{n-i+1} \leq tr(AB) \leq \sum_{i=1}^n a_i b_i$$

**Proposition 3.** *For any positive random variable $x \sim \mathcal{D}(x)$, we have following inequality holds true*

$$\left( \mathbb{E}_{x \sim \mathcal{D}(x)}(x^2) \right)^2 - \mathbb{E}_{x \sim \mathcal{D}(x)}(x)\mathbb{E}_{x \sim \mathcal{D}(x)}(x^3) \leq 0 \tag{A.5}$$

*Proof.* With the fact that

$$\begin{aligned}
\left( \mathbb{E}_{x \sim \mathcal{D}(x)}(x^2) \right)^2 &- \mathbb{E}_{x \sim \mathcal{D}(x)}(x)\mathbb{E}_{x \sim \mathcal{D}(x)}(x^3) \\
&= \left( \int_R x^2 p(x) \, \mathrm{d}x \right)^2 - \left( \int_R x p(x) \, \mathrm{d}x \right) \left( \int_R x^3 p(x) \, \mathrm{d}x \right)
\end{aligned} \tag{A.6}$$

where $x > 0$ and $p(x) > 0$.

Let $f = (xp(x))^{\frac{1}{2}} > 0$ and $g = (x^3 p(x))^{\frac{1}{2}} > 0$, with Cauchy-Schwarz Inequality,

$$\left( \int_R fg \, \mathrm{d}x \right)^2 \leq \left( \int_R f^2 \, \mathrm{d}x \right) \left( \int_R g^2 \, \mathrm{d}x \right) \tag{A.7}$$

we have that

$$\begin{aligned}
\left( \mathbb{E}_{x \sim \mathcal{D}(x)}(x^2) \right)^2 &= \left( \int_R \sqrt{xp(x)}\sqrt{x^3 p(x)} \, \mathrm{d}x \right)^2 = \left( \int_R fg \, \mathrm{d}x \right)^2 \\
&\leq \int_R \left( \sqrt{xp(x)} \right)^2 \, \mathrm{d}x \int_R \left( \sqrt{x^3 p(x)} \right)^2 \, \mathrm{d}x \\
&= \mathbb{E}_{x \sim \mathcal{D}(x)}(x)\mathbb{E}_{x \sim \mathcal{D}(x)}(x^3)
\end{aligned} \tag{A.8}$$

$\square$

## B    PROOF OF THEOREM 1

**Proof Sketch**    We list our proof steps as follows

1. Get global minima of ERM and MAML by first order optimality condition.
2. Let average MAML population risk $\bar{\mathcal{L}}_m$ as the target in order to eliminate the randomness (Proposition 1 guarantees the upper bound between $\bar{\mathcal{L}}_m$ and population risk $\mathcal{L}_m$).
3. Approximate this target function $\bar{\mathcal{L}}_m$ by another function $L_m^{apx}$ which is the limit of $\bar{\mathcal{L}}_m$ as number of tasks $N \to \infty$.
4. According to positive definiteness, we can get the range of $\alpha$.
5. With notion of gamma convergence, the minimizer of $\bar{\mathcal{L}}_m$ will also converge to the minimizer of $L_m^{apx}$.
6. Our estimation of $\alpha^*$ is in the range of $\alpha$ in Step 4.

**Notation for this proof:** For simplicity, we omit the arguments of the function if its symbol has a index e.g. $\Phi_i = \Phi(X_i), C_i = C_i(\alpha)$. Then we give the full proof on estimation of $\alpha^*$.

*Proof.*    Recall that the global minimum closed form for ERM and MAML are

$$\boldsymbol{w}_r(N,K) = \boldsymbol{w}_r\left(\{\Phi(X_i), \boldsymbol{a}_i\}_{i\in[N]}\right) = \left(\sum_{i=1}^{N} \Phi_i^\top \Phi_i\right)^{-1}\left(\sum_{i=1}^{N} \Phi_j^\top \Phi_j \boldsymbol{a}_j\right)$$

$$\boldsymbol{w}_m(\alpha,N,K) = \boldsymbol{w}_m\left(\{C_i(\alpha), \boldsymbol{a}_i\}_{i\in[N]}\right) = \left(\sum_{i=1}^{N} C_i^\top C_i\right)^{-1}\left(\sum_{j=1}^{N} C_j^\top C_j \boldsymbol{a}_j\right) \tag{B.1}$$

where $C_i = \Phi_i - \frac{2\alpha}{K}\Phi_i\Phi_i^\top\Phi_i, C_i \in \mathbb{R}^{K\times d}$.

Since $\boldsymbol{w}_m$ depends on random variables $\boldsymbol{a}_1, ..., \boldsymbol{a}_N$. The average population risk of MAML, defined in (3.2), where

$$\bar{\mathcal{L}}_m(\alpha,N,K) = \mathbb{E}_{\boldsymbol{w}_m}\mathcal{L}_m(\boldsymbol{w}_m,\alpha,K) = \mathbb{E}_{\boldsymbol{a}_1,...,\boldsymbol{a}_N \sim \mathcal{D}(\boldsymbol{a})}\mathcal{L}_m(\boldsymbol{w}_m,\alpha,K) \tag{B.2}$$

of global minimum $\boldsymbol{w}_m$ in (B.1) can be written as

$$\bar{\mathcal{L}}_m(\alpha,N,K) = \mathbb{E}_{\boldsymbol{a}_1,...,\boldsymbol{a}_N \sim \mathcal{D}(\boldsymbol{a})}\mathcal{L}_m(\boldsymbol{w}_m,\alpha,K)$$

$$= \frac{1}{K}\mathop{\mathbb{E}}_{\{\boldsymbol{a}_i\}_{i=1}^N \sim \mathcal{D}(\boldsymbol{a})}\mathop{\mathbb{E}}_{\boldsymbol{a} \sim \mathcal{D}(\boldsymbol{a})}\mathop{\mathbb{E}}_{X \sim \mathcal{D}(\boldsymbol{x})}\left\|C(\alpha)(\boldsymbol{w}_m - \boldsymbol{a})\right\|^2$$

$$= \frac{1}{K}\mathop{\mathbb{E}}_{\boldsymbol{a},\{\boldsymbol{a}_i\}_{i=1}^N \sim \mathcal{D}(\boldsymbol{a})}\mathop{\mathbb{E}}_{X \sim \mathcal{D}(\boldsymbol{x})}\left\|C(\alpha)\left[\left(\sum_{i=1}^{N} C_i^\top C_i\right)^{-1}\left(\sum_{j=1}^{N} C_j^\top C_j \boldsymbol{a}_j\right) - \boldsymbol{a}\right]\right\|^2 \tag{B.3}$$

Let $\Lambda_j = \left(\sum_{i=1}^{N} C_i^\top C_i\right)^{-1} C_j^\top C_j, j \in [1,N], \Lambda_j \in \mathbb{R}^{d\times d}$. The (B.3) can be rewritten as,

$$\bar{\mathcal{L}}_m(\alpha,N,K) = \frac{1}{K}\mathop{\mathbb{E}}_{\boldsymbol{a},\{\boldsymbol{a}_i\}_{i=1}^N \sim \mathcal{D}(\boldsymbol{a})}\mathop{\mathbb{E}}_{X \sim \mathcal{D}(\boldsymbol{x})}\left\|C(\alpha)\left(\sum_{j=1}^{N}\Lambda_j\boldsymbol{a}_j - \boldsymbol{a}\right)\right\|^2$$

$$= \frac{1}{K}\mathop{\mathbb{E}}_{\boldsymbol{a},\{\boldsymbol{a}_i\}_{i=1}^N \sim \mathcal{D}(\boldsymbol{a})}\mathop{\mathbb{E}}_{X \sim \mathcal{D}(\boldsymbol{x})}\left[\left(\sum_{i=1}^{N}\Lambda_i\boldsymbol{a}_i\right)^\top C(\alpha)^\top C(\alpha)\left(\sum_{j=1}^{N}\Lambda_j\boldsymbol{a}_j\right)\right.$$

$$- \boldsymbol{a}^\top C(\alpha)^\top C(\alpha)\left(\sum_{j=1}^{N}\Lambda_j\boldsymbol{a}_j\right) - \left(\sum_{i=1}^{N}\Lambda_i\boldsymbol{a}_i\right)^\top C(\alpha)^\top C(\alpha)\boldsymbol{a}$$

$$\left.+ \boldsymbol{a}^\top C(\alpha)^\top C(\alpha)\boldsymbol{a}\right] \tag{B.4}$$

Under Assumption 1 and $\boldsymbol{a}$ is independent to $X$, then we have

$$
\begin{aligned}
\bar{\mathcal{L}}_m(\alpha, N, K) =& \frac{1}{K} \underset{X \sim \mathcal{D}(\boldsymbol{x})}{\mathbb{E}} \left[ \underset{\{\boldsymbol{a}_i\}_{i=1}^N \sim \mathcal{D}(\boldsymbol{a})}{\mathbb{E}} \left( \sum_{i=1}^N \Lambda_i \boldsymbol{a}_i \right)^\top C(\alpha)^\top C(\alpha) \left( \sum_{j=1}^N \Lambda_j \boldsymbol{a}_j \right) \right. \\
& \left. + \sigma_a^2 \operatorname{tr} \left( C(\alpha)^\top C(\alpha) \right) \right] \\
=& \frac{\sigma_a^2}{K} \underset{X \sim \mathcal{D}(\boldsymbol{x})}{\mathbb{E}} \left[ \left( \sum_{j=1}^N \operatorname{tr} \left( \Lambda_j^\top C(\alpha)^\top C(\alpha) \Lambda_j \right) \right) + \operatorname{tr} \left( C(\alpha)^\top C(\alpha) \right) \right]
\end{aligned}
$$
(B.5)

Let $L_m^{apx}(\alpha)$ be an approximation function of $\bar{\mathcal{L}}_m(\alpha, N, K)$.

$$
\begin{aligned}
L_m^{apx}(\alpha) \triangleq & \frac{\sigma_a^2}{K} \underset{X \sim \mathcal{D}(\boldsymbol{x})}{\mathbb{E}} \operatorname{tr}[C(\alpha)^\top C(\alpha)] \\
=& \frac{\sigma_a^2}{K} \underset{X \sim \mathcal{D}(\boldsymbol{x})}{\mathbb{E}} \operatorname{tr} \left[ \left( I - \frac{2\alpha}{K} \Phi(X)^\top \Phi(X) \right)^\top \Phi(X)^\top \Phi(X) \left( I - \frac{2\alpha}{K} \Phi(X)^\top \Phi(X) \right) \right]
\end{aligned}
$$
(B.6)

Then the approximation error will be

$$
\begin{aligned}
& \frac{K}{\sigma_a^2} |\bar{\mathcal{L}}_m(\alpha, N, K) - L_m^{apx}(\alpha)| \\
=& \left| \underset{X \sim \mathcal{D}(\boldsymbol{x})}{\mathbb{E}} \left( \sum_{j=1}^N \operatorname{tr} \left( \Lambda_j^\top C(\alpha)^\top C(\alpha) \Lambda_j \right) \right) \right| \\
=& \left| \underset{X \sim \mathcal{D}(\boldsymbol{x})}{\mathbb{E}} \sum_{j=1}^N \operatorname{tr} \left[ C_j^\top C_j \left( \sum_{i=1}^N C_i^\top C_i \right)^{-1} C(\alpha)^\top C(\alpha) \left( \sum_{i=1}^N C_i^\top C_i \right)^{-1} C_j^\top C_j \right] \right|
\end{aligned}
$$
(B.7)

where

$$
C_i^\top C_i = \Phi_i^\top \Phi_i - \frac{4\alpha}{K} (\Phi_i^\top \Phi_i)^2 + \frac{4\alpha^2}{K^2} + (\Phi_i^\top \Phi_i)^3
$$
(B.8)

With Assumption 2, there exists constants $0 < c_1 < c_2$ where

$$
c_1 \le \|\Phi(X_i)\|_F^2 \le c_2, \forall i \in [N]
$$
(B.9)

With Proposition 2 hold, $\forall i \in [N]$ we have

$$
\begin{aligned}
\operatorname{tr}(C_i^\top C_i) =& \operatorname{tr} \left( \Phi_i^\top \Phi_i - \frac{4\alpha}{K} (\Phi_i^\top \Phi_i)^2 + \frac{4\alpha^2}{K^2} (\Phi_i^\top \Phi_i)^3 \right) \\
=& \operatorname{tr} \left( \Phi_i^\top \Phi_i \right) - \operatorname{tr} \left( \frac{4\alpha}{K} (\Phi_i^\top \Phi_i)^2 \right) + \operatorname{tr} \left( \frac{4\alpha^2}{K^2} (\Phi_i^\top \Phi_i)^3 \right) \\
\le& \sup_{i \in [N]} \|\Phi_i\|_F^2 - \frac{4\alpha}{K} \inf_{i \in [N]} \operatorname{tr}[(\Phi_i^\top \Phi_i)^2] + \frac{4\alpha^2}{K^2} \sup_{i \in [N]} \|\Phi_i\|_F^6 \\
=& c_2 - \frac{4\alpha}{K} \inf_{i \in [N]} \operatorname{tr}[(\Phi_i^\top \Phi_i)^2] + \frac{4\alpha^2}{K^2} c_2^3 \\
\le& c_2 - \frac{4\alpha}{Kd} c_1^2 + \frac{4\alpha^2}{K^2} c_2^3
\end{aligned}
$$
(B.10)

Then by applying multiple times of Proposition 2 we have

$$\mathop{\mathbb{E}}_{X \sim \mathcal{D}(\boldsymbol{x})} \sum_{j=1}^{N} \operatorname{tr} \left[ C_j^\top C_j \left( \sum_{i=1}^{N} C_i^\top C_i \right)^{-1} C(\alpha)^\top C(\alpha) \left( \sum_{i'=1}^{N} C_{i'}^\top C_{i'} \right)^{-1} C_j^\top C_j \right]$$

$$= \mathop{\mathbb{E}}_{X \sim \mathcal{D}(\boldsymbol{x})} \sum_{j=1}^{N} \operatorname{tr} \left[ C_j^\top C_j C_j^\top C_j \left( \sum_{i=1}^{N} C_i^\top C_i \right)^{-1} \left( \sum_{i'=1}^{N} C_{i'}^\top C_{i'} \right)^{-1} C(\alpha)^\top C(\alpha) \right]$$

$$\leq \mathop{\mathbb{E}}_{X \sim \mathcal{D}(\boldsymbol{x})} \sum_{j=1}^{N} \operatorname{tr} \left[ (C_j^\top C_j)^2 \right] \operatorname{tr} \left[ \left( \sum_{i=1}^{N} C_i^\top C_i \right)^{-2} \right] \operatorname{tr} \left( C(\alpha)^\top C(\alpha) \right) \tag{B.11}$$

$$\leq \mathop{\mathbb{E}}_{X \sim \mathcal{D}(\boldsymbol{x})} \sum_{j=1}^{N} \operatorname{tr} \left( C_j^\top C_j \right)^2 \operatorname{tr} \left[ \left( \sum_{i=1}^{N} C_i^\top C_i \right)^{-1} \right]^2 \operatorname{tr} \left( C(\alpha)^\top C(\alpha) \right)$$

$$\leq N \left( c_2 - \frac{4\alpha}{Kd} c_1^2 + \frac{4\alpha^2}{K^2} c_2^3 \right)^3 \mathop{\mathbb{E}}_{X \sim \mathcal{D}(\boldsymbol{x})} \operatorname{tr} \left[ \left( \sum_{i=1}^{N} C_i^\top C_i \right)^{-1} \right]^2$$

Next, we need upper bound the last inverse term. With Assumption 2 we know that all eigenvalues of $\Phi(X)^\top \Phi(X), X \sim \mathcal{D}(\boldsymbol{x})$ are bounded by $[\lambda_I, \lambda_S]$. Let $C_{all}(\alpha) = \sum_{i=1}^{N} C_i^\top C_i$, then with probability 1 the max/min eigenvalues of $C_{all}(\alpha)$ will have following constraints,

$$\begin{cases} \lambda_{min}(C_{all}(\alpha)) & \geq N(\lambda_I - 4\alpha\lambda_S^2/K + 4\alpha^2\lambda_I^3/K^2) \\ \lambda_{max}(C_{all}(\alpha)) & \leq N(\lambda_S - 4\alpha\lambda_I^2/K + 4\alpha^2\lambda_S^3/K^2) \end{cases} \tag{B.12}$$

Since the $C_{all}(\alpha)$ is a positive matrix, we need have the constrain on $\lambda_{min}(C_{all}(\alpha)) > 0$, which means

$$\alpha \in \left[ 0, \frac{K(\lambda_S^2 - \sqrt{\lambda_S^4 - \lambda_I^4})}{2\lambda_I^3} \right) \cup \left( \frac{K(\lambda_S^2 + \sqrt{\lambda_S^4 - \lambda_I^4})}{2\lambda_I^3}, \infty \right) \tag{B.13}$$

There exists a positive definite matrix $\lambda_s(\alpha, N)I$ where

$$C_{all}(\alpha) \succeq \lambda_{min}(C_{all})I \tag{B.14}$$

and the following inequality is easy to get

$$\begin{aligned} \operatorname{tr}(C_{all}(\alpha)) &\geq \operatorname{tr}(\lambda_{min}(C_{all})I) \\ \operatorname{tr}(C_{all}^{-1}(\alpha)) &\leq \operatorname{tr}(\lambda_{min}^{-1}(C_{all})I) \end{aligned} \tag{B.15}$$

So the last inverse term will be

$$\mathop{\mathbb{E}}_{X \sim \mathcal{D}(\boldsymbol{x})} \operatorname{tr} \left[ \left( \sum_{i=1}^{N} C_i^\top C_i \right)^{-1} \right]^2 \leq \left( \frac{1}{N(\lambda_I - 4\alpha\lambda_S^2/K + 4\alpha^2\lambda_I^3/K^2)d} \right)^2 = O\left( \frac{1}{N^2} \right) \tag{B.16}$$

Apply these inequalities to (B.7), we can get the upper bound

$$\left| \bar{\mathcal{L}}_m(\alpha, N, K) - L_m^{apx}(\alpha) \right| \leq \frac{\sigma_a^2}{K} \cdot \frac{N \left( c_1 - \frac{4\alpha}{Kd} c_2^2 + \frac{4\alpha^2}{K^2} c_1^3 \right)^3}{N^2(\lambda_I - 4\alpha\lambda_S^2/K + 4\alpha^2\lambda_I^3/K^2)^2 d^2} = O\left( \frac{1}{N} \right) \tag{B.17}$$

When $\alpha \in \left[ 0, \frac{K(\lambda_S^2 - \sqrt{\lambda_S^4 - \lambda_I^4})}{2\lambda_I^3} \right) \cup \left( \frac{K(\lambda_S^2 + \sqrt{\lambda_S^4 - \lambda_I^4})}{2\lambda_I^3}, \infty \right)$, the limit will go to zero,

$$\lim_{N \to \infty} \sup_{\alpha \in \left[ 0, \frac{K(\lambda_S^2 - \sqrt{\lambda_S^4 - \lambda_I^4})}{2\lambda_I^3} \right) \cup \left( \frac{K(\lambda_S^2 + \sqrt{\lambda_S^4 - \lambda_I^4})}{2\lambda_I^3}, \infty \right)} \left| \bar{\mathcal{L}}_m(\alpha, N, K) - L_m^{apx}(\alpha) \right| = 0 \tag{B.18}$$

which means $\bar{\mathcal{L}}_m(\alpha, N, K)$ will uniformly converge to $L_m^{apx}(\alpha)$ for $\alpha$ belongs to the interval above. Note that $\bar{\mathcal{L}}_m(\alpha, N, K)$ is a continuous function of $\alpha$. So with Lemma 1, we have

$$\Gamma - \lim_{N \to \infty} \bar{\mathcal{L}}_m(\alpha, N, K) = L_m^{apx}(\alpha) \tag{B.19}$$

So we have estimation of true $\alpha^*$, denoted as $\alpha_{lim}^*$

$$
\begin{aligned}
\alpha_{lim}^* &= \arg\min_\alpha L_m^{apx}(\alpha) \\
&= \arg\min_\alpha \mathbb{E}_{X \sim \mathcal{D}(X)} \operatorname{tr}\left[\Phi(X)^\top\Phi(X) - \frac{4\alpha}{K}(\Phi(X)^\top\Phi(X))^2 + \frac{4\alpha^2}{K^2}(\Phi(X)^\top\Phi(X))^3\right] \\
&= \frac{K \operatorname{tr}[\mathbb{E}_X[(\Phi(X)^\top\Phi(X))^2]]}{2 \operatorname{tr}[\mathbb{E}_X[(\Phi(X)^\top\Phi(X))^3]]}
\end{aligned}
$$

$$\tag{B.20}$$

According to Lemma 2 where $\alpha^*(N, K)$ is the minimizer of $\bar{\mathcal{L}}_m(\alpha, N, K)$

$$\alpha^*(N, K) = \arg\min_\alpha \bar{\mathcal{L}}_m(\alpha, N, K), \quad \lim_{N \to \infty} \alpha^*(N, K) = \alpha_{lim}^*. \tag{B.21}$$

$\square$

## C  PROOF OF PROPOSITION 1

**Proposition 4** (Formal state of Proposition 1). *Assume $\boldsymbol{u} = C(\alpha)^\top C(\alpha)\boldsymbol{a}$ is sub-gaussian random variable with sub-gaussian norm $\|\boldsymbol{u}_{(i)}\|_{\Psi_2} = \sup_{p \geq 1} p^{-1/2}\left(\mathbb{E}|\boldsymbol{u}_{(i)}|^p\right)^{1/p} \leq L$. Then with probability at least $1 - \delta$ that*

$$\left|\mathcal{L}_m(\boldsymbol{w}_m, \alpha, K) - \bar{\mathcal{L}}_m(\alpha, N, K)\right| \leq \frac{L^2}{K} \max\left\{\sqrt{\frac{d\varepsilon(\alpha, K)}{N^2} \log\frac{2}{\delta}}, \frac{\varepsilon(\alpha, K)}{N}\log\frac{2}{\delta}\right\} \tag{C.1}$$

*if $\alpha \in \left[0, \frac{K(\lambda_S^2 - \sqrt{\lambda_S^4 - \lambda_I^4})}{2\lambda_I^3}\right) \cup \left(\frac{K(\lambda_S^2 + \sqrt{\lambda_S^4 - \lambda_I^4})}{2\lambda_I^3}, \infty\right)$ and*

$$\varepsilon(\alpha, K) = (\lambda_S - 4\alpha\lambda_I^2/K + 4\alpha^2\lambda_S^3/K^2)/(\lambda_I - 4\alpha\lambda_S^2/K + 4\alpha^2\lambda_I^3/K^2)^2$$

*when assumption 1 & 2 holds. $d$ is the feature size, $N$ is the number of tasks and $K$ is the sample size per task.*

Here, we follow the same proof notation as Theorem 1.

*Proof.* By definition, we have

$$\bar{\mathcal{L}}_m(\alpha, N, K) = \mathbb{E}_{\boldsymbol{a}_1, \dots, \boldsymbol{a}_N \sim \mathcal{D}(\boldsymbol{a})} \mathcal{L}_m(\boldsymbol{w}_m(\alpha, N, K), \alpha, K) \tag{C.2}$$

Similar to (B.3), let $\Lambda_j = \left(\sum_{i=1}^N C_i^\top C_i\right)^{-1} C_j^\top C_j$, we have,

$$
\begin{aligned}
\mathcal{L}_m(\boldsymbol{w}_m, \alpha, K) &= \frac{1}{K} \mathbb{E}_{\boldsymbol{a} \sim \mathcal{D}(\boldsymbol{a})} \mathbb{E}_{X \sim \mathcal{D}(\boldsymbol{x})} \|C(\alpha)(\boldsymbol{w}_m(\alpha, N, K) - \boldsymbol{a})\|^2 \\
&= \frac{1}{K} \mathbb{E}_{\boldsymbol{a} \sim \mathcal{D}(\boldsymbol{a})} \mathbb{E}_{X \sim \mathcal{D}(\boldsymbol{x})} \left\|C(\alpha)\left(\sum_{j=1}^N \Lambda_j \boldsymbol{a}_j - \boldsymbol{a}\right)\right\|^2 \\
&= \frac{1}{K} \mathbb{E}_{\boldsymbol{a} \sim \mathcal{D}(\boldsymbol{a})} \mathbb{E}_{X \sim \mathcal{D}(\boldsymbol{x})} \left[\left(\sum_{i=1}^N \Lambda_i \boldsymbol{a}_i\right)^\top C(\alpha)^\top C(\alpha) \left(\sum_{j=1}^N \Lambda_j \boldsymbol{a}_j\right)\right. \\
&\quad \left. -\boldsymbol{a}^\top C(\alpha)^\top C(\alpha)\left(\sum_{j=1}^N \Lambda_j \boldsymbol{a}_j\right) - \left(\sum_{i=1}^N \Lambda_i \boldsymbol{a}_i\right)^\top C(\alpha)^\top C(\alpha)\boldsymbol{a}\right. \\
&\quad \left. +\boldsymbol{a}^\top C(\alpha)^\top C(\alpha)\boldsymbol{a}\right]
\end{aligned}
$$

$$\tag{C.3}$$

With Assumption 1, second term and third term of (C.3) will be cancelled. So the $\mathcal{L}_m$ is

$$\mathcal{L}_m = \frac{1}{K} \underset{X \sim \mathcal{D}(\boldsymbol{x})}{\mathbb{E}} \left[ \left( \sum_{i=1}^{N} \Lambda_i \boldsymbol{a}_i \right)^{\top} C(\alpha)^{\top} C(\alpha) \left( \sum_{j=1}^{N} \Lambda_j \boldsymbol{a}_j \right) \right] + \frac{\sigma_a^2}{K} \underset{X \sim \mathcal{D}(\boldsymbol{x})}{\mathbb{E}} \operatorname{tr}[C(\alpha)^{\top} C(\alpha)] \tag{C.4}$$

As the comparison, $\bar{\mathcal{L}}_m(\alpha, N, K)$ is the averaged function of $\mathcal{L}_m$, which is given by

$$\begin{aligned} \bar{\mathcal{L}}_m = &\frac{1}{K} \underset{X \sim \mathcal{D}(\boldsymbol{x})}{\mathbb{E}} \left[ \underset{\{\boldsymbol{a}_i\}_{i=1}^{N} \sim \mathcal{D}(\boldsymbol{a})}{\mathbb{E}} \left( \sum_{i=1}^{N} \Lambda_i \boldsymbol{a}_i \right)^{\top} C(\alpha)^{\top} C(\alpha) \left( \sum_{j=1}^{N} \Lambda_j \boldsymbol{a}_j \right) \right. \\ &\left. + \sigma_a^2 \operatorname{tr} \left( C(\alpha)^{\top} C(\alpha) \right) \right] \end{aligned} \tag{C.5}$$

Let $A$ be the common matrix of cross-term of $\boldsymbol{a}_i$ and $\boldsymbol{a}_j$, then for each term in (C.5),

$$\begin{aligned} \Lambda_i^{\top} C(\alpha)^{\top} C(\alpha) \Lambda_j = &\underbrace{C_i^{\top} C_i}_{\widetilde{C}_i} \underbrace{\left( \sum_{k=1}^{N} C_k^{\top} C_k \right)^{-1} C(\alpha)^{\top} C(\alpha) \left( \sum_{k'=1}^{N} C_{k'}^{\top} C_{k'} \right)^{-1}}_{A} C_j^{\top} C_j \\ = &\widetilde{C}_i A \widetilde{C}_j \end{aligned} \tag{C.6}$$

So cancel their second terms, the difference of $\mathcal{L}_m(\boldsymbol{w}_m, \alpha)$ and $\bar{\mathcal{L}}_m(\alpha, N, K)$ will be

$$\begin{aligned} &\mathcal{L}_m(\boldsymbol{w}_m, \alpha, K) - \bar{\mathcal{L}}_m(\alpha, N, K) \\ &= \frac{1}{K} \underset{X \sim \mathcal{D}(\boldsymbol{x})}{\mathbb{E}} \left[ \sum_{i=1}^{N} \sum_{j=1}^{N} \boldsymbol{a}_i^{\top} \widetilde{C}_i A \widetilde{C}_j \boldsymbol{a}_j \right] + \cancel{\frac{\sigma_a^2}{K} \underset{X \sim \mathcal{D}(\boldsymbol{x})}{\mathbb{E}} \operatorname{tr} \left( C(\alpha)^{\top} C(\alpha) \right)} \\ &\quad - \frac{1}{K} \underset{\{\boldsymbol{a}_i\}_{i=1}^{N} \sim \mathcal{D}(\boldsymbol{a})}{\mathbb{E}} \underset{X \sim \mathcal{D}(\boldsymbol{x})}{\mathbb{E}} \left[ \sum_{i=1}^{N} \sum_{j=1}^{N} \boldsymbol{a}_i^{\top} \widetilde{C}_i A \widetilde{C}_j \boldsymbol{a}_j + \cancel{\sigma_a^2 \operatorname{tr} \left( C(\alpha)^{\top} C(\alpha) \right)} \right] \\ &= \frac{1}{K} \underset{X \sim \mathcal{D}(\boldsymbol{x})}{\mathbb{E}} \left[ \sum_{i=1}^{N} \sum_{j=1}^{N} \boldsymbol{a}_i^{\top} \widetilde{C}_i A \widetilde{C}_j \boldsymbol{a}_j - \underset{\{\boldsymbol{a}_i\}_{i=1}^{N} \sim \mathcal{D}(\boldsymbol{a})}{\mathbb{E}} \sum_{i=1}^{N} \sum_{j=1}^{N} \boldsymbol{a}_i^{\top} \widetilde{C}_i A \widetilde{C}_j \boldsymbol{a}_j \right] \\ &= \frac{1}{K} \underset{X \sim \mathcal{D}(\boldsymbol{x})}{\mathbb{E}} \left[ \sum_{i=1}^{N} \sum_{j=1}^{N} \boldsymbol{u}_i^{\top} A \boldsymbol{u}_j - \underset{\{\boldsymbol{a}_i\}_{i=1}^{N} \sim \mathcal{D}(\boldsymbol{a})}{\mathbb{E}} \sum_{i=1}^{N} \sum_{j=1}^{N} \boldsymbol{u}_i^{\top} A \boldsymbol{u}_j \right] \end{aligned} \tag{C.7}$$

where $\boldsymbol{u}_i = \widetilde{C}_i \boldsymbol{a}_i = C_i^{\top} C_i \boldsymbol{a}_i$ is sub-gaussian random variable and with Assumption 1,

$$\mathbb{E} \boldsymbol{u}_i = \mathbb{E} \boldsymbol{a}_i = \mathbf{0} \tag{C.8}$$

Let $U_{[N]} = (\boldsymbol{u}_1; ...; \boldsymbol{u}_N), \in \mathbb{R}^{Nd}$, we write the quadratic form into a bilinear form for each product term $\boldsymbol{u}_i^{\top} A \boldsymbol{u}_j$

$$U_{[N]}^{\top} \widetilde{A} U_{[N]} = (\boldsymbol{u}_1^{\top}, ..., \boldsymbol{u}_N^{\top}) \begin{bmatrix} A & \cdots & A \\ \vdots & \ddots & \vdots \\ A & \cdots & A \end{bmatrix} \begin{pmatrix} \boldsymbol{u}_1 \\ \vdots \\ \boldsymbol{u}_N \end{pmatrix} = \sum_i \sum_j \boldsymbol{u}_i^{\top} A \boldsymbol{u}_j \tag{C.9}$$

where

$$\widetilde{A} = 1_N 1_N^{\top} \otimes \mathrm{A}, \|\widetilde{A}\| \in \mathbb{R}^{Nd \times Nd} \tag{C.10}$$

is a $N \times N$ block matrix and $\otimes$ is Kronecker product. And the relations of $A$ and $\tilde{A}$ are

$$\|\widetilde{A}\| = N\|A\|, \|\widetilde{A}\|_{HS} = N^2 \sum_{i,j} A_{(i,j)}^2 \tag{C.11}$$

By applying Hanson-Wright inequality we have

$$\Pr\left(\left|\mathcal{L}_m(\boldsymbol{w}_m, \alpha, K) - \bar{\mathcal{L}}_m(\alpha, N, K)\right| > t\right) = \Pr\left(\frac{1}{K}\left|U_{[N]}^\top \widetilde{A} U_{[N]} - \mathbb{E}U_{[N]}^\top \widetilde{A} U_{[N]}\right| > t\right)$$

$$\leq 2\exp\left[-c\min\left(\frac{t^2}{L^4\|\widetilde{A}\|_{\mathrm{HS}}^2}, \frac{t}{L^2\|\widetilde{A}\|}\right)\right] \quad \text{(C.12)}$$

Further with Cauchy Inequality, we can get the following equation

$$\|A\| = \left\|\left(\sum_{k=1}^N C_k^\top C_k\right)^{-1} C(\alpha)^\top C(\alpha)\left(\sum_{k=1}^N C_k^\top C_k\right)^{-1}\right\|$$

$$\leq \left\|\left(\sum_{k=1}^N C_k^\top C_k\right)^{-1}\right\|^2 \|C(\alpha)^\top C(\alpha)\| \quad \text{(C.13)}$$

According to Theorem 1, all eigenvalues of second term falls in $[\lambda_I - 4\alpha\lambda_S^2/K + 4\alpha^2\lambda_I^3/K^2, \lambda_S - 4\alpha\lambda_I^2/K + 4\alpha^2\lambda_S^3/K^2]$ and of order $1/N$ for first term.

$$\|A\| \leq \frac{(\lambda_S - 4\alpha\lambda_I^2/K + 4\alpha^2\lambda_S^3/K^2)}{N^2(\lambda_I - 4\alpha\lambda_S^2/K + 4\alpha^2\lambda_I^3/K^2)^2} \quad \text{(C.14)}$$

Let $\varepsilon(\alpha, K) = (\lambda_S - 4\alpha\lambda_I^2/K + 4\alpha^2\lambda_S^3/K^2)/(\lambda_I - 4\alpha\lambda_S^2/K + 4\alpha^2\lambda_I^3/K^2)^2$

$$\|\widetilde{A}\| = N\|A\| \leq \frac{\varepsilon(\alpha, K)}{N} \quad \text{(C.15)}$$

Next, we can bound $\|A\|_{HS}$ by $\|A\|$. It's obvious that $\|A\|_{HS} \leq \sqrt{\mathrm{rank}(A)}\|A\|$. So $\|A\|_{HS}^2$ can be upper bounded by

$$\|A\|_{HS}^2 \leq \mathrm{rank}(A)\|A\|^2 \leq \mathrm{rank}(A)\left\|\left(\sum_{k=1}^N C_k(\alpha)^\top C_k(\alpha)\right)^{-1}\right\|^4 \|C(\alpha)^\top C(\alpha)\|^2$$

$$\leq \frac{\mathrm{rank}(A)\varepsilon^2(\alpha, K)}{N^4} \leq \frac{d\varepsilon^2(\alpha, K)}{N^4} \quad \text{(C.16)}$$

Thus, the $\|\widetilde{A}\|_{HS}^2$ will no more than

$$\|\widetilde{A}\|_{HS}^2 \leq \varepsilon^2(\alpha, K)\frac{d}{N^2} \quad \text{(C.17)}$$

In summary, we can get the bound

$$\Pr\left(\left|\mathcal{L}_m(\boldsymbol{w}_m, \alpha, K) - \bar{\mathcal{L}}_m(\alpha, N, K)\right| > t\right) \leq 2\exp\left[-c\min\left(\frac{t^2 N^2}{L^4 d\varepsilon^2(\alpha, K)}, \frac{tN}{L^2\varepsilon(\alpha, K)}\right)\right] \quad \text{(C.18)}$$

Finally, we rewrite the inequality by eliminating $t$, we have at least $1 - \delta$,

$$\left|\mathcal{L}_m(\boldsymbol{w}_m, \alpha, K) - \bar{\mathcal{L}}_m(\alpha, N, K)\right| \leq \frac{L^2}{K}\max\left\{\sqrt{\frac{d\varepsilon(\alpha, K)}{N^2}\log\frac{2}{\delta}}, \frac{\varepsilon(\alpha, K)}{N}\log\frac{2}{\delta}\right\} \quad \text{(C.19)}$$

$$\square$$

## D  PROOF OF COROLLARY 1

*Proof.* Recall our estimation of $\alpha^*$ is given by,

$$\alpha_{lim}^* = \arg\min_\alpha L_m^{apx}(\alpha) = \frac{K\,\mathrm{tr}[\mathbb{E}_X[(\Phi(X)^\top \Phi(X))^2]]}{2\,\mathrm{tr}[\mathbb{E}_X[(\Phi(X)^\top \Phi(X))^3]]} \quad \text{(D.1)}$$

For each task, we have $K$ samples with $d$ dimensional features $\Phi(X) \in \mathbb{R}^{K \times d}$. Since $\Phi(X)^\top \Phi(X)$ is positive definite matrix, by applying spectral decomposition, we have

$$
\begin{aligned}
\operatorname{tr} \mathbb{E}_X[(\Phi(X)^\top \Phi(X))^2] &= \mathbb{E}_X \operatorname{tr}[(\Phi(X)^\top \Phi(X))^2] \\
&= \mathbb{E}_X \operatorname{tr}[(U \Sigma_X U^\top)(U \Sigma_X U^\top)] \\
&= \operatorname{tr} \mathbb{E}_X[\Sigma_X^2]
\end{aligned}
\tag{D.2}
$$

where $U$ is an orthogonal matrix and $\Sigma_X$ is the a diagonal matrix filled by eigenvalues $\lambda_1, ..., \lambda_d$ of the covariance matrix of the feature. It's easy to prove in Principle Component Analysis (PCA) that $\operatorname{tr} \mathbb{E}(\Sigma_X)$ is the variance of features where

$$
\begin{aligned}
\sigma^2(\phi(\boldsymbol{x}_1), \dots, \phi(\boldsymbol{x}_K)) &= \frac{1}{K} \operatorname{tr} \mathbb{E}_X[(\Phi(X) - \mu)^\top (\Phi(X) - \mu)] \\
&= \frac{1}{K} \sum_{i=1}^K (\phi(\boldsymbol{x}_i) - \mu)^2 = \frac{1}{K} \sum_{i=1}^d \lambda_i \\
&= \frac{1}{K} \operatorname{tr} \mathbb{E}(\Sigma_X)
\end{aligned}
\tag{D.3}
$$

where $\phi(\boldsymbol{x}_i) \in \mathbb{R}^d$ is each row of $\Phi(X)$ and $\mu$ is zero.

With Jensen's inequality, we have

$$
\frac{1}{d}[\operatorname{tr} \mathbb{E}(\Sigma_X)]^p \leq \operatorname{tr} \mathbb{E}(\Sigma_X^p) \leq [\operatorname{tr} \mathbb{E}(\Sigma_X)]^p, \quad (p \geq 1)
\tag{D.4}
$$

Thus, we can write the inequalities

$$
\begin{aligned}
\frac{K[\operatorname{tr} \mathbb{E}_X(\Sigma_X)]^2}{2d[\operatorname{tr} \mathbb{E}_X(\Sigma_X)]^3} &\leq \frac{K[\operatorname{tr} \mathbb{E}_X[(\Phi(X)^\top \Phi(X))^2]]}{2 \operatorname{tr}[\mathbb{E}_X[(\Phi(X)^\top \Phi(X))^3]]} \leq \frac{Kd[\operatorname{tr} \mathbb{E}_X(\Sigma_X)]^2}{2[\operatorname{tr} \mathbb{E}_X(\Sigma_X)]^3} \\
\frac{K}{2d \operatorname{tr} \mathbb{E}_X(\Sigma_X)} &\leq \frac{K[\operatorname{tr} \mathbb{E}_X[(\Phi(X)^\top \Phi(X))^2]]}{2 \operatorname{tr}[\mathbb{E}_X[(\Phi(X)^\top \Phi(X))^3]]} \leq \frac{Kd}{2 \operatorname{tr} \mathbb{E}_X(\Sigma_X)}
\end{aligned}
\tag{D.5}
$$

thereby

$$
\frac{1}{2d\sigma^2(\phi(\boldsymbol{x}_1), \dots, \phi(\boldsymbol{x}_K))} \leq \alpha_{lim}^* \leq \frac{d}{\sigma^2(\phi(\boldsymbol{x}_1), \dots, \phi(\boldsymbol{x}_K))}
\tag{D.6}
$$

$\square$

# E  EXAMPLES

**Example 1 (Normal, Polynomial feature)**   Assume we have $K$ i.i.d samples $x_1, ..., x_K \sim \mathcal{N}(0, \sigma^2)$ and $\boldsymbol{a}$ is a random vector from zero-mean distribution. Consider polynomial basis function $\phi : \mathbb{R} \to \mathbb{R}^d$, where $\phi(y) = (1, ..., y^{d-1})$.

$$
\Phi(X) = \begin{pmatrix} — & \phi(x_1) & — \\ \vdots & \vdots & \vdots \\ — & \phi(x_k) & — \end{pmatrix} = \begin{pmatrix} 1 & x_1 & \dots & x_1^{d-1} \\ \vdots & \vdots & \vdots & \vdots \\ 1 & x_K & \dots & x_K^{d-1} \end{pmatrix}
\tag{E.1}
$$

Since we have

$$
\alpha_{lim}^* = \frac{K tr[\mathbb{E}_X[(\Phi(X)^\top \Phi(X))^2]]}{2 tr[\mathbb{E}_X[(\Phi(X)^\top \Phi(X))^3]]}
\tag{E.2}
$$

So that

$$
\begin{aligned}
tr[\mathbb{E}_X[(\Phi(X)^\top \Phi(X))^2]] &= \mathbb{E}_x \sum_{j=1}^d \left( \sum_{i=1}^K x_i^{(j-1)+0} \right)^2 + \dots + \left( \sum_{i=1}^K x_i^{(j-1)+d} \right)^2 \\
&= \mathbb{E}_x \sum_{j=1}^d \sum_{m=1}^d \left( \sum_{i=1}^K x_i^{j+m-2} \right)^2 \\
&= K \sum_{j=1}^d \sum_{m=1}^d \mathbb{E}[x^{2(j+m-2)}] + (K-1)K \sum_{j=1}^d \sum_{m=1}^d \mathbb{E}^2[x^{(j+m-2)}]
\end{aligned}
\tag{E.3}
$$

Similarly, the denominator is

$$
\begin{aligned}
&tr[\mathbb{E}_X[(\Phi(X)^\top \Phi(X))^3]] \\
&=\mathbb{E}_x \sum_{j=1}^d \sum_{l=1}^d \left[ \sum_{m=1}^d \left( \sum_{i=1}^K x_i^{j+m-2} \right) \left( \sum_{i'=1}^K x_{i'}^{j+l-2} \right) \right] \left[ \left( \sum_{t=1}^K x_t^{j+l-2} \right) \right] \\
&=\mathbb{E}_x \sum_{j=1}^d \sum_{l=1}^d \left[ \underbrace{x_0^{2j+l-3} + \ldots + x_i^{j+d-2} x_{i'}^{j+l-2}}_{dK^2} \right] \left[ \left( \sum_{t=1}^K x_t^{j+l-2} \right) \right] \\
&=\mathbb{E}_x \sum_{j=1}^d \sum_{l=1}^d \left[ \underbrace{x_i^{2j+m+l-4} + \ldots}_{dK} + \underbrace{x_i^{j+d-2} x_{i'}^{j+l-2} + \ldots}_{dK(K-1)} \right] \left[ \left( \sum_{t=1}^K x_t^{j+l-2} \right) \right] \\
&=\sum_{j=1}^d \sum_{l=1}^d \sum_{m=1}^d \left( K\mathbb{E}[x^{3j+m+2l-6}] + 3K(K-1)\mathbb{E}^2[x^{2j+2l-4}]\mathbb{E}[x^{j+m-2}] \right. \\
&\qquad\qquad\qquad \left. + (K^3 - 3K^2 + 2K)\mathbb{E}^3[x^{(j+m+l-3)}] \right)
\end{aligned}
\tag{E.4}
$$

If $K = 1$, $\sigma \to 0$ the optimal $\alpha_{lim}^*$ will be

$$
\begin{aligned}
\alpha_{lim}^* &= \frac{\sum_{j=1}^d \sum_{m=1}^d \mathbb{E}[x^{2(j+m-2)}]}{2\sum_{j=1}^d \sum_{l=1}^d \sum_{m=1}^d \mathbb{E}[x^{3j+m+2l-6}]} \\
&= \frac{\sum_{i=0}^{2d-2} [C(i+1,1) - 2C(i-d+1,1)]\mathbb{E}[x^{2i}]}{2\sum_{j=1}^{3d} g_2(d,j)\mathbb{E}[x^j]} \\
&= \frac{\sum_{i=0}^{2d-2} g_1(d,i)\sigma^{2i}(2i-1)!!}{2\sum_{j=0}^{3d-3} g_2(d,j)\sigma^{2j}(2j-1)!!} = \mathcal{O}\left(\frac{1}{\sigma^2}\right)
\end{aligned}
\tag{E.5}
$$

where $C(n,k) = \begin{pmatrix} n \\ k \end{pmatrix}$ is the binomial coefficient and $g_1(d,i) = C(i+1,1) - 2C(i-d+1,1)$.

If $K \to \infty$, $\sigma \to 0$

$$
\begin{aligned}
\alpha_{lim}^* &= \frac{(K-1)K^2 \sum_{j=1}^d \sum_{m=1}^d \mathbb{E}^2[x^{(j+m-2)}]}{2\sum_{j=1}^d \sum_{l=1}^d \sum_{m=1}^d (K^3 - 3K^2 + 2K)\mathbb{E}^3[x^{(j+m+l-3)}]} \\
&= \frac{\sum_{j=1}^d \sum_{m=1}^d \mathbb{E}^2[x^{(j+m-2)}]}{\sum_{j=1}^d \sum_{l=1}^d \sum_{m=1}^d \mathbb{E}^3[x^{(j+m+l-3)}]} \mathcal{O}(1) \\
&= \frac{\sum_{i=0}^{2d-2} g_1(d,i)\mathbb{E}^2[x^i]}{\sum_{j=0}^{3d-3} g_3(d,j)\mathbb{E}^3[x^j]} \mathcal{O}(1) \\
&= \frac{\sum_{i=1}^{d-1} g_1(d,i)[\sigma^{2i}(2i-1)!!]^2}{\sum_{j=1}^{\lceil 3d/2\rceil-1} g_3(d,j)[\sigma^{2j}(2j-1)!!]^3} \mathcal{O}(1) = \mathcal{O}\left(\frac{1}{\sigma^2}\right)
\end{aligned}
\tag{E.6}
$$

We show the coefficients of each moment in the Figure 6. As we can see, denominator becomes dominant since the coefficient of every moment, number of terms and order of moment are all larger (higher) than numerator.

So in this case, the $\alpha_{lim}^*$ has an inverse relationship with $\sigma^2$.

**Example 2 (Random Matrices)**  Assume all elements $Y_{ij}$ in feature matrix are independent. Then let $Y$ be a random matrix we have

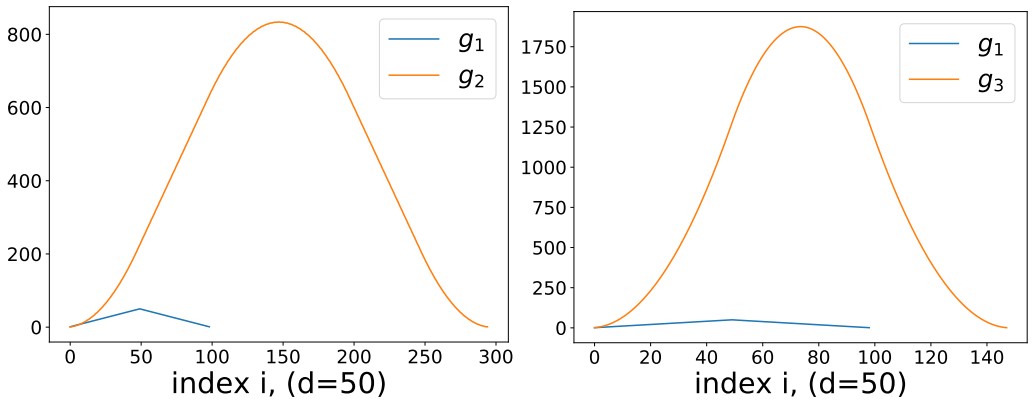

Figure 6: Example of $d = 50$ with coefficients of each moment given by $g_1(d, i), g_2(d, i), g_3(d, i)$.

$$\Phi(X)^\top \Phi(X) = \begin{pmatrix} \sum_{i=1}^K Y_{i1}^2 & \cdots & \sum_{i=1}^K Y_{i1} Y_{id} \\ \vdots & \vdots & \vdots \\ \sum_{i=1}^K Y_{id} Y_{i1} & \cdots & \sum_{i=1}^K Y_{id}^2 \end{pmatrix} \tag{E.7}$$

For the numerator in (D.1),

$$\begin{aligned} tr[\mathbb{E}_{\boldsymbol{x}}[(\Phi(X)^\top \Phi(X))^2]] =& \mathbb{E}_{Y_{ij}, i \in [K], j \in [d]} \sum_{t=1}^d \sum_{s=1}^d \left( \sum_{i=1}^K Y_{it} Y_{is} \right) \left( \sum_{i'=1}^K Y_{i't} Y_{i's} \right) \\ =& dK\mathbb{E}[Y^4] + (dK(K-1) + d(d-1)K)\mathbb{E}^2[Y^2] \\ & + d(d-1)K(K-1)\mathbb{E}^4[Y] \end{aligned} \tag{E.8}$$

Here, the row $m$ column $n$ entry of $(\Phi(X)^\top \Phi(X))^2$ is

$$(\Phi(X)^\top \Phi(X))^2_{(mn)} = \sum_{s=1}^d \left( \sum_{i=1}^K Y_{im} Y_{is} \right) \left( \sum_{i'=1}^K Y_{i'n} Y_{i's} \right) \tag{E.9}$$

The diagonal entry of $(\Phi(X)^\top \Phi(X))^3$ at row $m$ will given by

$$(\Phi(X)^\top \Phi(X))^3_{(mm)} = \sum_{n=1}^d \left[ \sum_{s=1}^d \left( \sum_{i=1}^K Y_{im} Y_{is} \right) \left( \sum_{i'=1}^K Y_{i'n} Y_{i's} \right) \left( \sum_{j=1}^K Y_{jm} Y_{js} \right) \right] \tag{E.10}$$

Similarly, the denominator is

$$tr[\mathbb{E}_{\boldsymbol{x}}[(\Phi(X)^\top \Phi(X))^3]]$$

$$=\mathbb{E}_{Y_{ij}, i \in [K], j \in [d]} \sum_{m=1}^d \sum_{n=1}^d \left[ \sum_{s=1}^d \left( \sum_{i=1}^K Y_{im} Y_{is} \right) \left( \sum_{i'=1}^K Y_{i'n} Y_{i's} \right) \left( \sum_{j=1}^K Y_{jm} Y_{js} \right) \right]$$

$$= \underbrace{Kd\mathbb{E}[Y^6] + Kd(d-1)(\mathbb{E}[Y^5]\mathbb{E}[Y] + \mathbb{E}[Y^4]\mathbb{E}[Y^2] + \mathbb{E}^2[Y^3])}_{i=i'=j}$$

$$+ \underbrace{b_1 + K(K-1)d(d-1)(\mathbb{E}[Y^3]\mathbb{E}^3[Y] + \mathbb{E}[Y^3]\mathbb{E}[Y^2]\mathbb{E}[Y] + \mathbb{E}^2[Y^2]\mathbb{E}^2[Y])}_{i=i'\neq j}$$

$$+ \underbrace{b_1 + K(K-1)d(d-1)(\mathbb{E}[Y^3]\mathbb{E}^3[Y] + \mathbb{E}[Y^3]\mathbb{E}[Y^2]\mathbb{E}[Y] + \mathbb{E}^2[Y^2]\mathbb{E}^2[Y])}_{i\neq i'=j}$$

$$+ \underbrace{b_1 + K(K-1)d(d-1)(\mathbb{E}[Y^4]\mathbb{E}^2[Y] + \mathbb{E}^3[Y^2] + \mathbb{E}^2[Y^2]\mathbb{E}^2[Y])}_{i=j\neq i'}$$

$$+ \underbrace{b_2 + (K^3 - 3K^2 + 2K)d(d-1)(\mathbb{E}^6[Y] + \mathbb{E}^2[Y^2]\mathbb{E}^2[Y] + \mathbb{E}[Y^2]\mathbb{E}^4[Y])}_{i\neq i'\neq j} \tag{E.11}$$

where $b_1 = K(K-1)d\mathbb{E}[Y^4]\mathbb{E}[Y^2]$, $b_2 = (K^3 - 3K^2 + 2K)d\mathbb{E}^3[Y^2]$.

If $K = 1$, the optimal $\alpha^*_{lim}$ will be

$$\alpha^*_{lim} = \frac{d\mathbb{E}[Y^4]}{2d\mathbb{E}[Y^6]} = \frac{\mathbb{E}[Y^4]}{2\mathbb{E}[Y^6]} \tag{E.12}$$

If $K \to \infty$ and $d \to \infty$,

$$
\begin{aligned}
\alpha^*_{lim} =& \frac{K(dK(K-1) + d(d-1)K)\mathbb{E}^2[Y^2] + d(d-1)K^2(K-1)\mathbb{E}^4[Y]}{(K^3 - 3K^2 + 2K)[d\mathbb{E}^3[Y^2] + d(d-1)(\mathbb{E}^6[Y] + \mathbb{E}^2[Y^2]\mathbb{E}^2[Y] + \mathbb{E}[Y^2]\mathbb{E}^4[Y])]} \\
=& \frac{d\mathbb{E}^2[Y^2] + d(d-1)\mathbb{E}^4[Y]}{d\mathbb{E}^3[Y^2] + d(d-1)(\mathbb{E}^6[Y] + \mathbb{E}^2[Y^2]\mathbb{E}^2[Y] + \mathbb{E}[Y^2]\mathbb{E}^4[Y])}\mathcal{O}(1) \\
=& \frac{\mathbb{E}^4[Y]}{\mathbb{E}^6[Y] + \mathbb{E}^2[Y^2]\mathbb{E}^2[Y] + \mathbb{E}[Y^2]\mathbb{E}^4[Y]}\mathcal{O}(1) \approx \frac{\mathbb{E}[Y^4]}{\mathbb{E}[Y^6]}\mathcal{O}(1)
\end{aligned}
\tag{E.13}
$$

Both two examples are related to the high order moments of data distributions. In polynomial feature example, we focus on the gaussian distributed data and its inverse relationship to data variance. As for any random matrix, it depends on the fourth moment over sixth moment.

## F  PROPOSITION FOR THEOREM 2

**Proposition 5.** $\exists \varepsilon$, $\alpha \in [-\varepsilon, \varepsilon]$ global minimum of MAML $\boldsymbol{w}_m(\alpha)$ is given by following equation

$$
\begin{aligned}
\boldsymbol{w}_m(\alpha) =& \boldsymbol{w}_r + \alpha \left( \sum_i^N \Phi(X_i)^\top \Phi(X_i) \right)^{-1} \left[ \sum_j^N \frac{4}{K}(\Phi(X_j)^\top \Phi(X_j))^2 (\boldsymbol{w}_s - \boldsymbol{a}_j) \right] \\
&+ \int_0^\alpha \frac{\nabla_\alpha^2 \boldsymbol{w}_m^0(\xi)}{2!}(\alpha - \xi)^2 d\xi
\end{aligned}
\tag{F.1}
$$

where $\boldsymbol{w}_r$ is ERM global minimum.

*Proof.* As for the MAML, the global minimum is,

$$\boldsymbol{w}_m(\alpha) = \left( \sum_{i=1}^N C_i^\top C_i \right)^{-1} \left( \sum_{j=1}^N C_j^\top C_j \boldsymbol{a}_j \right) \tag{F.2}$$

with $C_i(\alpha) = \Phi_i - \frac{2\alpha}{K}\Phi_i\Phi_i^\top \Phi_i$, $C_i(\alpha) \in \mathbb{R}^{K \times d}$.

Let $W_\alpha = \sum_{i=1}^N C_i^\top(\alpha)C_i$ and $\boldsymbol{\nu}_\alpha$ denote $\sum_{j=1}^N C_j^\top C_j \boldsymbol{a}_j$, then

$$
\begin{aligned}
\nabla_\alpha W_\alpha =& \nabla_\alpha \left[ \sum_{i=1}^N \left( \Phi_i - \frac{2\alpha}{K}\Phi_i\Phi_i^\top \Phi_i \right)^\top \left( \Phi_i - \frac{2\alpha}{K}\Phi_i\Phi_i^\top \Phi_i \right) \right] \\
=& \left[ \sum_{i=1}^N -\frac{4}{K}(\Phi_i^\top \Phi_i)^2 + \frac{8\alpha}{K^2}(\Phi_i^\top \Phi_i)^3 \right]
\end{aligned}
\tag{F.3}
$$

Similarly, we have

$$\nabla_\alpha \boldsymbol{\nu}_\alpha = \left[ \sum_{i=1}^N -\frac{4}{K}(\Phi_i^\top \Phi_i)^2 \boldsymbol{a}_i + \frac{8\alpha}{K^2}(\Phi_i^\top \Phi_i)^3 \boldsymbol{a}_i \right] \tag{F.4}$$

For first-order derivative, we have the following form,

$$
\nabla_\alpha \boldsymbol{w}_m(0) = W_\alpha^{-1}|_{\alpha=0} \left[ \nabla_\alpha \boldsymbol{\nu}_\alpha|_{\alpha=0} - \nabla_\alpha W_\alpha|_{\alpha=0} \boldsymbol{w}_m(0) \right]
$$

$$
= \left( \sum_{i=1}^{N} \Phi_i^\top \Phi_i \right)^{-1} \left[ \sum_{j=1}^{N} -\frac{4}{K} (\Phi_j^\top \Phi_j)^2 \boldsymbol{a}_j + \sum_{l=1}^{N} \frac{4}{K} (\Phi_l^\top \Phi_l)^2 \boldsymbol{w}_m(0) \right] \tag{F.5}
$$

$$
= \left( \sum_{i=1}^{N} \Phi_i^\top \Phi_i \right)^{-1} \left[ \sum_{j=1}^{N} \frac{4}{K} (\Phi_j^\top \Phi_j)^2 (\boldsymbol{w}_m(0) - \boldsymbol{a}_j) \right]
$$

Recall that

$$
\boldsymbol{w}_m(0) = \boldsymbol{w}_s = \left( \sum_{i=1}^{N} \Phi_i^\top \Phi_i \right)^{-1} \left( \sum_{j=1}^{N} \Phi_j^\top \Phi_j \boldsymbol{a}_j \right) \tag{F.6}
$$

So for small $\alpha$, we have the Taylor expansion at zero that

$$
\boldsymbol{w}_m(\alpha) = \boldsymbol{w}_m(0) + \nabla_\alpha \boldsymbol{w}_m(0) + \int_0^\alpha \frac{\nabla_\alpha^2 \boldsymbol{w}_m^0(\xi)}{2!} (\alpha - \xi)^2 d\xi
$$

$$
= \boldsymbol{w}_r + \alpha \left( \sum_i^{N} \Phi(X_i)^\top \Phi(X_i) \right)^{-1} \left[ \sum_j^{N} \frac{4}{K} (\Phi(X_j)^\top \Phi(X_j))^2 (\boldsymbol{w}_s - \boldsymbol{a}_j) \right] \tag{F.7}
$$

$$
+ \int_0^\alpha \frac{\nabla_\alpha^2 \boldsymbol{w}_m^0(\xi)}{2!} (\alpha - \xi)^2 d\xi
$$

$\square$

# G   PROOF OF THEOREM 2

**Proof Sketch**   We list our proof steps as follows

1. Based on Definition 1, our target is to illustrate that fast adaptation distance gap between $\boldsymbol{w}_m$ and $\boldsymbol{w}_r$ is always negative which means MAML has smaller distance to all the tasks at any adaptation steps in expectation.

2. We first get the linearized expression of $\boldsymbol{w}_m$ by Proposition 5.

3. Compute fast adaptation distance gap $\boldsymbol{\Delta} = \mathbb{E}_{T_1,\ldots,T_N \sim \mathcal{D}(T)} F_t(\boldsymbol{w}_m) - \mathcal{F}_t(\boldsymbol{w}_r)$ across same task distribution $\mathcal{D}(T)$ and take expectations with respect to all random variables.

4. With lemma of trace inequalities and assumptions, we can get the upper bound for the dominant term of $\boldsymbol{\Delta}$, refer to (G.30).

5. Bound the reminder terms, we can get the range of $\alpha$.

**Notation for this proof** Follow the Theorem 1, we omit the arguments of the function if its symbol has a index e.g. $\Phi_T = \Phi(X_T)$. Covariance matrix $\Phi_T^\top \Phi_T = G_T$ and inverse of sum covariance matrix $V = (\sum_{i \in [N]} \Phi_i^\top \Phi_i)^{-1}$ for short.

*Proof.* For each task $T$ sampled from distribution $\mathcal{D}(T)$, gradient descent iteration yields

$$
\boldsymbol{w}_T^{t+1} = \boldsymbol{w}_T^t - \eta \nabla \ell_T(\boldsymbol{w}_T^t)
$$

$$
= \boldsymbol{w}_T^t - \frac{2\eta}{K} \Phi_T^\top (\Phi_T \boldsymbol{w}_T - \Phi_T \boldsymbol{a}_T)
$$

$$
= \left( I - \frac{2\eta}{K} \Phi_T^\top \Phi_T \right) \boldsymbol{w}_T^t + \frac{2\eta}{K} \Phi_T^\top \Phi_T \boldsymbol{a}_T \tag{G.1}
$$

$$
= \left( I - \frac{2\eta}{K} \Phi_T^\top \Phi_T \right)^{t+1} \boldsymbol{w}^0 + \sum_{j=1}^{t+1} \frac{2\eta}{K} \left( I - \frac{2\eta}{K} \Phi_T^\top \Phi_T \right)^{j-1} \Phi_T^\top \Phi_T \boldsymbol{a}_T
$$

where $w^0$ is the initialization. Let $G_T = \Phi_T^\top \Phi_T$, the adapted error will be

$$\|w_T^t - a_T\| = \left\| \left(I - \frac{2\eta}{K}G_T\right)^t w_m^0 + \sum_{j=1}^t \frac{2\eta}{K}\left(I - \frac{2\eta}{K}G_T\right)^{j-1} G_T a_T - a_T \right\|^2 \tag{G.2}$$

For simplicity, let

$$Q_T = \left(I - \frac{2\eta}{K}G_T\right), \quad S_T = \sum_{j=1}^t \frac{2\eta}{K}\left(I - \frac{2\eta}{K}G_T\right)^{j-1} G_T a_T - a_T$$

then with Definition 1, we can get $t$-step fast adaptation error for MAML,

$$\mathcal{F}_t(w_m^0) = \mathop{\mathbb{E}}_{T\sim\mathcal{D}(T)} \left\| Q_T^t w_m^0 + S_T \right\|^2 \tag{G.3}$$

and the ERM fast adaptation error is

$$\mathcal{F}_t(w_r^0) = \mathop{\mathbb{E}}_{T\sim\mathcal{D}(T)} \left\| Q_T^t w_r^0 + S_T \right\|^2 \tag{G.4}$$

Note that the sum of geometric series in $S_T$ is

$$S_T = \sum_{j=1}^t \frac{2\eta}{K}\left(I - \frac{2\eta}{K}G_T\right)^{j-1} G_T a_T - a_T = \left[I - \left(I - \frac{2\eta}{K}G_T\right)^t - I\right] a_T = -Q_T^t a_T \tag{G.5}$$

Now, let's focus on the error gap of MAML and ERM, denoted as $\boldsymbol{\Delta}$, then we have

$$\boldsymbol{\Delta} = \mathop{\mathbb{E}}_{T_1,\dots,T_N\sim\mathcal{D}(T)} \mathcal{F}_t(w_m^0) - \mathcal{F}_t(w_r^0)$$

$$= \mathop{\mathbb{E}}_{T_1,\dots,T_N,T\sim\mathcal{D}(T)} \left\langle Q_T^t\left(w_m^0 + w_r^0\right) + 2S_T, Q_T^t\left(w_m^0 - w_r^0\right) \right\rangle \tag{G.6}$$

For small $\alpha$, we get its linear expansion in Proposition 5

$$w_m^0(\alpha) = w_r^0 + \alpha\nabla_\alpha w_m^0(0) + \int_0^\alpha \frac{\nabla_\alpha^2 w_m^0(\xi)}{2!}(\alpha - \xi)^2 d\xi$$

$$= w_r^0 + \alpha\left(\sum_j^N G_j\right)^{-1}\left[\sum_j^N \frac{4}{K}G_j^2\left(w_r^0 - a_j\right)\right] + R_1 \tag{G.7}$$

where $R_1 = \int_0^\alpha \frac{\nabla_\alpha^2 w_m^0(\xi)}{2!}(\alpha - \xi)^2 d\xi$ is the reminder term. So it would be

$$\boldsymbol{\Delta} = \mathop{\mathbb{E}}_{w_m^0, w_r^0} \mathop{\mathbb{E}}_{T\sim\mathcal{D}(T)} \left\langle Q_T^t\left(2w_r^0 + \alpha\nabla_\alpha w_m^0(0) + R_1\right) + 2S_T, \alpha Q_T^t\nabla_\alpha w_m^0(0) \right\rangle \tag{G.8}$$

Let $V = \left(\sum_j^N G_j\right)^{-1}$ and the first derivative $\nabla_\alpha w_m^0(0)$ is split into

$$\alpha\nabla_\alpha w_m^0(0) = V\left(\sum_j^N \frac{4}{K}G_j^2 w_r^0\right) - V\left(\sum_j^N \frac{4}{K}G_j^2 a_j\right) \tag{G.9}$$

thus inner product will be four product terms and a reminder term which is

$$\boldsymbol{\Delta} = \frac{8\alpha}{K} \mathop{\mathbb{E}}_{\{a_i\}_{[N]}} \mathop{\mathbb{E}}_{T\sim\mathcal{D}(T)} \left\{ \left\langle Q_T^t w_r^0, Q_T^t V\left(\sum_j^N G_j^2 w_r^0\right)\right\rangle - \left\langle Q_T^t w_r^0, Q_T^t V\left(\sum_j^N G_j^2 a_j\right)\right\rangle \right.$$

$$\left. + \left\langle S_T, Q_T^t V\left(\sum_j^N G_j^2 w_r^0\right)\right\rangle - \left\langle S_T, Q_T^t V\left(\sum_j^N G_j^2 a_j\right)\right\rangle \right\} + R_2 + R_1' \tag{G.10}$$

where the remainder terms are

$$R_2 = \alpha^2 \left\langle \nabla_\alpha \boldsymbol{w}_m^0(0), \nabla_\alpha \boldsymbol{w}_m^0(0) \right\rangle, \quad R_1' = \alpha^2 \left\langle R_1, \nabla_\alpha \boldsymbol{w}_m^0(0) \right\rangle \tag{G.11}$$

Now, let's look at the expectation. Recall that task $T$ is defined by random variables $(\Phi(X_T), \boldsymbol{a}_T)$. With simultaneously diagonalizable assumption, all feature covariance matrix in tasks can be factorized to

$$\Phi(X_T)^\top \Phi(X_T) = G_T = U\Sigma_T U^* \tag{G.12}$$

where $U$ is the basis of features, $\Sigma_T$ is the only random variable of $G_T$. So the data of each task depends on eigenvalue diagonal matrix $\Sigma_T$. So taking expectation over $T$ means the two independent expectation $\mathbb{E}_{\Sigma_T \sim \mathcal{D}(\Sigma)}, \mathbb{E}_{\boldsymbol{a}_T \sim \mathcal{D}(\boldsymbol{a})}$. Similarly, $\boldsymbol{w}_r^0$ in previous section is expressed by

$$\boldsymbol{w}_r^0 = \left( \sum_{i=1}^N G_i \right)^{-1} \left( \sum_{j=1}^N G_j \boldsymbol{a}_j \right) \tag{G.13}$$

$$\Rightarrow \mathbb{E}_{T_1,\dots,T_N}(\boldsymbol{w}_r^0) = \mathbb{E}_{\{\boldsymbol{a}_i\}_{[N]}} \mathbb{E}_{\{G_i\}_{[N]}}(\boldsymbol{w}_r^0) = \mathbb{E}_{\{\boldsymbol{a}_i\}_{[N]}} \mathbb{E}_{\{\Sigma_i\}_{[N]}}(\boldsymbol{w}_r^0)$$

For each product term in (G.10), we list four main terms as following. First term is

$$\mathbb{E}_{T_1,\dots,T_N,T \sim \mathcal{D}(T)} \left\langle Q_T^t \boldsymbol{w}_r^0, Q_T^t V \sum_j^N G_j^2 \boldsymbol{w}_r^0 \right\rangle$$

$$= \mathbb{E}_{\{\boldsymbol{a}_i\}_{[N]} \sim \mathcal{D}(\boldsymbol{a})} \mathbb{E}_{\{G_i\}_{[N]}} \left\langle Q_T^t V \left( \sum_i^N G_i \boldsymbol{a}_i \right), \frac{4\alpha}{K} Q_T^t V \sum_j^N G_j^2 V \left( \sum_k^N G_k \boldsymbol{a}_k \right) \right\rangle \tag{G.14}$$

$$= \mathbb{E}_{\{G_i\}_{[N]}} \operatorname{tr} \left[ \sum_{i=1}^N G_i V Q_T^t Q_T^t V \left( \sum_j^N G_j^2 \right) V G_i \right]$$

Similarly, we can get the second term

$$\mathbb{E}_{\{T_i\}_{[N]}, T \sim \mathcal{D}(T)} \left\langle Q_T^t \boldsymbol{w}_r^0, Q_T^t V \sum_j^N G_j^2 \boldsymbol{a}_j \right\rangle = \mathbb{E}_{\{\boldsymbol{a}_i\}_{[N]}} \mathbb{E}_{\{G_i\}_{[N]}} \left\langle Q_T^t \boldsymbol{w}_r^0, Q_T^t V \sum_j^N G_j^2 \boldsymbol{a}_j \right\rangle$$

$$= \mathbb{E}_{\{G_i\}_{[N]}} \operatorname{tr} \left[ \sum_{i=1}^N G_i V Q_T^t Q_T^t V G_i^2 \right] \tag{G.15}$$

For third and fourth terms, $\mathbb{E}_{\boldsymbol{a}_T \sim \mathcal{D}(\boldsymbol{a})}$ is a marginal expectation of $\mathbb{E}_{T \sim \mathcal{D}(T)}$, thus

$$\mathbb{E}_{\{T_i\}_{[N]}, T \sim \mathcal{D}(T)} \left\langle S_T, Q_T^t V \sum_j^N G_j^2 \boldsymbol{w}_r^0 \right\rangle = \mathbb{E}_{\boldsymbol{a}_T \sim \mathcal{D}(\boldsymbol{a})} \mathbb{E}_{\boldsymbol{w}_r^0, \{G_i\}_{[N]}} \left\langle -Q_T^t \boldsymbol{a}_T, Q_T^t V \sum_j^N G_j^2 \boldsymbol{w}_r^0 \right\rangle$$

$$= \boldsymbol{0} \tag{G.16}$$

Similarly, the fourth term is

$$\mathbb{E}_{\boldsymbol{a}_{[N]} \sim \mathcal{D}(\boldsymbol{a})} \mathbb{E}_{T \sim \mathcal{D}(T)} \left\langle S_T, Q_T^t V \sum_j^N G_j^2 \boldsymbol{a}_j \right\rangle = \boldsymbol{0} \tag{G.17}$$

So overall, the we care about above four terms as a function of $\alpha, N, t, ...$ denoted as $\delta_t(\alpha, N)$

$$
\begin{aligned}
\delta_t(\alpha, N) =& \mathbf{\Delta} - R_2 - R_1' \\
=& \frac{8\alpha}{K} \underset{\{G_i\}_{[N]}}{\mathbb{E}} \ \mathrm{tr} \left[ \sum_{i=1}^N G_i V Q_T^t Q_T^t V \left( \sum_j^N G_j^2 \right) V G_i \right] \\
& - \frac{8\alpha}{K} \underset{\{G_i\}_{[N]}}{\mathbb{E}} \ \mathrm{tr} \left[ \sum_{i=1}^N G_i V Q_T^t Q_T^t V G_i^2 \right] \\
=& \frac{8\alpha}{K} \underset{\{G_i\}_{[N]}}{\mathbb{E}} \ \mathrm{tr} \left[ \sum_{i=1}^N G_i V Q_T^t Q_T^t V \left( \sum_j^N G_j^2 \right) V G_i - \sum_{i=1}^N G_i V Q_T^t Q_T^t V G_i^2 \right] \\
=& \frac{8\alpha}{K} \underset{\{G_i\}_{[N]}}{\mathbb{E}} \ \mathrm{tr} \left[ \sum_{i=1}^N G_i V Q_T^t Q_T^t V \left( \left( \sum_j^N G_j^2 \right) V G_i - G_i^2 \right) \right] \\
=& \frac{8\alpha}{K} \underset{\{G_i\}_{[N]}}{\mathbb{E}} \ \mathrm{tr} \left[ \sum_{i=1}^N V Q_T^t Q_T^t V \left( \left( \sum_j^N G_j^2 \right) V G_i^2 - G_i^3 \right) \right]
\end{aligned}
\tag{G.18}
$$

By simultaneously diagonalizable assumption, we have

$$
\begin{aligned}
\delta_t(\alpha, N) =& \frac{8\alpha}{K} \underset{\{\Sigma_i\}_{[N]}}{\mathbb{E}} \ \mathrm{tr} \left[ \sum_{i=1}^N V Q_T^t Q_T^t V \left( \sum_j^N (U\Sigma_j^2 U^\top)(U\widehat{\Sigma}_N U^\top)(U\Sigma_i^2 U^\top) - (U\Sigma_i^3 U^\top) \right) \right] \\
=& \frac{8\alpha}{K} \underset{\{\Sigma_i\}_{[N]}}{\mathbb{E}} \ \mathrm{tr} \left[ V Q_T^t Q_T^t V \left( \sum_i^N \sum_j^N U\Sigma_i^2 \widehat{\Sigma}_N \Sigma_j^2 U^\top - \sum_i^N U\Sigma_i^3 U^\top \right) \right]
\end{aligned}
\tag{G.19}
$$

where $\widehat{\Sigma}_N = \left( \sum_{k \in [N]} \Sigma_k \right)^{-1}$ is a PD matrix and $Q_T^t = (I - (2\eta/K)G_T)^t$ is a exponential decay term w.r.t $\eta$. With probability 1, $\lambda_s I \preceq \Sigma_i \preceq \lambda_x I$

$$
(N\lambda_x)^{-1} I \preceq \widehat{\Sigma}_N \preceq (N\lambda_s)^{-1} I
\tag{G.20}
$$

Note that $V Q_T^t Q_T^t V = (Q_T^t V)^\top Q_T^t V$ is a symmetric positive definite matrix where

$$
\begin{aligned}
V Q_T^t Q_T^t V =& U\widehat{\Sigma}_N U^\top Q_T^{2t} U\widehat{\Sigma}_N U^\top \\
=& U\widehat{\Sigma}_N U^\top \left( UU^\top - \frac{2\eta}{K} U\Sigma_T U^\top \right)^{2t} U\widehat{\Sigma}_N U^\top \\
=& U\widehat{\Sigma}_N U^\top U \left( I - \frac{2\eta}{K}\Sigma_T \right)^{2t} U^\top U\widehat{\Sigma}_N U^\top \\
=& U\widehat{\Sigma}_N \left( I - \frac{2\eta}{K}\Sigma_T \right)^{2t} \widehat{\Sigma}_N U^\top
\end{aligned}
\tag{G.21}
$$

Note that $m$-th diagonal entry of $\mathbb{E}_{\{\Sigma_i\}_{[N]}} \left( \sum_i^N \sum_j^N U\Sigma_i^2 \widehat{\Sigma}_N \Sigma_j^2 U^\top - \sum_i^N U\Sigma_i^3 U^\top \right)$ is

$$
\vec{e}_{(m)}^\top \left( \frac{(E\lambda^2)^2}{NE\lambda} - \frac{E\lambda E\lambda^3}{NE\lambda} \right) \vec{e}_{(m)} = \frac{(E\lambda^2)^2 - E\lambda E\lambda^3}{NE\lambda} \|\vec{e}_{(m)}\|^2 \overset{\text{(C-S)}}{\leq} 0
\tag{G.22}
$$

where (C-S) is according to Cauchy-Schwarz inequality for integrals in Proposition 3.

So the above matrix is a negative definite matrix. Now, let us can derive following trace inequality for NSD and PSD.

**Proposition 6.** *If A is a n-by-n negative definite matrix and B is a n-by-n PSD matrix, we have*

$$
tr(AB) \leq \lambda_{\min}(B) tr(A)
\tag{G.23}
$$

*Proof.* By Ruhe's trace inequality (Lemma 4), we have $tr(AB) \geq \sum_i \lambda_i(A)\lambda_{n-i+1}(B)$. Eigenvalues of $A$ are negative where $\lambda_i(A) < 0, \forall i \in [n]$. So we have $tr(AB) \leq \lambda_{\min}(B)\sum_i \lambda_i(A) = \lambda_{\min}(B)tr(A)$ $\square$

So with Proposition 6, we have

$$\delta_t(\alpha, N) \leq \frac{8\alpha}{K} \mathop{\mathbb{E}}_{\{\Sigma_i\}_{[N]}} \lambda_{\min}\left(VQ_T^tQ_T^tV\right) \text{tr}\left(\sum_i^N \sum_j^N U\Sigma_i^2 \widehat{\Sigma}_N \Sigma_j^2 U^\top - \sum_i^N U\Sigma_i^3 U^\top\right) \quad \text{(G.24)}$$

with (G.20) and (G.21)

$$\lambda_{\min}\left(VQ_T^tQ_T^tV\right) = \lambda_{\min}\left(\widehat{\Sigma}_N\left(I - \frac{2\eta}{K}\Sigma_T\right)^{2t}\widehat{\Sigma}_N\right) = \frac{1}{N^2\lambda_x^2}\left(I - \frac{2\eta\lambda_x}{K}\right)^{2t} \quad \text{(G.25)}$$

For the second part, we have

$$\text{tr}\left(\sum_i^N \sum_j^N U\Sigma_i^2 \widehat{\Sigma}_N \Sigma_j^2 U^\top - \sum_i^N U\Sigma_i^3 U^\top\right) = \text{tr}\left[\sum_i^N \sum_j^N \Sigma_i^2\left(\sum_k^N \Sigma_k\right)^{-1}\Sigma_j^2 - \sum_i^N \Sigma_i^3\right]$$

$$= \text{tr}\left[\left(\sum_k^N \Sigma_k\right)^{-1}\sum_i^N \sum_j^N \Sigma_j^2\Sigma_i^2 - \sum_i^N \Sigma_i^3\right]$$

$$\leq \lambda_{\min}\left[\left(\sum_k^N \Sigma_k\right)^{-1}\right]\text{tr}\left[\sum_i^N \sum_j^N \Sigma_j^2\Sigma_i^2 - \sum_i^N \sum_k^N \Sigma_i^3\Sigma_k\right]$$

$$= \frac{1}{\lambda_x N}\left[\left(\sum_k^N \Sigma_k\right)^{-1}\right]\text{tr}\left[\sum_i^N \sum_j^N \Sigma_j^2\Sigma_i^2 - \sum_i^N \sum_k^N \Sigma_i^3\Sigma_k\right]$$

$$\quad \text{(G.26)}$$

Overall, with probability 1, we have

$$\delta_t(\alpha, N) \leq \frac{8\alpha d}{N^3 K \lambda_x^3}\left(1 - \frac{2\eta\lambda_x}{K}\right)^{2t}\mathop{\mathbb{E}}_{\{\Sigma_i\}_{[N]}} \text{tr}\left[\sum_i^N \sum_j^N \Sigma_j^2\Sigma_i^2 - \sum_i^N \sum_k^N \Sigma_i^3\Sigma_k\right] \quad \text{(G.27)}$$

Specifically,

$$\mathop{\mathbb{E}}_{\{\Sigma_i\}_{[N]}} \text{tr}\left[\sum_i^N \sum_j^N \Sigma_j^2\Sigma_i^2 - \sum_i^N \sum_k^N \Sigma_i^3\Sigma_k\right]$$

$$= \text{tr}\left[\underbrace{N\mathbb{E}(\Sigma^4)}_{i=j} - \underbrace{N\mathbb{E}(\Sigma^4)}_{i=k} + \underbrace{(N^2 - N)\mathbb{E}^2(\Sigma^2)}_{i\neq j} - \underbrace{(N^2 - N)\mathbb{E}(\Sigma^3)\mathbb{E}(\Sigma)}_{i\neq k}\right] \quad \text{(G.28)}$$

$$= (N^2 - N)\sum_{i=1}^d [\mathbb{E}(\Sigma^2)]_{(i)}^2 - [\mathbb{E}(\Sigma^3)]_{(i)}[\mathbb{E}(\Sigma)]_{(i)}$$

With Proposition 3 we know that any eigenvalue $\lambda = \Sigma_{(i)} > 0$ obeys the statistical condition $\mathbb{E}^2[\lambda^2] - \mathbb{E}[\lambda^3]\mathbb{E}[\lambda]$. Thus there exists a constant $\widetilde{c} > 0$ such that

$$\mathop{\mathbb{E}}_{\{\Sigma_i\}_{[N]}} \text{tr}\left[\sum_i^N \sum_j^N \Sigma_j^2\Sigma_i^2 - \sum_i^N \sum_k^N \Sigma_i^3\Sigma_k\right] < -\widetilde{c}d(N^2 - N) \quad \text{(G.29)}$$

Finally, we have

$$\delta_t(\alpha, N) \leq -\left(1 - \frac{2\eta}{K}\lambda_x\right)^{2t}\frac{8\alpha d^2\widetilde{c}}{K\lambda_x^3}\left(\frac{1}{N} - \frac{1}{N^2}\right) \quad \text{(G.30)}$$

Now, let's bound the remainder terms $R'_1, R_2$ where

$$R'_1 = \mathop{\mathbb{E}}_{\boldsymbol{a}_{[N]} \sim \mathcal{D}(\boldsymbol{a})} \mathop{\mathbb{E}}_{T \sim \mathcal{D}(T)} \left\langle Q^t_T R_1, \alpha Q^t_T V \left( \sum_j^N \frac{4}{K} G^2_j \left( \boldsymbol{w}^0_r - \boldsymbol{a}_j \right) \right) \right\rangle \tag{G.31}$$

$R_1$ is the remainder term in Taylor expansion (G.7). With Integral Mean Value Theorem

$$R_1 = \int_0^\alpha \frac{\nabla^2_\alpha \boldsymbol{w}^0_m(\xi)}{2!} (\alpha - \xi)^2 d\xi = \frac{\alpha^2}{2} \nabla^2_\alpha \boldsymbol{w}^0_m(\xi') \tag{G.32}$$

We have the locally Lipschitz property for $C^\infty$ function $\boldsymbol{w}_m(\alpha)$, in a small region with small $\alpha$ such that

$$R_1 \approx \frac{\alpha^2}{2} \nabla^2_\alpha \boldsymbol{w}^0_m(0) \tag{G.33}$$

Then we have

$$\begin{aligned}
\nabla^2_\alpha \boldsymbol{w}^0_m(0) =& \frac{8}{K^2} \left( \sum_{k=1}^N G_k \right)^{-1} \left( \sum_{i=1}^N G^3_i \boldsymbol{a}_i - G^3_i \boldsymbol{w}^0_s + K G^2_i \nabla_\alpha \boldsymbol{w}^0_m(\alpha) \right) \\
=& \frac{8}{K^2} \left( \sum_{k=1}^N G_k \right)^{-1} \left( \sum_{i=1}^N \sum_{j=1}^N G^3_i G_j \boldsymbol{a}_i - G^3_i G_j \boldsymbol{w}^0_s + 4 G^2_i G^2_j \left( \boldsymbol{w}^0_r - \boldsymbol{a}_j \right) \right) \\
=& \frac{8}{K^2} \left( \sum_{k=1}^N G_k \right)^{-2} \left[ \sum_{i=1}^N \sum_{j=1}^N (4 G^2_i G^2_j \boldsymbol{w}^0_r - G^3_i G_j \boldsymbol{w}^0_s) + \left( G^3_i G_j \boldsymbol{a}_i - 4 G^2_i G^2_j \boldsymbol{a}_j \right) \right] \\
=& \frac{8}{K^2} \left( \sum_{k=1}^N G_k \right)^{-2} \left[ \sum_{i=1}^N \sum_{j=1}^N (4 G^2_i G^2_j - G^3_i G_j)(\boldsymbol{w}^0_r - \boldsymbol{a}_j) \right]
\end{aligned} \tag{G.34}$$

Recall (F.6) the $\boldsymbol{w}^0_r = (\sum_i^N \Phi^\top_i \Phi_i)^{-1}(\sum_{i'}^N \Phi^\top_{i'} \Phi_{i'} \boldsymbol{a}_{i'})$ while $\mathbb{E}_{\boldsymbol{a}_{[N]} \sim \mathcal{D}(\boldsymbol{a})} \boldsymbol{w}^0_s - \boldsymbol{a}_i = 0$. The $R'_1$ in above equation is

$$\begin{aligned}
R'_1 =& \frac{4\alpha^3}{K^3} \mathop{\mathbb{E}}_{T \sim \mathcal{D}(T)} \mathrm{tr} \left[ \sum_{i=1}^N \sum_{j=1}^N (4 G^2_i G^2_j - G^3_i G_j) V^2 Q^{2t}_T V G^2_j \right] \\
=& \frac{4\alpha^3}{K^3} \mathop{\mathbb{E}}_{T \sim \mathcal{D}(T)} \mathrm{tr} \left[ \sum_{i=1}^N \sum_{j=1}^N U(4\Sigma^2_j \Sigma^2_i \Sigma^2_j - \Sigma^2_j \Sigma^3_i \Sigma_j) U^\top U \widehat{\Sigma}^2_N Q^{2t}_T \widehat{\Sigma}_N U^\top \right] \\
\leq& \frac{4\alpha^3}{K^3} \mathop{\mathbb{E}}_{T \sim \mathcal{D}(T)} \mathrm{tr} \left[ \sum_{i=1}^N \sum_{j=1}^N (4\Sigma^2_i \Sigma^4_j - \Sigma^3_i \Sigma^3_j) \right] \mathrm{tr} \left[ \widehat{\Sigma}^3_N Q^{2t}_T \right] \\
\leq& \left( \frac{4\alpha^3 d}{N^3 K^3} \right) \left( 1 - \frac{2\eta}{K} \lambda_s \right)^{2t} \sup_{i,j \in [N]} \mathop{\mathbb{E}}_{\Sigma_i, \Sigma_j} \mathrm{tr} \left[ \sum_{i=1}^N \sum_{j=1}^N (4\Sigma^2_i \Sigma^4_j - \Sigma^3_i \Sigma^3_j) \right]
\end{aligned} \tag{G.35}$$

And the eigenvalues of $\Sigma_i$ are bounded in $[\lambda_s, \lambda_x]$ where

$$\begin{aligned}
\mathop{\mathbb{E}}_{\Sigma_i, \Sigma_j} \mathrm{tr} \left[ \sum_{i=1}^N \sum_{j=1}^N (4\Sigma^2_i \Sigma^4_j - \Sigma^3_i \Sigma^3_j) \right] &= \mathrm{tr} \left[ 3N \mathbb{E}\Sigma^6 + (N^2 - N)\mathbb{E}(4\Sigma^2 \Sigma^4 - \Sigma^3 \Sigma^3) \right] \\
&\leq d \left[ 3N\lambda^6_x + (N^2 - N)(4\lambda^6_x - \lambda^6_s) \right] \\
&\leq dN^2(4\lambda^6_x - \lambda^6_s)
\end{aligned} \tag{G.36}$$

In summary,

$$R_1' \le \frac{4\alpha^3 d^2 (4\lambda_x^6 - \lambda_s^6)}{NK^3} \left(1 - \frac{2\eta}{K}\lambda_s\right)^{2t} \tag{G.37}$$

Similar to $R_1'$ let's bound the $R_2$ in (G.10),

$$\begin{aligned}
R_2 &= \alpha^2 \operatorname*{\mathbb{E}}_{\boldsymbol{a}_{[N]} \sim \mathcal{D}(\boldsymbol{a})} \operatorname*{\mathbb{E}}_{T \sim \mathcal{D}(T)} \left\langle Q_T^t V \sum_j^N \frac{4}{K} G_j^2 \left(\boldsymbol{w}_r^0 - \boldsymbol{a}_j\right), Q_T^t V \sum_j^N \frac{4}{K} G_j^2 \left(\boldsymbol{w}_r^0 - \boldsymbol{a}_j\right) \right\rangle \\
&= \frac{16\alpha^2}{K^2} \operatorname*{\mathbb{E}}_{T \sim \mathcal{D}(T)} \operatorname{tr} \left[\sum_{i=1}^N G_i^2 V Q_T^{2t} V G_i^2\right] \le \operatorname*{\mathbb{E}}_{T \sim \mathcal{D}(T)} \operatorname{tr} \left[\sum_{i=1}^N \Sigma_i^4\right] \operatorname{tr} \left[\widehat{\Sigma}_N^2 Q_T^{2t}\right] \\
&= \frac{16\alpha^2 d^2 \lambda_x^4}{\lambda_s^2 N K^2} \left(1 - \frac{2\eta}{K}\lambda_s\right)^{2t}
\end{aligned} \tag{G.38}$$

Thus the final constraint of $\alpha$ will be

$$\begin{aligned}
&\left(1 - \frac{2\eta}{K}\lambda_s\right)^{2t} \left[\frac{4\alpha^3 d^2 (4\lambda_x^6 - \lambda_s^6)}{NK^3} + \frac{16\alpha^2 d^2 \lambda_x^4}{\lambda_s^2 N K^2} - \hat{r}\frac{8\alpha d^2 \widetilde{c}}{K\lambda_x^3}\left(\frac{1}{N} - \frac{1}{N^2}\right)\right] \le 0 \\
&\left(1 - \frac{2\eta}{K}\lambda_s\right)^{2t} \left[\frac{\alpha^2 (4\lambda_x^6 - \lambda_s^6)}{K^2} + \frac{4\alpha \lambda_x^4}{\lambda_s^2 K} - \hat{r}\frac{2\widetilde{c}}{\lambda_x^3}\left(1 - \frac{1}{N^2}\right)\right] \le 0 \\
&\frac{\alpha^2 (4\lambda_x^6 - \lambda_s^6)}{K^2} + \frac{4\alpha \lambda_x^4}{\lambda_s^2 K} - \hat{r}\frac{2\widetilde{c}}{\lambda_x^3}\left(1 - \frac{1}{N}\right) \le 0
\end{aligned} \tag{G.39}$$

where ratio factor $\hat{r} = \left[(1 - 2\eta/K\lambda_x) / (1 - 2\eta/K\lambda_s)\right]^{2t}$. We have the extreme points of $\alpha$

$$\alpha(N) = \frac{-\frac{4\lambda_x^4}{\lambda_s^2 K} \pm \sqrt{\left(\frac{4\lambda_x^4}{\lambda_s^2 K}\right)^2 + 4\frac{(4\lambda_x^6 - \lambda_s^6)}{K^2}\hat{r}\frac{2\widetilde{c}}{\lambda_x^3}\left(1 - \frac{1}{N}\right)}}{2\frac{(4\lambda_x^6 - \lambda_s^6)}{K^2}} \tag{G.40}$$

For $K \in \mathbb{Z}^+ \ge 1$, small $\alpha$, $t = 0$ and large $t$ we have

$$\hat{r} = \left[(1 - 2\eta/K\lambda_x) / (1 - 2\eta/K\lambda_s)\right]^{2t} = \mathcal{O}(1) \tag{G.41}$$

So finally, the $\alpha$ needs to satisfy

$$\alpha(N) \le \alpha(2) = \frac{-2\lambda_x^4 K + K\sqrt{4\lambda_x^8 + 1.5\widetilde{c}\lambda_s^4(4\lambda_x^6 - \lambda_s^6)/\lambda_x^3}}{\lambda_s^2(4\lambda_x^6 - \lambda_s^6)} \tag{G.42}$$

$\square$

# H EXPERIMENTS

## H.1 PRACTICAL FORM OF THEOREM 1

For practical use, we show a numerical form to estimate $\alpha^*$ for the case where number of tasks is finite and training data $\boldsymbol{u} \in \mathbb{R}^{k_1}$ is different from validation data $\boldsymbol{t} \in \mathbb{R}^{k_2}$. The corresponding feature matrices are $\Phi(\boldsymbol{u})$ and $\Phi(\boldsymbol{t})$. Let's derive corollary of Theorem 1 for realistic meta-learning setting.

**Corollary 2.** *If training feature $\Phi(\boldsymbol{u}) \in \mathbb{R}^{k_1 \times d}$ is different from validation feature $\Phi(\boldsymbol{t}) \in \mathbb{R}^{k_2 \times d}$ for every task, then*

$$\alpha_{lim}^*(k_1) = \frac{k_1 \mathbb{E}_{\boldsymbol{x}} tr[\Phi(\boldsymbol{u})^\top \Phi(\boldsymbol{u})(\Phi(\boldsymbol{t})^\top \Phi(\boldsymbol{t})]}{2\mathbb{E}_{\boldsymbol{x}} tr[\Phi(\boldsymbol{u})^\top \Phi(\boldsymbol{u})(\Phi(\boldsymbol{t})^\top \Phi(\boldsymbol{t})\Phi(\boldsymbol{u})^\top \Phi(\boldsymbol{u})]}. \tag{H.1}$$

*Proof.* Similar to proof of Theorem 1, we have

$$\boldsymbol{w}_m \left(\{\boldsymbol{x}_i, \boldsymbol{a}_i\}_{i \in [N]}, N, k_1, k_2, \alpha\right) = \left(\sum_{i=1}^N \widehat{C}_i(\alpha)^\top \widehat{C}_i(\alpha)\right)^{-1} \left(\sum_{i=1}^N \widehat{C}_i(\alpha)^\top \widehat{C}_i(\alpha)\boldsymbol{a}_i\right) \tag{H.2}$$

where $\widehat{C}_i(\alpha) = \Phi(\boldsymbol{t}_i)(I - \frac{2\alpha}{k_1}\Phi(\boldsymbol{u}_i)^\top\Phi(\boldsymbol{u}_i)) \in \mathbb{R}^{k_2 \times d}$.

The corresponding average population risk will be

$$
\begin{aligned}
\bar{\mathcal{L}}_m(N, k_1, k_2, \alpha) &= \mathbb{E}_{\boldsymbol{a}_1,\ldots,\boldsymbol{a}_N \sim \mathcal{D}(\boldsymbol{a})} \mathcal{L}_m(\boldsymbol{w}_m, \alpha, k_1, k_2) \\
&= \mathop{\mathbb{E}}_{\boldsymbol{a},\{\boldsymbol{a}_i\}_{i=1}^N \sim \mathcal{D}(\boldsymbol{a})} \mathop{\mathbb{E}}_{\boldsymbol{x} \sim \mathcal{D}(\boldsymbol{x})} \left\| \widehat{C}(\alpha) \left( \boldsymbol{w}_m(\{\boldsymbol{x}_i, \boldsymbol{a}_i\}_{i \in [N]}, N, k_1, k_2, \alpha) - \boldsymbol{a} \right) \right\|^2
\end{aligned}
\tag{H.3}
$$

Then we can define similar approximation function $L_m^{apx}(\alpha)$ as (B.6) where

$$
L_m^{apx}(\alpha) \triangleq \mathbb{E}_{\boldsymbol{x} \sim \mathcal{D}(\boldsymbol{x})} \operatorname{tr}\left[ \widehat{C}(\alpha)^\top \widehat{C}(\alpha) \right]
\tag{H.4}
$$

And with same bound for $\|\Phi(\boldsymbol{u}_i)\|, \|\Phi(\boldsymbol{t}_i)\| \in [c_1, c_2]$, we have

$$
\Gamma - \lim_{N \to \infty} \bar{\mathcal{L}}_m(k_1, k_2, N, \alpha) = \hat{L}_m^{apx}(\alpha)
\tag{H.5}
$$

With Gamma convergence lemma, we can get the final estimation $\alpha_{lim}^*$ in (H.1).

$\square$

In experiments, we use the following numerical form

$$
\alpha_{lim}^* = \frac{k_1 \sum_{i=1}^N tr[\Phi(\boldsymbol{u}_i)^\top \Phi(\boldsymbol{u}_i)(\Phi(\boldsymbol{t}_i)^\top \Phi(\boldsymbol{t}_i)]}{2 \sum_{j=1}^N tr[\Phi(\boldsymbol{u}_j)^\top \Phi(\boldsymbol{u}_j)(\Phi(\boldsymbol{t}_j)^\top \Phi(\boldsymbol{t}_j)\Phi(\boldsymbol{u}_j)^\top \Phi(\boldsymbol{u}_j)]}.
\tag{H.6}
$$

to evaluate our estimation.

## H.2 OVERPARAMETERIZED SETTING

Let's consider overparameterized setting. Thus, we have feature for each task ($K < d$), $\Psi = [\psi(x_1), \ldots, \psi(x_K)]^\top \in \mathbb{R}^{K \times d}$. Correspondingly, empirical objective of ERM is given by

$$
\hat{\mathcal{L}}_r(\boldsymbol{w}) = \frac{1}{NK} \sum_{i=1}^N \|\Psi_i \boldsymbol{w} - \boldsymbol{y}_i\|.
$$

Assume meta-initialization is the mean of all task optima that $\bar{\boldsymbol{a}} = \text{mean}(\boldsymbol{a}_1, \ldots, \boldsymbol{a}_N)$. Then concatenate all task features we have

$$
\Psi_{all} = \begin{pmatrix} - & \Psi_{1(1)} & - \\ & \cdots & \\ - & \Psi_{N(K)} & - \end{pmatrix}, \Psi^{all} \in \mathbb{R}^{NK \times d}
\tag{H.7}
$$

So MAML objective is given by

$$
\begin{aligned}
\hat{\mathcal{L}}_m(\boldsymbol{w}, \alpha, N, K) &= \frac{1}{NK} \|\Psi_{all}\boldsymbol{w}' - \boldsymbol{y}_{all}\|^2 \\
&= \frac{1}{NK} \|C_{all}(\alpha)\boldsymbol{w} - \bar{\boldsymbol{a}}\|^2
\end{aligned}
\tag{H.8}
$$

where $\boldsymbol{w}' = \left[\boldsymbol{w} - 2\alpha/(NK)\Psi_{all}\left(\Psi_{all}^\top \boldsymbol{w} - \Psi_{all}^\top \bar{\boldsymbol{a}}\right)\right]$ is adapted parameters and $C_{all}(\alpha) = \Psi_{all}(I - (2\alpha/NK)\Psi_{all}^\top \Psi_{all})$. The minimum norm solution is

$$
\boldsymbol{w}_m(\ldots, N, K, \alpha) = C_{all}(\alpha)^\top \left(C_{all}(\alpha)C_{all}(\alpha)^\top\right)^{-1} C_{all}(\alpha)\bar{\boldsymbol{a}}
\tag{H.9}
$$

Note that, in overparameterized setting, Theorem 1 will not perfectly give the precise form for $\alpha^*$. But the technique can be easily extend to large $d$ setting where

$$
\begin{aligned}
\bar{\mathcal{L}}_m(\alpha, N, K) &= \mathbb{E}_{\boldsymbol{a}, \bar{\boldsymbol{a}}, \boldsymbol{x}} \|C(\alpha)(\boldsymbol{w}_m - \boldsymbol{a})\|^2 \\
&= \mathbb{E}_{\boldsymbol{a}, \bar{\boldsymbol{a}}, \boldsymbol{x}} \left[ \boldsymbol{w}_m^\top C(\alpha)^\top C(\alpha)\boldsymbol{w}_m + \boldsymbol{a}^\top C(\alpha)^\top C(\alpha)\boldsymbol{a} \right] \\
&= \mathbb{E}_{\boldsymbol{x}} \operatorname{tr}[C_{all}(\alpha)^\top C_{all}^{gram} C_{all}(\alpha)C(\alpha)^\top C(\alpha)C_{all}(\alpha)^\top C_{all}^{gram} C_{all}(\alpha)] \\
&\quad + \mathbb{E}_{\boldsymbol{x}} \operatorname{tr}[C(\alpha)^\top C(\alpha)]
\end{aligned}
\tag{H.10}
$$

where $C_{all}^{gram} = (C_{all}(\alpha)C_{all}(\alpha)^\top)^{-1}$.

By Proposition 2, the $\bar{\mathcal{L}}_m(\alpha, N, K)$ will be upper bounded where

$$
\begin{aligned}
\bar{\mathcal{L}}_m(\alpha, N, K) &= \mathbb{E}_{\boldsymbol{x}} \operatorname{tr}[C_{all}^{gram} C_{all}(\alpha) C(\alpha)^\top C(\alpha) C_{all}(\alpha)^\top] + \mathbb{E}_{\boldsymbol{x}} \operatorname{tr}[C(\alpha)^\top C(\alpha)] \\
&= \mathbb{E}_{\boldsymbol{x}} \operatorname{tr}[C_{all}(\alpha)^\top C_{all}^{gram} C_{all}(\alpha) C(\alpha)^\top C(\alpha)] + \mathbb{E}_{\boldsymbol{x}} \operatorname{tr}[C(\alpha)^\top C(\alpha)] \\
&\leq \mathbb{E}_{\boldsymbol{x}} \operatorname{tr}[C_{all}(\alpha)^\top C_{all}^{gram} C_{all}(\alpha)] \mathbb{E}_{\boldsymbol{x}} \operatorname{tr}[C(\alpha)^\top C(\alpha)] + \mathbb{E}_{\boldsymbol{x}} \operatorname{tr}[C(\alpha)^\top C(\alpha)] \\
&= 2\mathbb{E}_{\boldsymbol{x}} \operatorname{tr}(C(\alpha)^\top C(\alpha))
\end{aligned}
\tag{H.11}
$$

Hence, minimizing the upper bound also tells us how to select $\alpha^*$. In another word, we are seeking an estimation $\alpha_{est}^*$ nearly minimize the upper bound.

$$
\alpha_{est}^* = \arg\min_{\alpha} \mathbb{E}_{\boldsymbol{x}} \operatorname{tr}(C(\alpha)^\top C(\alpha))
\tag{H.12}
$$

and the $C(\alpha)^\top C(\alpha)$ is a covariance matrix where

$$
C(\alpha)^\top C(\alpha) = \Psi^\top \Psi - \frac{4\alpha}{NK}(\Psi^\top \Psi)^2 + \frac{4\alpha^2}{N^2 K^2}(\Psi^\top \Psi)^3
\tag{H.13}
$$

We can derive that

$$
\alpha_{est}^* = \frac{NK \mathbb{E}_{\boldsymbol{x}} \operatorname{tr}(\Psi^\top \Psi)^2}{2\mathbb{E}_{\boldsymbol{x}} \operatorname{tr}(\Psi^\top \Psi)^3}
\tag{H.14}
$$

Since $d$ is large, its computational cost is high. Here we assume second moment of all elements of $\Psi^\top \Psi$ are $\tilde{\sigma}^2$, which means $\tilde{\sigma}$ is the **variance of all elements of features**. Finally we have

$$
\boxed{\alpha_{est}^* = \frac{1}{2NK\tilde{\sigma}^2}}
\tag{H.15}
$$

Let's take the Neural Tangent Kernel (NTK) (Jacot et al., 2018) for example,

$$
f(\boldsymbol{w}, \boldsymbol{x}) = f(\boldsymbol{w}^{init}, \boldsymbol{x}) + \nabla f(\boldsymbol{w}^{init}, \boldsymbol{x})^\top (\boldsymbol{w} - \boldsymbol{w}^{init}).
\tag{H.16}
$$

Then we have neural tangent feature

$$
\begin{aligned}
\Psi_i^\top \boldsymbol{w} &= \nabla f(\boldsymbol{w}^{init}, X_i)^\top (\boldsymbol{w} - \boldsymbol{w}^{init}) \\
\Rightarrow \Psi_i &\approx f(\boldsymbol{w}^{init}, X_i)
\end{aligned}
\tag{H.17}
$$

Next, we stack all the features to $\Psi_{all}$ to compute the variance e.g. recall $\sigma = \Psi_{all}.\text{std}()$. After that, we can compute the estimation $\alpha_{est}^*$ using (H.15).

## H.3 EXPERIMENTAL SETUP

**Estimation of $\alpha^*$, underparameterized model**  We set hyperparameters dimension $d = 20$, number of training/validation samples per task $K = 50$, number of tasks $N = 5000$. Each $x$ is i.i.d sampled from a distribution $\mathcal{U}(-5, 5)$ while each $\boldsymbol{a}$ is i.i.d sampled from high dimension Gaussian distribution $\mathcal{N}(\boldsymbol{0}, 3I)$. Then computing the Ordinary Least Square (OLS) solution with different $\alpha$, we can show the training loss landscape in terms of $\alpha$. The true $\alpha^*(N) = \arg\min_{\alpha} \min_{\boldsymbol{w}} \mathcal{L}(\alpha, 5000, 50, \boldsymbol{w})$ is the minimizer of the training loss. Our estimation $\alpha_{lim}^*$ described in Theorem 1 is evaluated by comparing the error to true $\alpha^*$.

**Estimation of $\alpha^*$, overparameterized model**  We perform the nonlinear regression on two different models. The first is quadratic regression using neural tangent feature (see H.2). All hyperparameters are set to be same as (Bernacchia, 2021) where

$$
\boldsymbol{w} \sim \mathcal{N}\left(\boldsymbol{w}_0, \frac{\nu^2}{p} I_p\right) \quad b \sim \mathcal{N}(0, \sigma^2) \quad \mathbf{x} \sim \mathcal{N}(0, I_p) \quad y \mid \mathbf{x}, \boldsymbol{w}, b \sim \mathcal{N}\left((\mathbf{x}^T \boldsymbol{w} + b)^2, \sigma^2\right)
$$

But to guarantee the overparameterization, we set hidden size with $10,000$, which means the total dimension will be $30,001$. Then we perform quadratic function regression with ranging $\alpha$ value to see the test loss. After that, we can evaluate the how accurate our estimation using (H.15) is by

comparing to optimum of test loss. Second experiment is sine function regression using 3-layer MLPs activated with ReLU. The data and labels are generated from a stochastic function

$$y = a\sin(x+b), a, b \sim \mathcal{U}(0, \pi), x \sim \mathcal{U}(-5, 5).$$

To get a good representation, we pre-train the model with ERM loss and then freeze the first two layers as the feature extractor. Then we use the output from second layer as the random feature to train last layer on $1,000$ training tasks. At the same time, $\alpha_{lim}^*$ can be computed from the features of $1,000$ training tasks. Then last layer trained with differnet $\alpha$ will be evaluated on $10,000$ test tasks.

**Fast adaptation distance** We run experiments with random matrices. For each task, the data are i.i.d drawn for the prescribed distributions which represent three common types, Uniform $\mathcal{U}(-5, 5)$, Gaussian $\mathcal{N}(0, 2)$ and Exponential $Exp(1)$. Specifically, each entry in random matrix $X \in \mathbb{R}^{K \times d}, (K = 50, d = 20)$ is sampled variable from a distribution $X_{(ij)} \sim \mathcal{D}(x)$. The feature map is an identity map $\Phi(X) = X$. First we sample 5000 training tasks to compute the closed-form meta-initializations for ERM and MAML with a small $\alpha$ ($10^{-4}$). Then we perform $t$-step adaptation on 5000 test tasks and compare the fitting losses and the *Average Distance under Fast Adaptation* $\mathcal{F}_t(w_m), \mathcal{F}_t(w_r)$. The learning rate $\eta$ in fast adaptation evaluation is $10^{-5}$.

**Estimation of $\alpha^*$, deep learning** In our experiments, we valid our estimation of $\alpha^*$ on sine regression and few-shot classification.

- For deep regression, we follow the (Finn et al., 2017) to perform sine regression with 3-layer MLP whose hidden size is $40$. Then each task is an instance in stochastic function $y = a\sin(x+b), a, b \sim \mathcal{U}(0, \pi)$ while the training set is 10 i.i.d sampled data pair from the corresponding sine function and test set is consists of another sampled 256 points. During test, we sample $10,000$ tasks to evaluate the learned model.

- For deep classification, we follow the Omniglot experiments in (Raghu et al., 2020). Here, we adopt the online estimation scheme to compute the $\alpha^*$ for the adaptation learning rate of last layer. To this end, we apply $\alpha^* = 1/(2 \times N_{way} \times N_{shot} \times \hat{\sigma}^2)$ where $\hat{\sigma}^2$ is mean of covariance of the normalized feature $F^\top F$, $F = (F_1/\|F_1\|, ..., F_{N_w ay}/\|F_{N_w ay}\|)$. Then $\alpha_{buffer}^*$ in the buffer is online updated using $\alpha_{buffer}^* \leftarrow 0.9\alpha_{buffer}^* + 0.1\alpha^*$.

## H.4 ADDITIONAL RESULT

### H.4.1 OPTIMIZATION BEHAVIOR

We add more illustrative experiments on visualizing the trajectory of global minima of MAML. We consider normally distributed task optima with centralized data and uncentralized data. As shown in 7, the MAML minimum still try to balance the distances to different task optima. But the situation is more complex in uncentralized data (second row). However, $\alpha$ always minimize the geometric distance at beginning where the shape of mean distance function appears to be convex. This has confirmed our Theorem 2 where small $\alpha$ always lead to a shorter mean distance to different task optima than ERM algorithm.

### H.4.2 ESTIMATION OF $\alpha^*$ ON BASIS FUNCTION FEATURE

Firstly, we used the random matrix for task $i$ as the feature matrix, $\Phi_i \in \mathbb{R}^{K \times d}$. All elements of $\Phi_i$ are i.i.d sampled from $\mathcal{U}(-5, 5)$. Secondly, we created the random features using Gaussian basis function. Gaussian basis function $\Phi(X)_{(ij)} = \exp(-(X_{(ij)} - \mu_j)^2/2\sigma_i^2)$ is a function whose value depends only on the distance between the input and some fixed point. Thirdly, we used polynomial feature $\Phi(X)_{(i,:)} = (c_0, \cdots, c_n x_{(i)}^{d-1})$ which is based on Taylor series. With $N$ tasks, we compute the one-step adaptation loss. Optimal learning rate minimizing the loss is denoted by $\alpha^*(N)$. The error gap between true optimum and estimation $|\alpha^*(N) - \alpha_{lim}^*|$ with three random features are shown in Figure 8 (a), (b) & (c) respectively. To reduce random errors, there was an average of 10 sampling trials, shown as the solid lines. The shadow area represents standard deviation. As the number of tasks $N$ increasing, the estimation error will shrink down to zero and its uncertainty reduces as well. So our estimation is reliable and accurate when number of tasks becomes large.

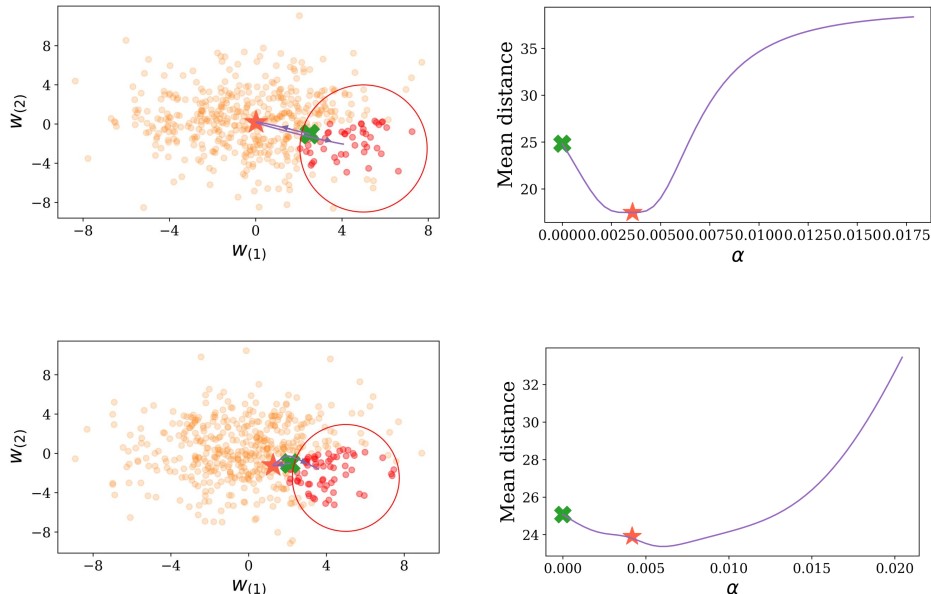

Figure 7: Additional results for optimization behavior with normally distributed task optima. Left column: Visualization of trajectory of MAML solution. Orange dots are task optima $\{\boldsymbol{a}\}_{[N]}$ of sampled tasks, where location of $\boldsymbol{a}_i$ is decided by its entries. Red dot highlighted in circle is new coming task. Green cross is $\boldsymbol{w}_r, (\alpha = 0)$ while the purple trajectory is generated as $\alpha$ increasing. Red star is $\boldsymbol{w}_m(\alpha^*_{lim}, ...)$. Right column: Average euclidean distance of $\boldsymbol{w}_m(\alpha, ...)$ and $\{\boldsymbol{a}\}_{[N]}$ and corresponding points in left figure. First row: centralized data $x \sim \mathcal{N}(0,2), \boldsymbol{a} \sim \mathcal{N}(\boldsymbol{0}, 3I)$. Second row: uncentralized data $x \sim \mathcal{U}(0,5), \boldsymbol{a} \sim \mathcal{N}(\boldsymbol{0}, 3I)$. Best viewed in colors.

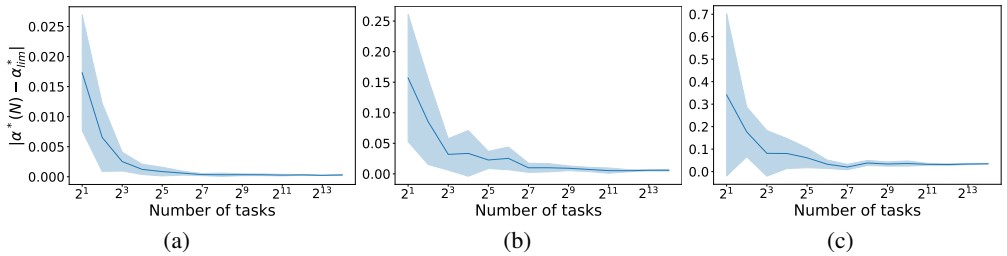

Figure 8: Estimation error $|\alpha^*(N) - \alpha^*_{lim}|$ along task number $N$ increasing ($K$ is fixed). The blue line in the shadow is mean of the error. The shadow area is the standard deviation of the errors. (a) Random matrices (b) Gaussian basis function (c) Polynomial basis function.

We use Gaussian basis function as the random feature and conduct the experiments on different types of distribution to evaluate our estimation quality. Then, we use uniform/normal distribution with zero mean $\mathcal{U}, \mathcal{N}$ as the stereotype of central symmetric distribution. In experiments, we set $d = 10, K = 15, N = 3000$ and the parameters in Gaussian function depends on the range of data. As shown in the Figure 9, the estimation $\alpha^*_{lim}$ is close to true optimum, $\alpha^*$ in four different cases: (a) data is sampled from a central uniform distribution $\mathcal{U}(-5,5)$, task optima are sampled from a central normal distribution $\mathcal{N}(\boldsymbol{0}, 3^2 I)$; (b) data is sampled from a non-central normal distribution $\mathcal{N}(0, 2^2 I)$, task optima are sampled from a central normal distribution $\mathcal{N}(\boldsymbol{0}, 3^2 I)$; (c) data is sampled from a central uniform distribution $\mathcal{U}(-5,5)$, task optima are sampled from a non-central normal distribution $\mathcal{N}(\boldsymbol{5}, I)$; (d) data is sampled from a non-central Chi-Sqaure distribution $\chi^2(7)$, task optima are sampled from a imbalanced Zipf distribution $P(x = k) = \frac{1}{\zeta(s)} k^{-s}$ where $\zeta(s)$ is the

| $\alpha$ | 0.001 | 0.005 | 0.01 | 0.05 | 0.08 | **0.1** | 0.15 | 0.2 | 0.3 | 0.4 | 0.5 |
|---|---|---|---|---|---|---|---|---|---|---|---|
| Pre MSE | 829 | 822 | 822 | 827 | 829 | 827 | 825 | 821 | 826 | 825 | 826 |
| Post MSE | 829 | 820 | 817 | 807 | 800 | **797** | 805 | 820 | 861 | 955 | 989 |

Table 2: Test loss of one-step sinusoid regression with neural network feature. First row is the discrete test values of $\alpha$, second row is the Mean Square Error(MSE) loss before adaptation and third row is the loss after adaptation. All loss values are digits after the decimal point

Riemann Zeta function. Note that results in (c) and (d) are beyond our Assumption 1. So our theorem can be extended to more general scenarios.

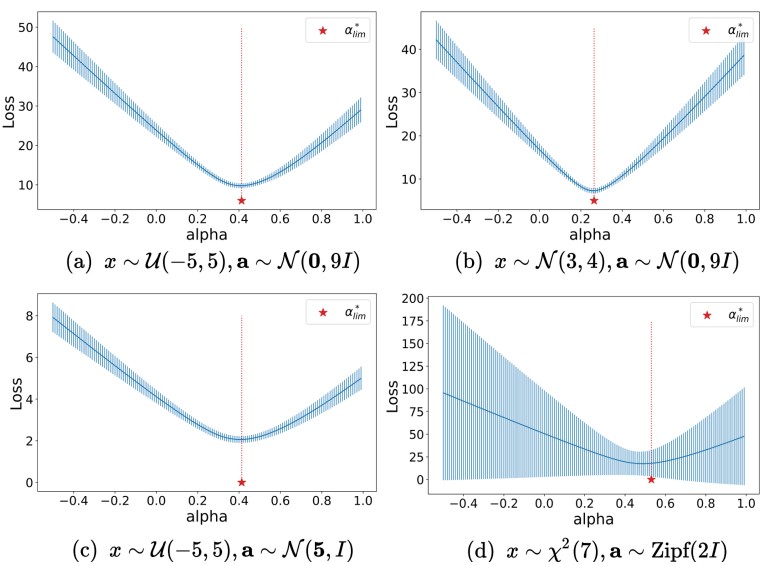

Figure 9: Evaluate estimation $\alpha_{lim}^*$ on different types of distributions. (a) Central data distribution and central task optima distribution. (b) Non-central data distribution and central task optima distribution. (c) Central data distribution and Non-central task optima distribution. (d) Non-symmetric non-central distributions for data and task optima.

### H.4.3   ESTIMATION OF $\alpha^*$ ON NEURAL NETWORK FEATURE

We used neural network based feature to verify our theorem in underparameterized (original model size in Finn et al. (2017)) and overparameterized setting ($NK < d$). In former setting, we used 3-layer Multilayer Perceptron (MLP) activated with ReLU for sine functions family regression where each task is to regress an instance in stochastic function $y = a \sin(x + b), a, b \sim \mathcal{U}(0, \pi)$. We used ERM to train and freeze the first 2 layers (as feature extractor) and then only fine-tune last layer with MAML. Compute $\alpha_{lim}^*$ through features of sampled training tasks we got $\alpha_{lim}^* = 0.10319$. As shown in Table. 2, the optimal $\alpha$ of lowest MSE after adapting is $0.1$ which is nearest discrete value in table to $\alpha_{lim}^*$.

### H.4.4   HEURISTIC ESTIMATION RANGE FOR DEEP LEARNING

To make it practical for deep learning, we give the heuristic estimation range where $\alpha^*$ it might be based on our theorem. Previously, we show (3.3) for underparameterized model ($K > d$) and (H.15) for overparameterized model ($NK \ll d$). Besides, the trace term for covariance matrix in underparameterized setting can be simplied by $Kd\tilde{\sigma}^2$ where $\tilde{\sigma}^2$ is the covariance of the feature (second moment). So here, the heuristic estimation by merging these two settings, where it derived as

$$\alpha_{lim}^* = \frac{1}{2 \min(NK, d)\tilde{\sigma}^2} \tag{H.18}$$

where $N, K$ are number of training task and its training sample size, $d$ is model size. Next, we show a simple way to estimate the range of $\tilde{\sigma}^2$. In general, each element of feature with probability 1 will fall into the $[0, 1]$. Then, given $\tilde{n}$ observations, we have (Popoviciu's inequality on covariance (Sharma et al., 2010))

$$\frac{1}{2\tilde{n}} \leq \tilde{\sigma}^2 \leq \frac{1}{4} \tag{H.19}$$

*Proof.* Assume with probability 1, each element $x$ of $\Phi(X)$ has $x \in [m, M]$.

Define a function $f$ in terms of random variable $x$ by

$$f(t) = \mathbb{E}\left[(x - t)^2\right] \tag{H.20}$$

Computing the derivative $f'$, and solving the minimum

$$f'(t) = -2\mathbb{E}[x] + 2t = 0 \Rightarrow f(\mathbb{E}[x]) = \min_{t \in \mathbb{R}} f(t) \tag{H.21}$$

So we have the covariance has following upper bound

$$\tilde{\sigma}^2 = f(\mathbb{E}[x]) \leq f\left(\frac{M + m}{2}\right) \tag{H.22}$$

where

$$f\left(\frac{M + m}{2}\right) = \mathbb{E}\left[\left(x - \frac{M + m}{2}\right)^2\right] = \frac{1}{4}\mathbb{E}\left[((x - m) + (x - M))^2\right] = \frac{(M - m)^2}{4} \tag{H.23}$$

Thus for $m = 0, M = 1$, we have

$$\tilde{\sigma}^2 \leq \frac{(1 - 0)^2}{4} = \frac{1}{4} \tag{H.24}$$

for an independent sample of $\tilde{n}$ observations from a bounded probability distribution, the von Szokefalvi Nagy inequality shows that

$$\tilde{\sigma}^2 \geq \frac{(M - m)^2}{2\tilde{n}} = \frac{1}{2\tilde{n}} \tag{H.25}$$

$\square$

Here, we conduct following deep regression experiments to evaluate our heuristic estimation. We plot the estimated range of $\alpha^*$ given by our bounds – the red area between the two star-lines in Figure 10. Then we perform quadratic regression with 2-layer neural network follow the hyperparameter in (Bernacchia, 2021). From Figure 10(a), we can see, the optimal $\alpha$ for this task is positive and our estimated range includes suboptimal points. Follow the setting of (Finn et al., 2017) (All hyperparameters are same), we use 3-layer NN with hidden size 40 to test sine regression tasks. As shown in the Figure 10(b), our estimated range includes the optimal $\alpha$ and other good $\alpha$.

## H.5 RELATION TO NEGATIVE LEARNING RATE

As we mentioned before, (Bernacchia, 2021) show negative learning rate minimizing the test loss of MAML. In this section, we compare their results with ours. Specifically, we follow the setting of underparameterized experiment (Bernacchia, 2021) where the defined hyperparameters are set to be same, $n_t = 5, n_v = 25, n_r = 10, m = 40, p = 30, \sigma = 0.2, \nu = 0.2$. Parameters are sampled from following distributions

$$\mathbf{w} \sim \mathcal{N}\left(\mathbf{w}_0, \frac{\nu^2}{p} I_p\right) \quad \mathbf{x} \sim \mathcal{N}\left(0, I_p\right) \quad y \mid \mathbf{x}, \mathbf{w} \sim \mathcal{N}\left(\mathbf{x}^T \mathbf{w}, \sigma^2\right)$$

We conduct experiments on numerical fitting loss on meta-learning instead of the closed-forms $\bar{\mathcal{L}}^{test}$ in theorems (Bernacchia, 2021)[2]. As we can see from the Figure 11, (Bernacchia, 2021) only give the result on special case where $\alpha_r = 0.2$ is fixed. However, this strategy highly depends on the selection of $\alpha_r$ that may not achieve the minimum of meta-learning loss.

---

[2]Test losses are computed on standard meta-learning regression

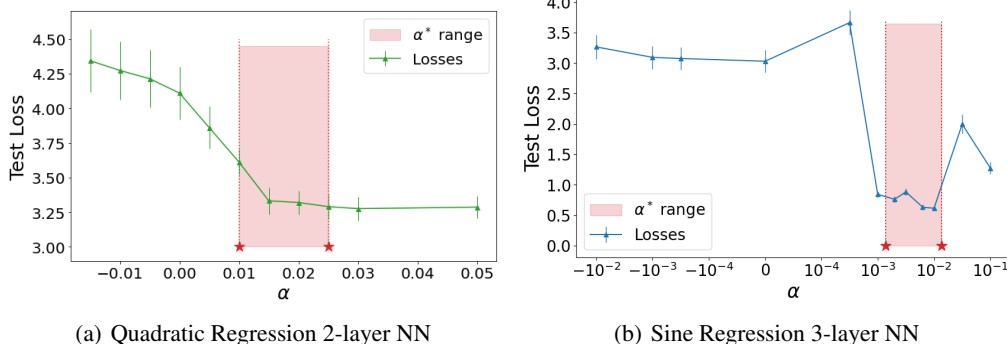

(a) Quadratic Regression 2-layer NN  (b) Sine Regression 3-layer NN

Figure 10: Heuristic estimation of $\alpha^*$ range of deep learning. (a) Quadratic regression on 2-layer neural network. (b) Sine regression on 3-layer neural network.

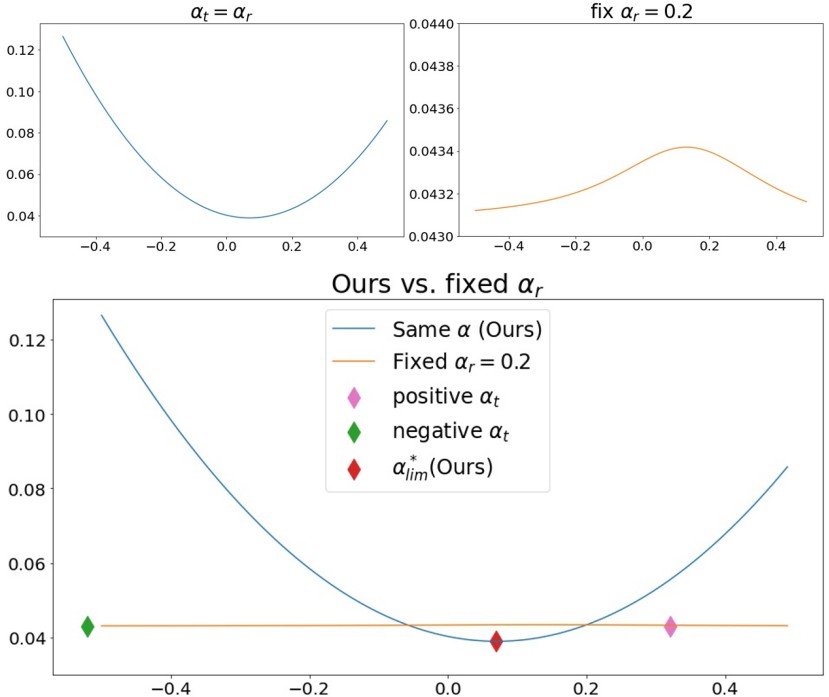

Figure 11: Comparison of our estimation and (Bernacchia, 2021) on underparameterized mixed linear regression. $X$-axis is the discrete values of $\alpha_t$ and $Y$-axis is the test loss of MAML. First row, test loss with respect to $\alpha_t$ while the left one shows same $\alpha$ for meta-training and meta-testing and right one is fixed $\alpha_t = 0.2$ strategy. Second row, comparison of test loss of different strategies and the suggested range of minimizers given by their paper (pink and green diamonds) and our estimation (red diamond). Best viewed in color.

In addition, we run deep learning experiments to demonstrate that optimal learning rate $\alpha$ is positive. All hyperparameters and generating process are set to be same as the non-linear regression experiments in (Bernacchia, 2021). Furthermore, we train the 2-layer neural network to regress quadratic functions with 5 adaptation steps and evaluate models on same 10 folds with each fold consists of 1000 test tasks. The results (with error bar) are shown in the Figure 12. As we can see, the optimal learning rate for both strategies are positive.

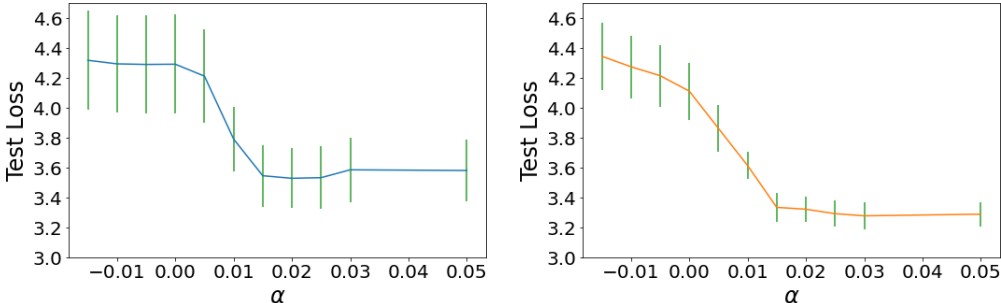

Figure 12: Test losses with reference to adaptation learning rate in meta-training of deep quadratic regression in (Bernacchia, 2021). $X$-axis is the discrete values of $\alpha_t$ and $Y$-axis is the test loss of MAML. Left: test loss with fixed $\alpha_t = 0.01$ and varying $\alpha_r$. Right: test loss with same $\alpha_t$ and $\alpha_r$

