# OpenReview forum: "Unraveling Model-Agnostic Meta-Learning via The Adaptation Learning Rate"
_ICLR.cc/2022/Conference — ICLR 2022 Poster_

### Official Review · Reviewer_9gLM · 2021-11-02

**Correctness:** 3
**Technical Novelty And Significance:** 3
**Empirical Novelty And Significance:** Not applicable
**Recommendation:** 6
**Confidence:** 1

**Main Review:**

## Strengths

The questions investigated by the paper are interesting and are well-positioned in relation to prior work. The contributions appear to develop on closely related studies such as Bernacchia et al, ICLR 2021. Supporting the theoretical results with empirical studies also provides some intuition and further insights.

 ## Weaknesses

I am not familiar enough with the methods and related work to gauge the significance of the theoretical results. However, a couple points on clarity that came up when reading:

- For clarity, the estimator of optimal LR in Eq 145 could be included in the main paper, since this is used in the experimental sections.
- Fig 5 was a little hard to understand as a standalone -- could thetwo differences in loss/solution distance be defined in the caption or in the figure itself to help with this?


**Summary Of The Paper:**

This paper considers the popular MAML algorithm, and provide more theoretical understanding of (1) the optimal inner loop learning rate and (2) how the MAML initialization compares to that learned by empirical risk minimization. They consider the mixed linear regression setting when analyzing these questions.

Firstly, to estimate the optimal inner loop learning rate (adaptation rate), the authors pose this as a problem of minimizing MAML's population risk, and develop an estimator for this optimal learning rate.

They then aim to understand how the initialization learned by MAML compares to that from empirical risk minimization (ERM), finding that on average that MAML's initialization is closer to the optimal per-task initializations than the one learned by ERM.

The two main theoretical results (optimal adaptation and comparing the meta-initialization of MAML to that of ERM) are validated in synthetic experiments with different data generating distributions.

**Summary Of The Review:**

Overall this work is well-motivated and appears to provide new insight about the inner loop adaptation of MAML and the significance of the meta-initialization learned by MAML, so I would consider recommending acceptance. However, as stated, I am not familiar with related theoretical works so have selected my confidence score in accordance with that.

---

> ### Author Response · Authors · 2021-11-16
> **Responce to Reviewer 9gLM**
>
> We thank the reviewer for the comments and suggestions. Below, we answer each of the reviewer's questions and summarize the corresponding changes (in red) in the paper.
>
> ### [Q1: Estimator of optimal LR in Eq 145 could be included in the main paper]
>
> We included this equation in Section 5.1 (line 5).
>
> ### [Q2: Can loss/solution distance be defined in the caption?]
>
> We revised the caption in figures (please refer to Figure 5).

---

### Official Review · Reviewer_iGwz · 2021-11-02

**Correctness:** 3
**Technical Novelty And Significance:** 3
**Empirical Novelty And Significance:** Not applicable
**Recommendation:** 6
**Confidence:** 2

**Main Review:**

Strengths
- The paper was largely well-written.
- The theoretical results were validated by simulations (albeit toy), even when the assumptions were slightly violated.

Weaknesses
- Missing loosely related works in meta-learning as hierarchical Bayes [A] and Bayesian linear regression with learned priors [B].
- The large gap between the setting studied in the paper and meta-learning in practice limits my confidence in how impactful this work will be. Even works that limit adaptation to be optimizing a linear model (e.g. Raghu et al. (2020)) atttribute the success of meta-learning to representation learning; this work does not consider representation learning.

Questions
- Can we not think of MAML in this mixed linear regression setting as learning the mean of a Gaussian prior over $w$, and the choice of $\alpha$ as implicitly specifying the variance of the prior?
- Why is ERM a valid algorithm to consider for mixed linear regression? How can a single $w$ possibly fit multiple tasks, e.g. consider two tasks with the same $\Phi(X)$ but different $a$? Why not consider task-specific ERM, in which a different $w$ is fit to each task?
- Is there any way the estimator for $\alpha^*$ can be leveraged in a typical meta-learning experiment? E.g. with ANIL, using online estimation of the representation statistics?
- Given the emphasis in modern empirical deep learning on activation normalization, is the derived relationship between regressor variance and $\alpha$ practically significant?

References
- [A] Grant et al., Recasting gradient-based meta-learning as hierarchical Bayes, 2018, https://arxiv.org/pdf/1801.08930.pdf.
- [B] Harrison et al., Meta-Learning Priors for Efficient Online Bayesian Regression, 2018, https://arxiv.org/abs/1807.08912

**Summary Of The Paper:**

This work analyzes the model-agnostic meta-learning (MAML) algorithm in the linear regression setting. It first derives an estimator for the optimal inner-loop learning rate for one inner-loop update. The authors analyze the statistical properties of the optimal inner-loop learning rate in order to try to obtain insights applicable to settings that go beyond restrictive assumptions in the theoretical analysis, finding an inverse relationship between the estimator and regressor variance. As a second contribution, the authors analyze the geometric properties of MAML and characterize the distance between adapted parameters and true task optima, giving an expression for the deviation of this distance in expectation between MAML and empirical risk minimization (ERM). The authors validate their optimal learning rate estimator with simulations across a variety of feature mappings, and also corroborate the estimator's inverse relationship with the regressor variance.

**Summary Of The Review:**

The theoretical contributions of this work are somewhat interesting but are also of dubious practical relevance.

---

> ### Author Response · Authors · 2021-11-16
> **Responce to Reviewer iGwz (1/2)**
>
> We thank the reviewer for the comments and suggestions. Below, we answer each of the reviewer's questions and summarize the corresponding changes (in red) in the paper.
>
> ### [W1: Missing loosely related works]
>
> We added these missing references in the Related Work (paragraph 1 line 3).
> As a comparison, the goals of our analysis and the obtained results are different from theirs.
> While these methods regard meta-learning as maximum a posteriori (MAP) estimate and aim to improve MAML performance (e.g. via Hessian estimation),
> we aim to understand the simplest MAML algorithm in linear models and analyze the effect of the inner loop learning rate on learning dynamics.
>
> **[Summary of Revision to Paper]**
>
> Missing related works are added in Related work (paragraph 1 line 3).
>
> ### [W2: Large gap between the setting studied in the paper and meta-learning in practice]
>
> - Theorem 1: We choose the linear setting as a starting point of rigorous analysis of the effect of $\alpha$ on MAML. While this setting is limited from a practical point of view, we show that the analysis reveals non-trivial results,
> such as the relationship between optimal $\alpha^*$ and data statistics.
> Furthermore, we empirically demonstrate that such insights can be carried over to the nonlinear case to guide hyper-parameter selection, (Omniglot in Section 5.1 and deep regression in Appendix H.4).
>
> - Theorem 2: empirical observations in deep meta-learning suggest that MAML possesses fast adaptation due to closer solution distances (Nichol et al., 2018; Zhou et al., 2020). Although the success of meta-learning is likely attributed to various factors,
>  we focus on this interesting observation, where rigorous analysis of this empirical statement is limited to date.
>
> ### [Q1: Learning the mean of a Gaussian prior]
>
> Assume each entry of $\mathbf{w} \in \mathbb{R}^d$, $\mathbf{w}_{(i)} \sim \mathcal{N}(\mu, \sigma) $, all data $(X_i, \mathbf{y}_i)$ in $D_i$ of task $i$ are i.i.d sampled and $\mathbf{y}_i \sim \mathcal{N}(X_i \mathbf{w}, \hat{\sigma})$, then for ERM, we have
>
> \begin{align}
>     \mathbf{w}_{ERM} = \arg\max_\mathbf{w} (\log p (D | \mathbf{w}) + \log p(\mathbf{w}) )
> \end{align}
>
> \begin{align}
>  \log p(\mathbf{w}) =  \log \frac{d}{\sqrt{2\pi} \sigma} - \frac{1}{2\sigma^2} ||\mathbf{w} - \mu I ||^2  = C_1 - \frac{1}{2\sigma^2} ||\mathbf{w} - \mu I ||^2 \ ~~~ \log p (D | \mathbf{w}) = - \frac{1}{2 \hat{\sigma}^2} \sum_i^N ||\mathbf{y}_i - X_i \mathbf{w}
> ||^2
> \end{align}
>
> However, in MAML,
>
> \begin{align}
>     \mathbf{w}_{MAML} = \arg\max_w \log \left[ \int p(\mathbf{w}) \left[ \prod_i^M \int p\left(D_i \mid \mathbf{w}^\prime_i \right) p (\mathbf{w}^\prime_i |  \mathbf{w} ) d \mathbf{w}^\prime_i \right] d \mathbf{w} \right] ~~~~
> s.t. \mathbf{w}^\prime_i &= \mathbf{w} - \alpha X_i^\top (X_i \mathbf{w} - \mathbf{y}_i)
> \end{align}
>
> While $p(\mathbf{w})$ is learning the mean of Gaussian prior, other parts of $\mathbf{w}$ of MAML may not. For example, $p(\mathbf{w}^\prime_{i} \mid \mathbf{w} ) = \mathcal{N} (\mathbf{w}^\prime_{i}; \mathbf{w}, \sigma)$ is conditioned on $\alpha$ and $X_i$.
>
> \begin{align}
>     \log p\left(D_{i} \mid \mathbf{w}^\prime_{i}\right) &= - \sum_i^N || (1 - \alpha X_i X_i^\top ) \mathbf{y}_i - X_i (1 - \alpha X_i^\top X_i) \mathbf{w} ||^2
> \end{align}
>
> In the linear case, there is some connection. For example, if data covariance is scaled identity then it reduces to MAP with alpha determined by variance of Gaussian prior. In general, the situation is more complex.
>
>
> ### [Q2: Why not consider task-specific ERM]
>
> We follow the usual meta-learning setting, where the model is required to learn a set of common initial weights for all tasks (seen and unseen).
> When inference on a specific task is required, the model then adapts to this presented task (in a limited number of steps, usually by gradient algorithms)
> from the learned initialization.
> Hence, ERM here is not attempting to fit a common set of weights for all tasks, but rather to learn a common initialization for fast adaptation.
> In particular, ERM does so by simply minimizing the average loss over all tasks to find such an initialization.
> As our analysis shows, this is not optimal but is a simple baseline to understand the working principles of MAML.
>
> ### [Q3: Estimator for $\alpha^*$]
>
> Thank you for the nice suggestion, we ran additional experiments on Omniglot with ANIL and compared them in Experiment 5.1.
> The experimental setup is illustrated in Appendix H.3.
>
> **[Summary of Revision to Paper]**
>
> Online estimation with ANIL. Thanks for your suggestion, we added the classification experiments on Omniglot 20-way 1-shot. To this end, we use online estimation based on the ANIL algorithm. Then we found that our estimated $\alpha^*$ converges faster than the default configuration in previous papers (see Experiment 5.1, figure 3).

---

> > ### Author Response · Authors · 2021-11-16
> > **Responce to Reviewer iGwz (2/2)**
> >
> > ### [Q4:  Is the relationship between regressor variance and $\alpha$ practically significant?]
> >
> > According to our experiment on Neural Tangent Kernel (NTK) (Figure 4(b)) activated by ReLU, it is significant in the sense that the same relationship holds quantitatively in this case.
> > Note that the NTK is an idealized version of an overparameterized neural network.
> > In deep learning, it is harder to define the regressor variance because the weights, as well as the features, are dynamically evolving during the training stage.

---

> > > ### Comment · Reviewer_iGwz · 2021-11-19
> > > **Post-response update**
> > >
> > > Thank you for your detailed response to my comments and questions. I'm glad you found the experimental suggestion helpful. I have raised my score from a 5 to a 6, partially because after reading the other reviews and responses I have an increased understanding of the paper's contributions towards theoretically verifying empirical observations of the placement of the MAML-learned initialization. However, I retain a low confidence score as I think I am not sufficiently qualified to evaluate the technical novelty of the theoretical machinery used in the paper; on this, I defer to e.g. Reviewer ui6t.

---

### Official Review · Reviewer_ui6t · 2021-11-03

**Correctness:** 3
**Technical Novelty And Significance:** 2
**Empirical Novelty And Significance:** 2
**Recommendation:** 5
**Confidence:** 4

**Main Review:**

Strong points:
The paper is well structured, experiments make sense and well coordinated with the hypothesis

For the part that is hard to prove the hypothesis strictly in mathematic method, the paper put much effort to design meaningful experiments to illustrate and ‘prove’ those points

Though coro1 is weak (since the reverse relation with variance of features rather than data points themselves is intractable), the authors tried to reveal it by several concrete examples (i.e. example1) and experiments.

The first result offered a method to get a better initialized value for alpha, which is valuable both in theoretical way and in practical use.

Weak points

Some of the results are not surprising and inspiring, lack of novelty.

The first result offered a method to get a better initialized value for alpha, which is valuable both in theoretical way and in practical use. But the latter two, i.e. the interpretation of alpha are not inspiring enough. Intuitively, alpha has relation with data(or say task) distribution. And the geometric interpretation is another way to formulate the goal of MAML.



**Summary Of The Paper:**

The paper put effort to understand the adaptation learning rate alpha. It gave a principled way to estimate the optimum alpha* minimizing MAML population risk. It showed the relation between alpha* and data distribution (variance of features). It also showed that alpha in MAML, compared to ERM, helped to minimize the total distance to all the tasks’ optima.



**Summary Of The Review:**

I don't think those results well pass the bar of ICLR conference. According to the above, I would suggest a weak reject (barely below the bar). PS: I am not sure if result one (find the better alpha) could be applied in practical tasks. My confidence is mostly on the theoretical part of the paper

---

> ### Author Response · Authors · 2021-11-16
> **Responce to Reviewer ui6t**
>
> We thank the reviewer for the comments and suggestions. Below, we answer each of the reviewer's questions and summarize the corresponding changes (in red) in the paper.
>
> ### [Q: The latter two, i.e. the interpretation of alpha are not inspiring enough ]
>
> - **Relation to data distribution**:
>     To the best of our knowledge, the precise dependency between the learning rate $\alpha$ and the data distribution has not been explored, even in very simple settings.
>     Corollary 1 of this paper is a quantitative result of Theorem 1 (the trace expectation of $2$-nd and $3$-rd powers of the covariance matrix in Equation 9 ) in this direction:
>     it relates the optimal inner-loop learning rate to data variance.
>     This insight can guide us to select good learning rates in different task environments, even beyond linear models (Omniglot in Section 5.1 and deep regression in Appendix H.4).
>
> - **Geometric interpretation**: In our paper, we provide a rigorous backing, in the linear basis setting, to an empirical observation that *MAML learns a closer solution to all task optima* (Nichol et al., 2018).
>     The latter observation led to various improvements of MAML (Yao et al., 2019; Zhou et al., 2020),
>     but it has not been theoretically investigated before.
>     To our best knowledge, this is the first work proving a version of this empirical statement.
>
> Yao, H., Wei, Y., Huang, J., \& Li, Z. (2019, May). Hierarchically structured meta-learning. In ICML.

---

> ### Author Response · Authors · 2021-11-20
> **Additional comments to the review**
>
> On the concerns on the theoretical front, we have some additional comments on the novelty of our results.
>
> - **Motivation:** To find a principled way to estimate $\alpha^*$, we presented a fine-grained analysis of $\alpha$ and improved the range result in (Bernacchia, 2020) to a value result (Theorem 1). Beyond the estimation, we also provided the interpretation of our estimation aiming to understand the optimization behavior of MAML from various angles.
>
> - **Technique:** It is non-trivial to find the relation between $\alpha^*$ and the statistics of the features. Since the original problem entails solving a complicated equation, we derived an approximation formula (Equation 9)  which can be readily applied and justified theoretically (Lemma 1 \& 2). Without dependence on data distribution, we proved that MAML always has a shorter average distance to tasks than ERM at any number of adaptation steps (Theorem 2).
>
>  - **Assumption:** Weak assumptions are made in terms of task optima and data while no specific data distribution is assumed in both theorems. Proposition 3 illustrated that Theorem 2 doesn't depend on the data distribution.

---

### Official Review · Reviewer_6tyw · 2021-11-05

**Correctness:** 3
**Technical Novelty And Significance:** 3
**Empirical Novelty And Significance:** 3
**Recommendation:** 5
**Confidence:** 2

**Main Review:**

[Pros]

1. This paper is well written with numerous theoretical findings supported by derivations in the appendix.

2. Their method of how to properly set the inner-learning rate and their implication can benefit the meta-learning community.


[Cons]

1. As far as I understand, the derivation is based on the assumption that the adaptation only takes place at the last linear layer. This is clearly different from original MAML where adaptation takes place at all the layers. Also, as far as I understand, the analysis is only valid for regression problems, so I wonder how to make use of the findings to help classification problems.

2. Similarly, the experimental results are done only under the synthetic environment, which limits the practical impact of this paper. Can you extend the experiment to more realistic scenario such as miniImageNet few-shot classification?

3. In my opinion, the intuition behind their theoretical findings is a little bit straightforward, although understanding the phenomena is very important. For example, it is seems easy to imagine how nonzero inner-learning rate can help the shared initialization to stay robust against different task sampling. Of course it is always good to formalize such an intuition, but I just wonder how significant the importance is.

[Simple questions (minor)]

1. From reading the paper, I understood that how alpha=0 (ERM) can fail in Figure 1, but I do not understand why too large alpha fails as well, in terms of staying robust against the biased task sampling (the red circle). Can you provide an intuition in relation to the derivation?

**Summary Of The Paper:**

This paper explores a question of how to theoretically derive a form of optimal MAML inner-learning rate and how to interpret its meaning. In order to do this, the authors assumes that the given problem is linear regression with some possibly nonlinear feature transformations. Then they derive the form of optimal MAML inner-learning rate (alpha). Further, they demonstrate that MAML with proper inner-learning rate stabilizes the meta-learning of the shared initialization, such that the shared initialization can generally minimize the distance from all the given task predictors, whereas the baseline ERM is sensitive to the density of the task distribution. The experimental results also demonstrate their derivation and intuition.

**Summary Of The Review:**

The submission formalizes the MAML inner-learning rate theoretically, and it can benefit meta-learning community to some extent. However, I'm not fully convinced if the finding is practically important, or something new (in terms of intuition) as well.

---

> ### Author Response · Authors · 2021-11-16
> **Responce to Reviewer 6tyw**
>
> We thank the reviewer for the comments and suggestions. Below, we answer each of the reviewer's questions and summarize the corresponding changes (in red) in the paper.
>
> ### [Q1 & Q2: Experiment to more realistic scenario]
>
> We appreciate the reviewer's concern about the practical impact of our $\alpha^*$ estimation, due to its dependence on the feature matrix.
> However, understanding MAML in linear models (Finn et al., 2019; Collins et al., 2020) is still an important question.
> Hence, we chose such a starting point to build our theory.
> While the linear setting appears restrictive, it has been shown that theoretical understanding in linear settings can be generalized to deep learning when hidden layers go to infinite wide (Over-parameterized models, e.g. NTK (Jacot et al., 2018) ).
> Lastly, we show empirically that the insights revealed by our analysis in the linear setting can help one determine good $\alpha$ values even for deep learning applications (results are shown in Appendix H.1 & H.4).
>
> **[Summary of Revision to Paper]**
>
> Classification experiments. As suggested by one of the reviewers, we added the classification experiments on Omniglot 20-way 1-shot. To this end, we use online estimation based on the ANIL algorithm. Then we found that our estimated $\alpha^*$ converges faster than the default configuration in previous papers (see Experiment 5.1, figure 3).
>
>
> ### [Q3: Theoretical findings are straightforward and how significant the importance is]
>
> We emphasize that our main results are: 1) an explicit estimation of the optimal inner learning rate $\alpha^*$ (Theorem 1), and how it is affected by statistics of the data (ratio of moments), and 2) how MAML learns a solution that is provably closer in average distance to the individual task optima (Theorem 2).
> We believe that these results are quantitative and that they are not a priori straightforward.
>
> Result 1) is important because it gives a quantitative method to choose the inner learning rate for MAML using linear basis models.
> To the best of our knowledge, this is new.
> Furthermore, we demonstrate that this quantitative estimate can aid us in choosing inner learning rates
> even for deep learning applications beyond our chosen analytical setting,
> as shown in Section 5.1 and Appendix H.4.
>
> Thus we believe that this is a meaningful starting point to develop principled hyper-parameter selection methods for MAML applications.
> Result 2)'s significance lies in the confirmation, in a limited setting, of previous empirical observations.
> This may contribute to devising algorithms beyond our setting like distance-aware, task-clustered meta-learning (Yao et al., 2019; Zhou et al., 2020).
>
> ### [Minor Q: Too large alpha fails as well]
>
> Note that $\alpha$ is the learning for a gradient-based adaptation step.
> As with all gradient-based optimization algorithms, the descent of the loss function can only be guaranteed
> locally, i.e. for small step sizes.
> When $\alpha$ is too large, the inner loop update no longer points in the steepest descent direction,
> thus its performance diminishes.

---

### Official Review · Reviewer_pMe3 · 2021-11-08

**Correctness:** 4
**Technical Novelty And Significance:** 3
**Empirical Novelty And Significance:** 2
**Recommendation:** 5
**Confidence:** 4

**Main Review:**

The paper provides new insights into the theory of MAML. I have two main concerns/questions, which are listed below:

- First, the authors assume in equation (4) that the datasets used for updating the model and validating it are the same. This is an unusual assumption to me: if you know the true label to update the model, then validating the model on it might not make sense. I appreciate it if the authors explain how much the analysis depends on this assumption and whether it can be relaxed or not.

- Theorem 2 on ERM vs. MAML seems very similar to what (Collins et al., 2020) studied in their paper. I appreciate it if the authors explain how their results differ from the results of that paper.



**Summary Of The Paper:**

This paper studies the MAML problem. In particular, authors consider mixed linear regression problem and provide two sets of results:
- First, they characterize the optimal learning rate for test time, i.e., the stepsize that is used for updating the model at test time, when the number of tasks goes to infinity. More formally, they show that the optimal learning rate depends on the distribution of the feature vector.
- Second, the authors characterize the gain obtained from MAML in comparison with ERM. The obtained bound matches with basic intuitions. For instance, it increases with the test time learning rate up to some $\bar{\alpha}$ and also decreases as the number of deputation steps increases.

**Summary Of The Review:**

Overall, I find the paper's attempt to shed light on the theory of MAML interesting. I have two main concerns on the test set/validation set separation and comparison with another result in the literature. I am open to raising my score based on the authors' responses.

---

> ### Author Response · Authors · 2021-11-16
> **Response to Reviewer pMe3**
>
> We thank the reviewer for the comments and suggestions.
> Below, we answer each of the reviewer's questions
> and summarize the corresponding changes (in red) in the paper.
>
> ### [Q1 test set/validation set separation]:
>
> For simplicity, we use the same dataset for training and validation in loss computation.
> This is our chosen setting for analysis, and would not affect results for sufficiently large batch sizes.
> Moreover, we have provided proof for the case where training/validation data are separate in Appendix H.1 Corollary 2.
>
> ### [Q2 Comparison betweenTheorem 2 and (Collins et al., 2020)]:
>
> Our theoretical statement is different from (Collins et al., 2020)
>
> - In their work, they prove that MAML prioritizes "hard" tasks by defining the "task difficulty" with the minimum eigenvalue of the Hessian of task objective function. They show that when the task difficulty distribution is not uniform, MAML learns initial weights closer to hard task optima.
> - In our paper, we confirmed an empirical observation that MAML learns a closer solution to all task optima (Nichol et al., 2018). To the best of our knowledge, this is the first proof of such types of statements, albeit under a simplified linear basis model setting.
>
> The key difference is that our results are consistent with existing empirical results on MAML.
> Moreover, we do not require the definition of task hardness (e.g. based on eigenvalues or condition number),
> which is non-trivial in settings beyond linear regression.
>
> **[Summary of Revision to Paper]**
>
> Discussion about the difference to (Collins et al., 2020). We emphasized it in the last paragraph of Section 3.2 "We note that 'task hardness' may not always be easy to define, especially for non-linear cases (Collins et al., 2020).
> Here, we instead focus on directly analyzing the geometric distance Theorem 2, which has substantiated the aforementioned findings in optimization behavior from different angles."

---

### Author Response · Authors · 2021-11-18
**To all reviewers**

We appreciate all the reviewers and their efforts. Here we again list **our theoretical contributions** to the meta-learning community as follows.

- We present a principled way to estimate the optimal adaptation learning rate $\alpha$ of MAML in the linear basis setting. This is a value estimation compared to a range estimation in (Bernacchia, 2020) and the theoretical insights are distinct. (Please refer to Appendix H.5)
- We also interpreted the statistical meaning of this optimal $\alpha$ in terms of the input data. We show the concrete form of the statistical dependency (inverse of the data variance). This insight may help one explore optimal $\alpha$ beyond the linear setting.
- We prove a result consistent with empirical observations that MAML tends to find a solution closer in average to all individual task optima. This empirical result helped to design many MAML improvements, such as task-grouped (Yao, 2019) and multiple initializations (Zhou, 2020).

As for practical validation, we adopted reviewers' suggestions and added a deep classification experiment in the main paper.

Based on each reviewer's concerns, we summarize the **main changes of our paper** below:

- With the comment of **reviewer pMe3**, the following changes are presented in our revised paper.
   * Clearer illustration of the motivation of dataset separation setting where all data in task is used for training and validation. We also show a similar result for the case where training/validation are separate in Appendix H.1 Corollary 2.
   * We rewrote the conclusion of our Theorem 2 and emphasized the difference to (Collins et al., 2020) in the last paragraph of Section 3.2


- With the comment of **reviewer 6tyw**, the following changes are presented in our revised paper.
  * To illustrate the practical benefit of Theorem 1 in the realistic dataset, we added new experiments on Omniglot 20-way 1-shot classification in Section 5.1 (figure 3).

- With the comment of **reviewer ui6t**, the following changes are presented in our revised paper.
   * We rewrote the significance of our Theorem 2 and discussed the differences to other literature.

- With the comment of **reviewer iGwz**, the following changes are presented in our revised paper.
   * We added the missing loosely related works in our Related work section (at paragraph 1 line 3).
   * We adopted the nice suggestion where we use online estimation with ANIL (Raghu et al., 2019) to evaluate Theorem 1 on the deep classification experiments.

- With the comment of **reviewer 9gLM**, the following changes are presented in our revised paper.
   * We included the optimal LR (Eq 145) for empirical experiments in Section 5.1 (line 5).
   * We revised the caption in Figure 5 for a clearer presentation.

[1] Collins, L., Mokhtari, A., & Shakkottai, S. (2020). Why does MAML outperform ERM? an optimization perspective. arXiv preprint arXiv:2010.14672.

[2] Bernacchia, A. (2020, September). Meta-learning with negative learning rates. In ICLR.

[3] Yao, H., Wei, Y., Huang, J., & Li, Z. (2019, May). Hierarchically structured meta-learning. In ICML.

[4] Zhou, P., Zou, Y., Yuan, X., Feng, J., Xiong, C., & Hoi, S. C. (2020). Task similarity aware meta learning: Theory-inspired improvement on maml. In 4th Workshop on Meta-Learning at NeurIPS.

[5] Raghu, A., Raghu, M., Bengio, S., & Vinyals, O. (2019, September). Rapid Learning or Feature Reuse? Towards Understanding the Effectiveness of MAML. In ICLR.

---

### Decision · Program_Chairs · 2022-01-20

**Decision:**

Accept (Poster)

**Comment:**

After carefully reading all reviews and rebuttal, I actually think the paper provides sufficient new insight in understanding MAML that is worth being accepted. I want to thank the authors for actively engaging with the reviewers, and providing sufficient changes to the paper in order to clarify and improve its contributions.

Theoretical results tend to be harder to judge, as they often need to happen under assumptions that make them tractable. Nevertheless they provide intuitions and understanding of the underlying principle that end up having an impact even in more realistic scenarios where these assumptions might not hold. I think this is such a scenario, and I think better understanding the relationship between ERM and approaches as meta-learning is important for the field moving forward.